EMBO
reports

# A low-level Cdkn1c/p57$^{kip2}$ expression in spinal progenitors drives the transition from proliferative to neurogenic modes of division

Baptiste Mida [1], Nathalie Lehmann[2], Rosette Goïame[2], Fanny Coulpier[3], Kamal Bouhali[2], Isabelle Barbosa[2], Hervé le Hir [2], Morgane Thomas-Chollier[2,4], Evelyne Fischer [5,6] & Xavier Morin [5,6]

## Abstract

During vertebrate neurogenesis, a transition from symmetric proliferative to asymmetric neurogenic divisions is critical to balance growth and differentiation. Using single-cell RNA-seq data from the chick embryonic neural tube, we identify the cell cycle regulator Cdkn1c as a key regulator of this transition. While Cdkn1 is classically associated with neuronal cell cycle exit, we show that its expression initiates at low levels in neurogenic progenitors. Functionally targeting the onset of this expression impacts the course of neurogenesis: Cdkn1c knockdown impairs neuron production by favoring proliferative symmetric divisions. Conversely, inducing a low-level Cdkn1c misexpression in self-expanding progenitors forces them to prematurely undergo neurogenic divisions. Cdkn1c exerts this effect primarily by inhibiting the CyclinD1-CDK4/6 complex and G1 phase lengthening. We propose that Cdkn1c acts as a dual driver of the neurogenic transition whose low level of expression first controls the progressive entry of progenitors into neurogenic modes of division before higher expression mediates cell cycle exit in daughter cells. This highlights that the precise control of neurogenesis regulators' expression sequentially imparts distinct functions essential for proper neural development.

**Keywords** Neurogenesis; Cell Cycle; Modes of Division; Cdkn1c; Single-cell Transcriptomics
**Subject Categories** Development; Neuroscience

## Introduction

The vertebrate central nervous system (CNS) is a complex assembly of thousands of cell types, which are organized in an exquisite manner to form functional neural circuits. This amazing diversity develops through the sequential production of neuronal and glial cells from a limited pool of neuroepithelial stem cells (also called neural progenitors) (Noctor et al, 2001; Taverna et al, 2014). Precise coordination between growth and differentiation during the neurogenic period is paramount to produce the correct number of neural cells "at the right place and time" to ensure the formation of neural circuits. To achieve this complex organization, both the number of times progenitors enter a cell cycle and the proportion of progenitors that exit the cell cycle to produce neural cells after each round of division are crucial. After initial amplification via proliferative symmetric divisions, the progenitor pool progressively switches to neurogenic modes of division: neurogenic progenitors first perform asymmetric divisions, allowing the self-renewal of one daughter cell while its sibling commits to differentiation, and later switch to terminal symmetric divisions producing two differentiating neural cells (Taverna et al, 2014). By analyzing the fate of pairs of sister cells, clonal analysis approaches in various models have helped to identify and quantify these different modes of division in different wild-type and mutant contexts (Alexandre et al, 2010; Appiah et al, 2023; Hevia et al, 2022; Morin et al, 2007; Royall et al, 2023; Tozer et al, 2017). These studies have contributed to establish that the balance between these different modes of division is a crucial regulator of the pace of neurogenesis and of the relative size of different neuronal pools. In addition, analysis of the content of clones over longer periods of time in the mouse and rat embryonic cortices indicates that progenitors that have undergone a neurogenic division do not normally reenter a proliferative state, suggesting an irreversible switch in competence (Gao et al, 2014; Noctor et al, 2004). This switch was proposed to result from differences in transcriptomic and chromatin landscapes between proliferative and neurogenic progenitors (Aprea et al, 2013; Arai et al, 2011; Haubensak et al, 2004; Iacopetti et al, 1999; Saade et al, 2013; Albert et al, 2017). As an example, expression of the Tis21/Btg2/Pc3 transcription factor (thereafter Tis21) is initiated during the switch from proliferation

[1]Institut de Biologie de l'Ecole Normale Supérieure (IBENS), CNRS, Inserm, Ecole Normale Supérieure, Sorbonne University, Collège Doctoral, Paris 75005, France. [2]Institut de Biologie de l'ENS (IBENS), Département de biologie, Ecole Normale Supérieure, CNRS, INSERM, PSL Research University, Paris 75005, France. [3]GenomiqueENS, Institut de Biologie de l'ENS (IBENS), Département de biologie, Ecole Normale Supérieure, CNRS, INSERM, PSL Research University, Paris 75005, France. [4]Institut universitaire de France (IUF), Paris, France. [5]Institut de Biologie de l'Ecole Normale Supérieure (IBENS), CNRS, Inserm, Ecole Normale Supérieure, PSL Research University, Paris 75005, France. [6]These authors contributed equally: Evelyne Fischer, Xavier Morin. ✉E-mail: evelyne.fischer@ens.fr; xavier.morin@ens.fr

to neurogenesis from the forebrain (Iacopetti et al, 1999) to the spinal neural tube, in both the mouse (Haubensak et al, 2004) and chick models (Hämmerle et al, 2002; Saade et al, 2013). These observations suggest that self-expanding and neurogenic progenitors correspond to two distinct and successive stages in the neural developmental program.

Here, we generated single-cell transcriptomics data (scRNCA-seq) from embryonic chick spinal neural tube and identified several genes differentially expressed during the progression from proliferative to neurogenic progenitor states. Cdkn1c (Cyclin-dependent kinase inhibitor 1c/p57$^{kip2}$), a member of the CDK inhibitor (CDKI) family that also comprises Cdkn1a (p21$^{cip1}$) and Cdkn1b (p27$^{kip1}$), emerged as one of the most differentially expressed between proliferative and neurogenic progenitors. In the mammalian cortex, p57$^{kip2}$ plays a key role in regulation of the proliferation and differentiation of embryonic and adult neural stem cells (Colasante et al, 2015; Furutachi et al, 2013; Imaizumi et al, 2020; Itoh et al, 2007; Joseph et al, 2009, 2003; Mairet-Coello et al, 2012; Tury et al, 2011) and it was shown to promote gliogenesis during late embryogenesis and early postnatal stages (Tury et al, 2011). While in the spinal cord, Cdkn1c was previously described as a regulator of cell cycle exit via its expression in newborn neurons (Gui et al, 2007; Mairet-Coello et al, 2012; Tury et al, 2011), our results showed that it is already expressed at low levels in neurogenic progenitors. Using loss-of-function and controlled overexpression experiments, we demonstrated that the onset of Cdkn1c expression in cycling progenitors is a driver of the transition toward neurogenic modes of division. Mechanistically, Cdkn1c acts in progenitors by increasing the duration of the G1 phase of the cell cycle, mainly via inhibition of the CyclinD1/CDK6 complex. Our findings suggest a dual role for Cdkn1c: a low Cdkn1c expression initially promotes neurogenic modes of division before its higher expression facilitates cell cycle exit in newborn neurons.

# Results

## Transcriptional signature of the neurogenic transition

To identify potential drivers of the transition from the proliferative to the neurogenic state, we produced single-cell transcriptomics (scRNAseq) data from the cervical neural tube region of chick embryos at HH18 (E2.75), when proliferative and neurogenic modes of division are about equally represented (Bonnet et al, 2018; Saade et al, 2013). To overcome limitations in the number of reads assigned to genes resulting from the poor annotation of many genes' 3'UTRs in the chick genome, we built an improved genome annotation based on the galGal6 reference and bulk long-read RNA-seq from the same tissue (embryonic spinal neural tubes, HH18, see "Methods"). This considerably improved the assignment of scRNAseq reads to genes and therefore the reliability of expressed gene counts in each cell.

We restricted our analyses from the original dataset to central nervous system-related cells (1878 cells) (Fig. 1A), excluding neural crest and mesoderm derivatives (see "Methods"). In UMAP representations (Fig. 1B), neural cells do not arrange in isolated clusters. We therefore defined a scoring system based on the levels of expression of a list of progenitor and neuron-specific genes in each cell (Progenitor genes = *Sox2, Notch1, Rrm2, Hmgb2, Cenpa, Ube2c, Hes5*; Neuron genes = *Tubb3, Stmn2, Stmn3, Nova1, Rtn1, Mapt*). This system shows the highest progenitor (P) and neuron (N) scores at opposite ends of the UMAP representation, with intermediate values of both scores in between (Fig. 1B), indicating that the progression from progenitor to neuron spontaneously emerges as a strong discriminating factor. Importantly, the expression of *Tis21* peaks in the region containing cells with intermediate values of the progenitor score and low neuron score (Fig. 1B). Hence, this region likely hosts neurogenic progenitors, in agreement with clonal analyses showing that progenitors undergo a progressive maturation from proliferative to neurogenic states. We refined this analysis through pseudotemporal classification of these cells to identify gene clusters with similar transcriptomic profiles along the pseudo-time axis (Fig. 1C). Interestingly, the gene cluster that contained *Tis21* also contained genes encoding proteins with known expression and/or functions at the transition from proliferation to differentiation, such as the Notch ligand Dll1 (Henrique et al, 1995), the bHLH transcription factors Hes6 (Fior and Henrique, 2005), NeuroG1 and NeuroG2 (Lacomme et al, 2012; Sommer et al, 1996) and the coactivator Gadd45g (Kawaue et al, 2014). We used this list of genes to define a "neurogenic progenitor" (PN) score (Fig. 1D) and performed a differential expression analysis based on P, N, and PN scores (Fig. 1E). The top ten differentially expressed genes between PN and the other two populations contain five of the six genes used to define the PN score. In addition, it includes (a) three genes with unknown function in the neurogenic transition (*Zc3h12c, LOC107051857, Bambi*), (b) the centrosomal gene Ninein (*Nin*), whose differential regulation at that stage has previously been described (Zhang et al, 2016), and (c) *Cdkn1c* (Fig. 1E).

*Cdkn1c/p57kip2*, an inhibitor of Cyclin/CDK complexes classically involved in G1 phase progression and G1 to S phase transition (Hatada and Mukai, 1995; Lee et al, 1995; Matsuoka et al, 1995; Taniguchi et al, 1997), was one of the strong candidates emerging from our analyses. Cdkn1c expression is associated with the choice of cells to exit the cell cycle in many developmental and postnatal contexts (Creff and Besson, 2020). In addition, Cdkn1c also controls specific cell fates by directing differentiation in various tissues via CDK-independent mechanisms (Campbell et al, 2020; Duquesnes et al, 2016).

The analysis of allele-specific mouse mutants of this imprinted gene has shown that both the paternal and maternal alleles contribute, though differently, to normal corticogenesis (Imaizumi et al, 2020). In addition, clonal analyses suggest that Cdkn1c regulates cortical development through distinct cell-autonomous and non-cell-autonomous mechanisms that respectively promote and inhibit growth (Laukoter et al, 2020). The Cdkn1c protein is present in some progenitors in the mammalian embryonic cortex (Mairet-Coello et al, 2012; Tury et al, 2011), and *Cdkn1c* transcripts are specifically enriched in neurogenic Tis21-positive progenitors (Arai et al, 2011). An increased proliferation of cortical progenitors associated with a global shortening of G1 phase length has been reported in the *Cdkn1c* knock-out mouse (Mairet-Coello et al, 2012). Conversely, in the context of cerebral cortex-specific inactivation of the *Aristaless-related homeobox* (ARX) gene, upregulation of Cdkn1c in the ventricular and subventricular zones was associated with reduced proliferation of intermediate progenitor cells at E12/14 (Colasante et al, 2015).

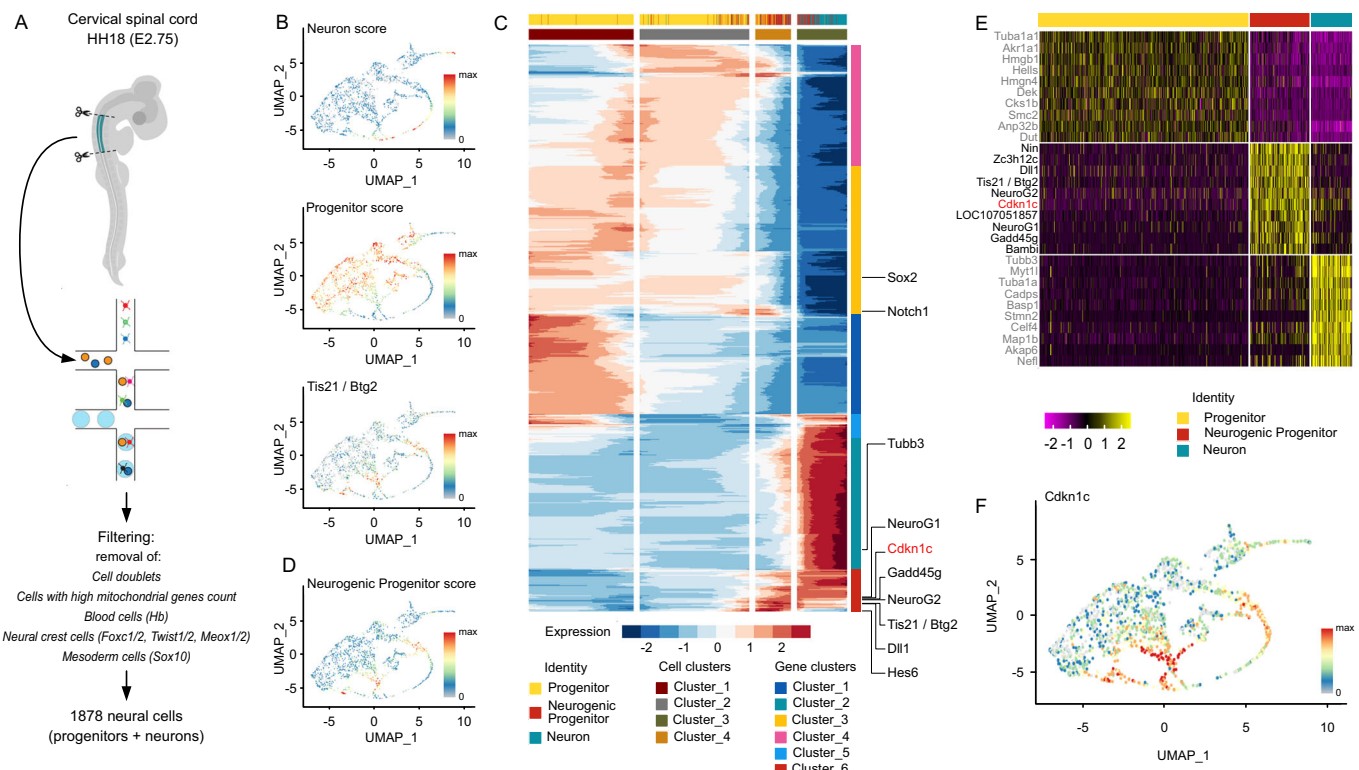

**Figure 1. scRNAseq data analyses from embryonic neural tube led to the identification of Cdkn1c as a potential regulator of the transition of modes of division.**

(A) Scheme of the dissection and protocol for scRNA seq generation from chick cervical spinal tube at E2.75. (B) Visualization of progenitor and neuron scores and of *Tis21/Btg2* expression on the UMAP representation of 1878 chick cervical spinal tube neural cells. (C) Representation of the pseudotime analysis of the chick scRNAseq dataset. The heatmap shows six clusters of genes with a similar pattern represented on the vertical axis and four cell clusters on the horizontal (pseudotemporal) axis. A subset of genes used to define Progenitor, Neuron, and Neurogenic Progenitor scores is indicated on the right side of the heatmap, illustrating that the three signatures relate to different gene clusters. Cdkn1c is found in the same gene cluster as Neurogenic Progenitor genes. The top horizontal row indicates the cell subtype assigned to each cell along the temporal axis. The blue/red color gradient represents the value of the Z-score. (D) Visualization of the neurogenic progenitor score on the UMAP representation of 1878 chick cervical spinal tube neural cells. (E) Heatmap of the ten most differentially expressed genes between Progenitor, Neurogenic Progenitor, and Neuron populations. (F) Visualization of *Cdkn1c* expression on the UMAP representation of 1878 chick cervical spinal tube neural cells.

During spinal cord development, Cdkn1c expression was mostly described in newborn neurons (Gui et al, 2007; Tury et al, 2011). Contrasting with these data, our pseudotime analyses and UMAP visualization in the chick spinal cord show an onset of its expression in the Tis21-positive neurogenic progenitor population (Fig. 1C,F). However, *Cdkn1c* expression is maintained longer and transiently peaks at high levels after *Tis21* expression is switched off. Given that Tis21 is no longer expressed in neurons (Iacopetti et al, 1999), this suggests that Cdkn1c expression is transiently upregulated in nascent neurons before fading off in more mature cells. This peculiar expression profile suggests a dual role of Cdkn1c during neurogenesis: its onset of expression in cycling progenitors would first favor a transition from proliferative to neurogenic modes of division, before higher levels of its expression in daughter cells fated to become neurons would drive them out of the cell cycle.

## Progressive increase in Cdkn1c/p57^kip2 expression underlies different cellular states in the embryonic spinal neural tube

To test this hypothesis, we explored the dynamics of *Cdkn1c* expression by in situ hybridization in the chick embryonic spinal

tube (Fig. 2A) during the neurogenic transition. While *Cdkn1c* was not expressed at E2, before neurogenesis really starts, its transcript was detected at E3 and E4, when neurogenesis is well underway, as underscored by the expression of the neuronal maker HuC/D in the mantle zone (see lower panels of Fig. 2A). At E3, the transcript was expressed at low levels in a salt and pepper fashion in the ventricular zone, where the cell bodies of neural progenitors reside (Saade and Martí, 2025). One day later, at E4, this salt and pepper expression was still detected in the ventricular zone, while it markedly increased in the region of the mantle zone that is immediately adjacent to the ventricular zone. This region is enriched in nascent neurons on their way to differentiation that are still HuC/D negative. In contrast, the transcript was completely excluded from the more basal region of the mantle zone, where mature HuC/D-positive neurons accumulate. This is consistent with the dynamics of the in silico profile described above.

We then checked whether the Cdkn1c protein is translated in the population of progenitors in which the transcript is detected. In the absence of functional antibodies in the chick embryo, we used a clustered regularly interspaced short palindromic repeats (CRISPR)/Cas9-based somatic knock-in strategy to insert an array of six Myc tags at the C-terminus of *Cdkn1c* (Figs. 2B and EV1A,B

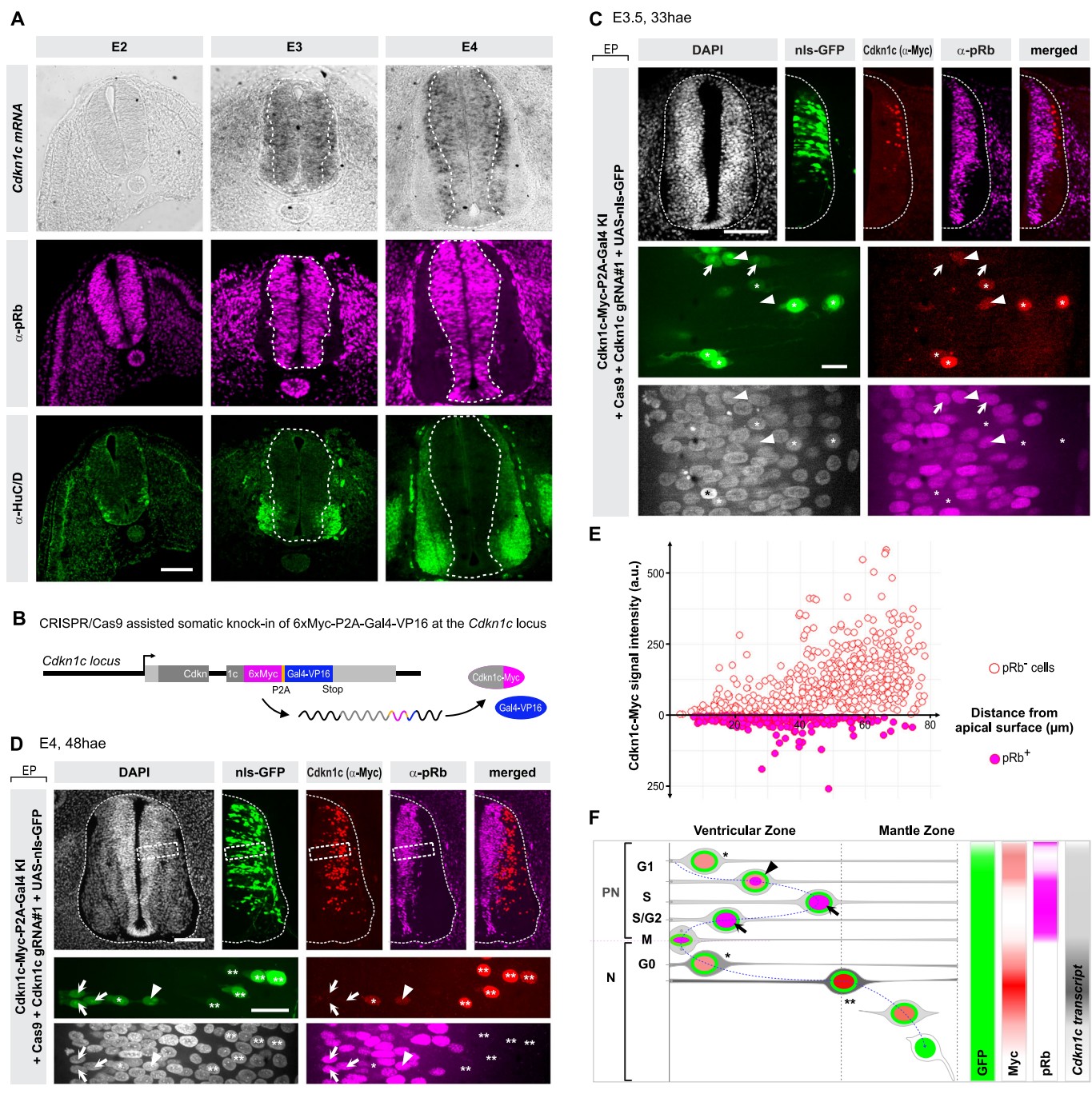

**A**

**B** CRISPR/Cas9 assisted somatic knock-in of 6xMyc-P2A-Gal4-VP16 at the *Cdkn1c* locus

**C** E3.5, 33hae

**D** E4, 48hae

**E**

**F**

and "Methods"). The Myc tags insertion approach offers a direct read-out of the presence of the protein, and should report any cell cycle-dependent stabilization or degradation of Cdkn1c. The Myc tags were immediately followed by a P2A pseudocleavage site and the *Gal4-VP16* transcription factor sequence (Figs. 2B and EV1A,B) whose simultaneous translation can be used to activate the transcription of a stable fluorescent reporter from a UAS promoter. This allows us to identify both Cdkn1c-Myc-positive cells and cells in which Cdkn1c-Myc is no longer present but has previously been expressed.

We electroporated the knock-in (KI) vector together with a plasmid expressing Cas9 and guide RNA sequences (gRNAs) targeting the *Cdkn1c* C-terminus and a UAS-nls-GFP (nuclear localization sequence Green Fluorescent Protein) reporter plasmid in the neural tube of E2 chick embryos. At E3 (24 hae), we observed a GFP signal in the electroporated side of the embryo using three different *Cdkn1c* gRNAs, whereas no signal was observed when a control gRNA that does not target any chick sequence was used, indicating successful and specific knock-in events (Fig. EV1C). As the KI efficiency of guide #2 was weak, only guides #1 and #3 were

◄  **Figure 2.   Dynamics of Cdkn1c expression during spinal tube neurogenesis.**

(A) mRNA expression of *Cdkn1c* increases during neurogenesis in the chick embryonic neural tube. In situ hybridization of *Cdkn1c* (top row) and immunostainings for pRb (middle row, magenta) and HuC/D (bottom row, green) on cryosections of the thoracic region of chick embryonic neural tube at sequential days during development (from left to right: E2, E3, and E4, respectively). For each stage, light and multichannel confocal fluorescence imaging are from a single section. Scale bars: 50 μm. The white dotted lines in E3 and E4 images delineate the limit between the ventricular zone and the mantle zone. (B) Schematic representation of the modified *Cdkn1c* locus upon 6xMyc and Gal4-VP16 knock-in. The insertion of the P2A pseudocleavage site (orange) between the 6xMyc and Gal4-VP16 sequences ensures that Cdkn1c-6xMyc and Gal4-VP16 are present as independent proteins. (C, D) Myc-tagged Cdkn1c protein is detected at low level in cycling progenitors. Somatic knock-in of 6xMyc-tags fused to the C-terminus of *Cdkn1c* allows the visualization of Cdkn1c protein (Myc, red) on E3.5 (C) and E4 (D) transverse vibratome sections. The inclusion of a Gal4-VP16 transcription factor in the knock-in construct allows to identify all the cells which express or have previously expressed Cdkn1c via the stable UAS-nls-GFP reporter (green). Staining with anti-phospho-Rb antibody (pRb, magenta) labels cycling progenitors from late G1 to M phase. The bottom part of (C) shows a close-up at high magnification from a different section than the lower magnification in the upper panel, whereas in (D) the bottom panels show a close-up of the region highlighted by a dashed rectangle in the top panels. The key to the meaning of asterisks, arrows, and arrowheads pointing to cells with different combinations of the markers is illustrated in the scheme in (F). Scale bars: 50 μm and 10 μm in close-ups. The top row shows maximal projections of 4 μm (C) and 5 μm (E) confocal z-stacks acquired with a ×20 objective; close-ups in the bottom rows are maximal projections of 4 μm (C) and 5 μm (E) confocal z-planes acquired with a ×100 objective. (E) Quantification of Cdkn1c-Myc expression in relationship with pRb expression along the apico-basal axis. Quantification of the Myc signal intensity from a Cdkn1c-Myc knock-in insertion was performed in vibratome sections from embryos electroporated with the donor vector, gRNA#1, and a UAS-nls-GFP reporter (C). Knock-in cells that express or have expressed Cdkn1c were identified on the basis of Myc and/or GFP positivity. The Myc signal intensity in individual cells is plotted on the Y axis toward the upper (pRb +) and lower (pRb−) parts of the graph as a function of their position along the apico-basal axis (X axis). Data are from 942 cells from 12 sections from 5 embryos. (F) Scheme summarizing the proposed dynamic expression of *Cdkn1c* transcript and protein in cycling progenitors and newborn neurons, as deduced from scRNAseq, in situ hybridization, and somatic knock-in experiments. In a subset of neurogenic progenitors (PN), the *Cdkn1c* transcript is expressed at low levels (light gray), before it peaks transiently in newborn neurons (N, dark gray) and fades off in more mature neurons (N, white). Cdkn1c protein, visualized with the anti-Myc signal (red) is present at low levels in early G1 in neurogenic progenitors (light red nuclear signal, black asterisk) and shortly overlaps with pRb staining in late G1 (black arrowhead, light red and magenta nuclear signals). The Myc signal disappears in later phases of the cell cycle during which pRb is still detected (black arrow, magenta nuclear signal). In newborn neurons, the Cdkn1c/Myc signal is initially detected at a low level (single black asterisk, light red nucleus) and later peaks at its maximal intensity (double asterisks, dark red nucleus) during the early phases of differentiation, before fading out in mature neurons. pRb is absent in the neuronal population. The GFP signal (green) expressed from the UAS reporter is detected throughout this temporal sequence. hae hours after electroporation.

used for further characterization of Cdkn1c-Myc expression, and both gave identical results in all subsequent analyses. Specificity of insertion at the *Cdkn1c* locus was further confirmed by several approaches: first, we did not obtain any signal when a donor vector that lacked arms of homology to the *Cdkn1c* locus was used in combination with gRNA#1 and #3 (Fig. EV1C); second, western blot analyses using an anti-Myc antibody revealed a strong band with both gRNA#1 and #3, and no signal at all with a control gRNA (Fig. EV1D).

To ascertain that *Cdkn1c* is translated in neural progenitors, we used an anti-pRb antibody, recognizing a phosphorylated form of the Retinoblastoma (Rb) protein that is specifically detected in cycling cells (Gookin et al, 2017; Moser et al, 2018; Spencer et al, 2013), including neural progenitors of the developing chick spinal cord (Molina et al, 2022). In the ventricular zone of transverse sections at E3.5 (33 hae) (Fig. 2D) and E4 (48 hae) (Figs. 2C and EV1E), we detected triple Cdkn1c-Myc/GFP/pRb-positive cells (arrowheads in Fig. 2C,D), providing direct evidence for the Cdkn1c protein in cycling progenitors. We also observed many double GFP/pRb-positive cells that were Myc negative (arrows in Fig. 2C,D). The observation of UAS-driven GFP in these pRb-positive cells is evidence for the translation of Gal4 and therefore provides a complementary demonstration that the *Cdkn1c* transcript is translated in progenitors. The absence of Myc detection in these double GFP/pRb-positive cells also suggests that Cdkn1c/Cdkn1c-Myc stability is regulated during the cell cycle.

Finally, we observed double Myc/GFP-positive cells that were pRb-negative (Figs. 2C and EV1G, asterisks). One characteristic of Rb phosphorylation as a marker of cycling cells is a period in early G1 during which it is not detectable, as described in several cell types (Gookin et al, 2017; Moser et al, 2018; Spencer et al, 2013), including chick spinal cord neural progenitors (Molina et al, 2022).

Using a method that specifically labels a synchronous cohort of dividing cells in the neural tube, we similarly observed a period in early G1 during which pRb is not detectable in some progenitors at E3 (see Fig. EV2 and "Methods"). Hence, the double Myc/GFP-positive and pRb-negative cells may correspond to progenitors in early G1. Alternatively, they may be nascent neurons whose cell body has not yet translocated basally (see Fig. 2F). Finally, we observed a pool of GFP-positive/pRb-negative nuclei with a strong Myc signal in the region of the mantle zone that is in direct contact with the ventricular zone (VZ), corresponding to the region where the transcript is most strongly detected (see Fig. 2A). This pool of cells with a high Cdkn1c expression likely corresponds to immature neurons exiting the cell cycle and on their way to differentiation (Fig. 2C,D, double asterisks). In addition, a few Myc-positive cells were located deeper in the mantle zone, where the transcript is no longer present, suggesting that the protein is more stable than the transcript. The quantification of the level of expression of Cdkn1c-Myc along the apico-basal axis in relationship to pRb immunoreactivity confirms that Cdkn1c is expressed at low level in the ventricular zone in both pRb-positive and negative cells, and that it peaks in more basal, pRb-negative cells (Fig. 2E). Of note, similar results were obtained with gRNA#3 (Fig. EV1E,F).

In summary, our dual Myc and Gal4 knock-in strategy which reveals the history of *Cdkn1c* transcription and translation confirms that Cdkn1c is expressed at low level in a subset of progenitors in the chick spinal neural tube. In addition, the restricted overlap of Cdkn1c-Myc detection with Rb phosphorylation suggests that in progenitors, Cdkn1c is degraded or less transcribed close to and/or after G1 completion. Indeed, we detect an overall lower expression in S/G2/M compared to G1 phase in neurogenic progenitors in the scRNAseq analysis (Fig. EV3). Overall, this may explain why a classical immunohistochemistry approach with an anti-Cdkn1c

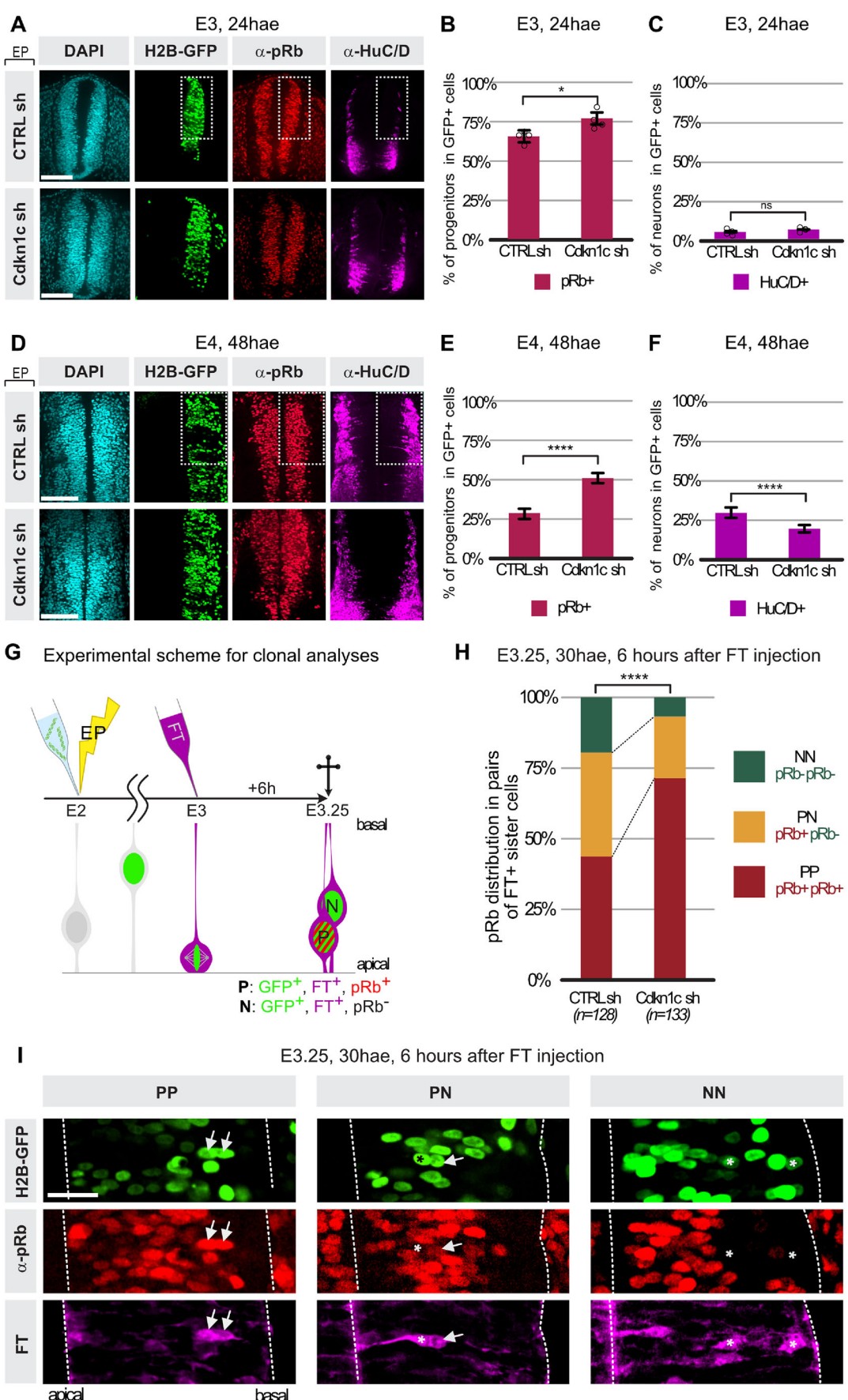

◀ **Figure 3. Downregulation of *Cdkn1c* delays the neurogenic transition in the spinal cord by favoring a proliferative symmetric mode of division.**

(A) Transverse vibratome sections of the chick neural tube (thoracic level) at E3 stained with HuC/D antibody (magenta) to label neurons and pRb (red) antibody to label progenitors in Cdkn1c (Cdkn1c sh) or control (CTRL sh) shRNA conditions. The H2B-GFP reporter (green) is expressed from the shRNA vector and labels electroporated cells in both conditions. Quantifications shown in (B, C) were performed in the dorsal 2/3rd of the neural tube, highlighted by a dotted white rectangle. Scale bar: 50 μm. (B) Distribution of pRb-positive progenitors (red) in Cdkn1c (Cdkn1c sh) or control (CTRL sh) shRNA at E3 24 h after electroporation (hae). Error bars show means ± SD. *P = 0.0144 (unpaired Student t test). N = 4 embryos per condition; electroporated cells counted: 1286 and 1626 cells for CTRL sh and Cdkn1c sh, respectively. (C) Distribution of HuC/D-positive neurons (magenta) in Cdkn1c (Cdkn1c sh) or control (CTRL sh) shRNA at E3 24 h after electroporation (hae). Error bars show means ± SD; ns: P = 0.19; (unpaired Student t test). N = 4 embryos per condition; electroporated cells counted: 1286 and 1626 cells for CTRL sh and Cdkn1c sh, respectively. (D) Transverse vibratome sections of the chick neural tube (thoracic level) at E4 stained with HuC/D antibody (magenta) to label neurons and pRb (red) antibody to label progenitors in Cdkn1c (Cdkn1c sh) or control (CTRL sh) shRNA conditions. H2B-GFP reporter (green) is expressed from the shRNA vector and labels electroporated cells in both conditions. Quantifications shown in (E, F) were performed in the dorsal 2/3rd of the neural tube, highlighted by a dotted white rectangle. Scale bar: 100 μm. (E) Distribution of pRb-positive progenitors (red) in Cdkn1c (Cdkn1c sh) or control (CTRL sh) shRNA at E4 48 h after electroporation (hae). Error bars show means ± SD ****P < 0.0001 (unpaired Student t test). CTRL sh: 3993 electroporated cells from 8 embryos, Cdkn1c sh: 7159 electroporated cells from 10 embryos. (F) Distribution of HuC/D-positive neurons (magenta) in Cdkn1c (Cdkn1c sh) or control (CTRL sh) shRNA at E4 48 h after electroporation (hae). Error bars show means ± SD ****P < 0.0001 (unpaired Student t test). CTRL sh: 3993 electroporated cells from 8 embryos, Cdkn1c sh: 7159 electroporated cells from 10 embryos. (G) Principle of the analysis of pairs of sister cells. Embryos are electroporated at HH13-14 (top left, yellow thunder) with Cdkn1c or control shRNA plasmids co-expressing a H2B-GFP reporter. Embryos are injected with the FlashTag dye (FT) 24 h after electroporation to label a synchronous cohort of mitotic progenitors, and collected 6 h later. At this time point after FT injection, anti-pRb immunofluorescence on thoracic vibratome sections determines the progenitor (pRb-positive) or neuron (pRb-negative) status of electroporated (GFP-positive) pairs of FlashTag-positive sister cells. (H) Diagram indicating the percentage of PP, PN, and NN clones for control and shRNA electroporated embryos. PP, PN, and NN stand for divisions producing two progenitors, one progenitor and one neuron, or two neurons, respectively. The distribution of PP, PN, and NN clones between control and Cdkn1c shRNAs was compared using a Chi-square test. Chi-square value = 21.78, ****P < 0.0001. N = 6 embryos for CTRL sh and n = 7 embryos for Cdkn1c sh conditions. The number of pairs of cells (n) analyzed for each condition is indicated at the bottom of the diagram. (I) Representative examples of two cell clones in transverse vibratome neural tube sections. From left to right panels: PP, PN, and NN pairs. Arrows show pRb-positive (red) progenitors and asterisks show pRb-negative neurons in FlashTag-positive (magenta) pairs of GFP-positive (green) sister cells. Scale bar: 25 μm. hae hours after electroporation.

antibody only detected the protein in very few progenitors in the developing mouse CNS (Gui et al, 2007; Mairet-Coello et al, 2012).

Altogether, our scRNAseq analyses, in situ hybridization, and knock-in experiments are consistent with our hypothesis of two sequential roles of *Cdkn1c* in progenitors and neurons.

## Downregulation of Cdkn1c in neural progenitors delays the transition from proliferative to neurogenic modes of division

To functionally investigate the role of Cdkn1c as a potential player in the transition between division modes, we reduced its expression in progenitors. We used a knock-down strategy based on the short hairpin RNA interference (shRNA) approach, with the aim to abolish its low-level expression in neurogenic progenitors, while only partially affecting the higher level of expression required in newborn neurons to trigger cell cycle exit. Of the six shRNAs that were tested against *Cdkn1c* (see "Methods"), only two (shRNA1 and shRNA4) induced an observable reduction of *Cdkn1c* mRNA expression on transverse sections, while the effect of the other four was not clearly visible (Fig. EV4A). We investigated the effect of shRNA1 at a cellular resolution by combining it with the knock-in approach described above. Quantification of Cdkn1c-Myc signal in comparison to a control shRNA confirmed a total silencing in progenitors and reduced but persistent expression in nascent neurons (Fig. EV4B,C). Of note, this approach further validates the specificity of the Myc signal in our knock-in approach.

We next explored the effect of downregulating *Cdkn1c* in progenitors on the production of neurons at the tissue level. In order to target and investigate specifically the neurogenic transition, we concentrated our analyses on the dorsal two-thirds of the neural tube, where this transition has barely started at the time of electroporation of the shRNA vectors (E2, HH13-14). Neuron and progenitor populations were evaluated 24 or 48 h after

electroporation (hae) via immunofluorescence (Fig. 3A–F, see "Methods" for the choice of the markers of these populations).

shRNA1 and shRNA4 led to a significant increase in the number of pRb-positive progenitors 48 hae (Figs. 3E and EV4D). Accordingly, the neuron population, identified with HuC/D staining, was decreased 48 h after *Cdkn1c* knockdown by shRNA1 (Figs. 3F and EV4E). This switch towards proliferation was already apparent 24 hae, as illustrated by a modest but significant increase in the pRb-positive progenitor population in shRNA1 condition (Fig. 3B), although neural differentiation, as assessed by HuC/D staining, was not yet affected at that stage (Fig. 3C). The reduction in neuronal production was not due to cell death, as we observed no excess in the number of Caspase3 positive cells in the knock-down condition (Fig. EV4F). The following experiments were carried out using shRNA1 (thereafter called Cdkn1c shRNA), which showed the most efficient downregulation in *Cdkn1c* expression and the most significant dysregulation in the ratio of progenitor versus neuron populations in our functional studies.

In the context of a full knock-out of *Cdkn1c* in the mouse spinal cord, a reduction in neurogenesis was also observed, which was attributed to a failure of prospective neurons to exit the cell cycle, resulting in the observation of ectopic mitoses in the mantle zone (Gui et al, 2007). In contrast with this phenotype, using an anti-phospho-Histone3 antibody, we did not observe any ectopic mitoses 24 or 48 h after electroporation in our knock-down condition (Fig. EV4G,H). This is consistent with the fact that we also do not observe ectopic cycling cells with pRb (Fig. 3A,D) and Sox2 (Fig. EV4G,H) antibodies. We therefore postulated that the reduced neurogenesis that we observe upon a partial *Cdkn1c* knockdown may result from a delayed transition of progenitors from the proliferative to neurogenic modes of division. To more directly address this hypothesis, we developed a clonal analysis strategy to analyze the fate of pairs of sister cells born from the division of mother cells downregulated for *Cdkn1c* (Fig. 3G). This

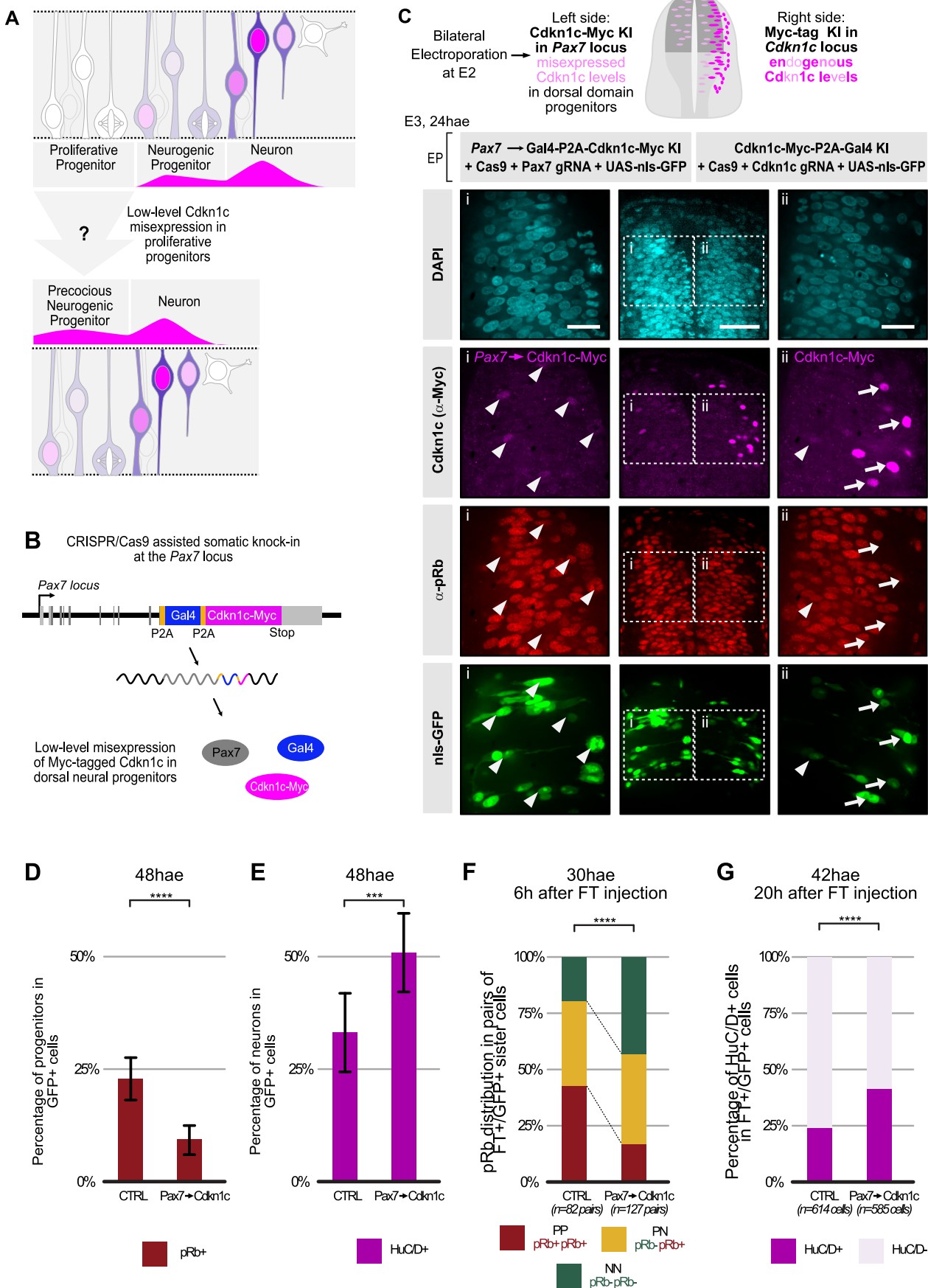

◀ **Figure 4.  Premature expression of Cdkn1c at low levels in proliferative progenitors converts them to a neurogenic mode of division.**

(A) Schematic representation of the misexpression strategy used to investigate whether low-level Cdkn1c expression in progenitors induces neurogenic divisions. (B) Schematic representation of the modified *Pax7* locus driving low-level expression of Cdkn1c-Myc and Gal4-VP16 in dorsal progenitors. *Pax7* (gray), *Gal4-VP16* (blue), and *Cdkn1c -Myc* (magenta) coding sequences are transcribed from the *Pax7* locus and co-translated. The insertion of P2A pseudocleavage sites (orange) between the three sequences ensures that all three proteins are present as independent proteins (see Fig. EV7B,C and "Methods" for further details). (C) *Pax7*-driven misexpression of Cdkn1c in dorsal progenitors mimics the levels of endogenous Cdkn1c expression in neurogenic progenitors. Bilateral electroporation was performed at a 3 h interval to compare *Pax7*-driven levels of Cdkn1c expression (electroporation 1, right side hemi-tube, knock-in of Cdkn1c-Myc in the *Pax7* locus) with endogenous Cdkn1c levels (electroporation 2, left side hemi-tube, knock-in of a Myc tag in the *Cdkn1c* locus). The Cdkn1c -Myc (magenta) signal driven by *Pax7* is mostly restricted to the ventricular region, and its intensity is comparable to the endogenous levels of Cdkn1c-Myc in the contralateral side (i and ii, arrowheads) and never reaches the high levels of endogenous Cdkn1c-Myc signal observed in the intermediate zone (i, arrows). Knock-in cells are identified via the expression of a co-electroporated UAS-nls-GFP reporter (green). Images are single z-level confocal images. The insets shown in Columns 1 and 3 are ×100 confocal images acquired in the same section and presented with the same display parameters. Columns 1 and 3: ×100 objective, column 2: ×20 objective. Scale bars: 50 µm in ×20 images, 20 µm in ×100 close-ups. KI knock-in. (D) Distribution of pRb-positive progenitors in control versus *Pax7*-driven Cdkn1c misexpression 48 hae. The control condition consists of a Gal4-VP16 knock-in at the *Pax7* locus. For both conditions, knock-in cells were identified via the expression of a co-electroporated UAS-nls-GFP reporter. Error bars show means ± SD ****$P < 0.0001$, unpaired Student $t$ test. CTRL (Pax7->Gal4): 919 GFP-positive cells from 8 embryos, Pax7->Cdkn1c: 1261 GFP-positive cells from 9 embryos. (E) Distribution of HuC/D-positive neurons in control versus *Pax7*-driven Cdkn1c misexpression 48 hae. The control condition consists of a Gal4-VP16 knock-in at the *Pax7* locus. For both conditions, knock-in cells were identified via the expression of a co-electroporated UAS-nls-GFP reporter. Error bars show means ± SD ***$P = 0.0006$, unpaired Student $t$ test. CTRL (Pax7->Gal4): 919 GFP-positive cells from 8 embryos, Pax7-> Cdkn1c: 1261 GFP-positive cells from 9 embryos. (F) Distribution of PP, PN, and NN pairs of sister cells in control versus *Pax7*-driven Cdkn1c misexpression. For the control condition, we used a knock-in construct where Gal4-VP16 alone is inserted downstream of *Pax7*. For both conditions, knock-in events in dorsal progenitors were revealed thanks to a co-electroporated UAS-nls-GFP reporter. PP, PN, and NN stand for divisions producing two progenitors, one progenitor and one neuron, or two neurons, respectively. The analysis was performed 30 h after electroporation (hae) and 6 h after FT injection in the dorsal half of the neural tube. The distribution of PP, PN, and NN clones between both conditions was compared using a Chi-square test. Chi-square value = 3, ****$P < 0.0001$. Five and seven embryos were used for CTRL (Pax7->Gal4) and Pax7->Cdkn1c conditions, respectively. The number of pairs analyzed is indicated at the bottom of the diagram. (G) Distribution of HuC/D-positive neurons in a cohort of cells 20 h after FlashTag labeling in control versus *Pax7*-driven Cdkn1c misexpression. For the control condition, we used a knock-in construct where Gal4-VP16 alone is inserted downstream of *Pax7*. The analysis was performed 42 h after electroporation (hae) in the dorsal half of the neural tube. The injection of Flash Tag was performed 22 h post electroporation. For both conditions, knock-in events in dorsal progenitors were revealed thanks to a co-electroporated UAS-nls-GFP reporter. The distribution of HuC/D-positive neurons between both conditions was compared using a Chi-square test. Chi-square value = 41, ****$P < 0.0001$; CTRL (Pax7->Gal4): 614 GFP-FlashTag-positive cells from 8 embryos, Pax7->Cdkn1c: 585 GFP-FlashTag-positive cells from 7 embryos. hae hours after electroporation.

approach allows for the retrospective deducing of the mode of division used by the mother progenitor cell. We injected the cell-permeant dye "FlashTag" (FT) at E3 to specifically label a cohort of progenitors that undergoes mitosis synchronously at the time of injection (Baek et al, 2018; Telley et al, 2016 and see "Methods"), and analyzed the fate of their progeny 6 h later (Figs. 3G–I and EV5). Our characterization of pRb immunoreactivity in the tissue had established beforehand that 6 h after mitosis, all progenitors can reliably be detected with this marker (Fig. EV2, "Methods"). Therefore, at this timepoint after FT injection, two-cell clones selected on the basis of FT incorporation and similar GFP signal intensity can be categorized as PP, PN, or NN based on pRb positivity (P) or not (N) (see "Methods", Figs. 3I, EV2, and EV5). Strikingly, upon *Cdkn1c* knockdown, we observed a massive increase in the number of PP clones at the expense of PN and NN clones (Fig. 3H). These knock-down experiments support the hypothesis that the onset of a low-level Cdkn1c expression favors the transition to neurogenic progenitors. In agreement with this conclusion, analysis of sister cells born from Cdkn1c-positive progenitors (identified by knock-in of the Gal4 driver in the *Cdkn1c* locus) showed a significantly higher proportion of neurogenic pairs (PN and NN) compared to the total progenitor population (Fig. EV6).

## Inducing a premature expression of Cdkn1c in progenitors triggers the transition to neurogenic modes of division

We next explored whether low Cdkn1c activity is sufficient to induce the transition to neurogenic modes of division. A previous study has shown that a massive overexpression of Cdkn1c driven by

the strong CAGGS promoter triggers cell cycle exit of chick spinal cord progenitors, revealed by a drastic loss of BrdU incorporation 1 day after electroporation (Gui et al, 2007). As this precludes the exploration of our hypothesis, we developed an alternative approach designed to prematurely induce a pulse of Cdkn1c in progenitors, with the aim to emulate in proliferative progenitors the modest level of expression observed in neurogenic progenitors (Fig. 4A). We took advantage of the *Pax7* locus, which is expressed in progenitors in the dorsal domain (Jostes et al, 1990) at a level similar to that observed for *Cdkn1c* in neurogenic precursors (Fig. EV7A).

We used the CRISPR/Cas9-based somatic approach to introduce a sequence including a Myc-tagged *Cdkn1c* coding sequence and the *Gal4-VP16* transcription factor downstream of *Pax7* (Figs. 4B and EV7B–D). As a control for these experiments, we used a construct that introduces the Gal4-VP16 reporter alone to the *Pax7* locus (Petit-Vargas et al, 2024), allowing us to target and analyze the same cell population in control and overexpression conditions. We first confirmed that the expression of the Cdkn1c-Myc and of a UAS-nls-GFP reporter resulting from the *Cdkn1c-Myc-P2A-Gal4* knock-in at the *Pax7* locus was restricted to progenitors in the dorsal domain (Fig. EV7E), with minimal perturbation of the Pax7 protein expression (Fig. EV7F,G). Importantly, Myc signal intensity was similar to the one observed for endogenous Myc-tagged *Cdkn1c* in progenitors, and remained below the endogenous level of Myc-tagged *Cdkn1c* observed in nascent neurons, confirming the validity of our strategy (Figs. 4C and EV7H,I).

We therefore proceeded to analyze the consequences of Cdkn1c premature expression in progenitors. At the population level, at E4 (48 h after electroporation), this resulted in a strong reduction in the number of progenitors (pRb-positive cells) and a significant

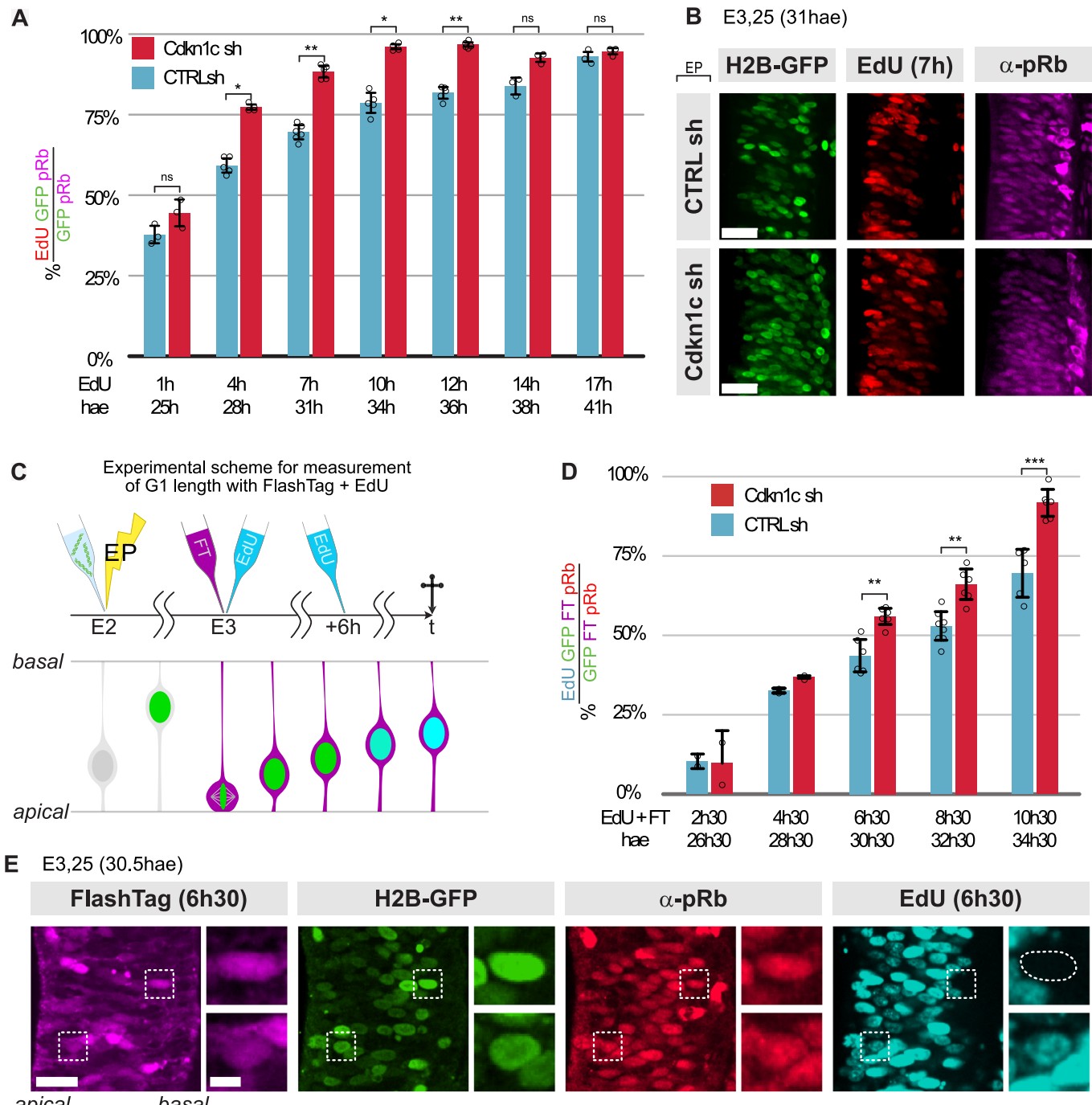

increase in the proportion of neurons (HuC/D-positive) born from the Pax7-positive knock-in population (revealed with a UAS-nls-GFP reporter) (Figs. 4D,E and EV8A).

We next examined whether the increased neurogenesis 48 hae is linked to a change in the mode of division of progenitors misexpressing Cdkn1c. Using the FlashTag cohort labeling approach described above, we traced the fate of daughter cells born 24 hae. We observed an increase in the proportion of terminal neurogenic (NN) divisions and a decrease in proliferative (PP) divisions (Fig. 4F). As above, the P and N identities in this analysis

are based on our model of a plateau of pRb 6 h after mitosis at that stage (Fig. EV2). In parallel, to rule out a possible misinterpretation of daughter cells' identity resulting from a delay in reaching the Rb phosphorylation plateau upon Cdkn1c overexpression, we monitored HuC/D expression in a synchronous cohort of newborn cells 20 h after FlashTag injection (Figs. 4G and EV8B). The increase in HuC/D-positive neurons observed at this time point confirms an increase in neurogenic divisions.

Overall, this suggests that Cdkn1c premature expression in PP progenitors converts them to the PN mode of division, while the

**Figure 5.   Cdkn1c controls cell cycle and G1 phase duration in neural progenitors.**

(A) Cumulative EdU incorporation in the overall electroporated progenitor population. The columns represent the percentages of EdU/pRb/GFP triple-positive cells in the electroporated progenitor (GFP/pRb double-positive cells) population of control (blue) versus Cdkn1c shRNA (red) conditions at each time point. EdU 5-ethynyl-20-deoxyuridine. Error bars show means ± SD; 1 h: ns: P = 0.099; 4 h: *P = 0.0159; 7 h: **P = 0.0043; 10 h: *P = 0.0159; 12 h: **P = 0.0079; 14 h and 17 h: ns, P = 0.1 (Mann–Whitney test). Three to five embryos were used per condition and at each time point (see Table EV1 for the number of pRb/GFP double-positive cells analyzed). (B) Representative images of EdU (red), anti-pRb immunostaining (magenta), and H2B-GFP (green) from the EdU incorporation experiment in transverse vibratome sections at the 7 h timepoint. A close-up of the electroporated side is shown. The apical surface of the ventricular zone is on the left side. H2B-GFP is expressed from the shRNA vectors and labels electroporated cells. (C) Schematic representation of the experimental strategy to measure G1 length. Embryos were electroporated with Cdkn1c or control shRNA at E2.25. One day later, FlashTag injection in the neural tube and EdU administration were performed simultaneously. Embryos were harvested at consecutive time points every 2 h between 2 h 30 min and 10 h 30 min. For time points beyond 6 h, a second EdU injection was performed 6 h after the first one. (D) Dynamics of EdU incorporation in a FlashTag-positive cohort of electroporated progenitors. The columns represent the percentages of EdU-positive cells in FlashTag/pRb/GFP triple-positive cells in Cdkn1c (red) and control shRNA (blue) populations at each time point. Error bars show means ± SD; 6 h 30: **P = 0.0023; 8 h 30, ** = 0.0013; 10 h 30: ***P = 0.00043 (Mann–Whitney test). Three to seven embryos were used per condition and at each time point (see Table EV1 for the number of FlashTag/pRb/GFP triple-positive cells analyzed) except for the first two timepoints, where 2 embryos were analyzed for each condition. (E) Representative example of images used for the quantification of EdU-positive progenitors in (D). A region of the electroporated side in a transverse vibratome section at the 6 h 30 min timepoint is shown. Dashed white boxes and the corresponding insets show examples of triple GFP-positive/FlashTag-positive/pRB-positive cells used in the quantification. The EdU-negative cell (top) is still in G1-phase, while the EdU-positive cell (bottom) has already entered S-phase. hae hours after electroporation.

combined endogenous and *Pax7*-driven expression of Cdkn1c converts PN progenitors to the NN mode of division. Coincidentally, at the stage analyzed, PP to PN conversions are balanced by PN to NN conversions, leaving the PN proportion artificially unchanged in our quantification (Fig. 4F). The alternative interpretation of a direct conversion of symmetric PP into symmetric NN divisions is less likely, because the PN compartment was affected in the reciprocal *Cdkn1c* shRNA approach (see Fig. 3H). Altogether, these data show that inducing a premature low-level expression of Cdkn1c in cycling progenitors is sufficient to accelerate the transition toward neurogenic modes of division.

## The proneurogenic activity of Cdkn1c in progenitors is mediated by modulation of cell cycle dynamics

Previous data in the developing cortex of *Cdkn1c* knock-out mice described a transient increase in proliferation (between E14.5 and E16.5) linked to a shorter progenitor cell cycle duration mainly due to a reduction of the G1 phase length (Mairet-Coello et al, 2012). However, in addition to its role as a cell cycle regulator, Cdkn1c also performs other functions, as illustrated by its role as a transcription co-factor or as a pro- or anti-apoptotic factor in different contexts (Creff and Besson, 2020).

To assess the mechanisms of action of Cdkn1c in the neurogenic transition, we monitored cell cycle parameters in *Cdkn1c* knock-down conditions (see "Methods"). Cumulative EdU incorporation in spinal progenitors (pRb-positive) at E3 (24 hae) showed that the proportion of EdU-positive progenitors reached a plateau faster and in a sharper manner in the *Cdkn1c* shRNA population (Fig. 5A,B). This indicates that the total duration of the cell cycle is shorter upon *Cdkn1c* knockdown. To specifically assess a possible reduction in G1 length, we developed an approach that provides a direct measurement of G1 duration, contrary to the classical method of G1 inference (Nowakowski et al, 1989). Our approach uses precise landmarks to delineate G1 phase borders: mitosis exit (through the FlashTag (FT) labeling of a synchronized cohort of dividing progenitors) and S phase entry (through cumulative EdU labeling; Fig. 5C–E, see "Methods"). We found that the proportion of pRb-positive progenitors having entered S phase (EdU-positive cells) was significantly higher at all time points examined more than 4h30 after FT injection in the *Cdkn1c* knock-down condition

compared to the control population (Fig. 5D). Importantly, virtually all progenitors electroporated with *Cdkn1c* shRNA had reached S phase 10h30 after FT injection, whereas approximately one-third of control progenitors were still in G1 at that time. Taken together, these results demonstrated a shorter cell cycle in the *Cdkn1c*-downregulated condition at the population scale, which is, at least partially, related to a decrease in G1 duration.

To explore whether these changes in cell cycle parameters are responsible for the decrease in neuron production observed upon *Cdkn1c* knock-down, we targeted the CyclinD1-CDK4/6 complex, which promotes cell cycle progression and proliferation in the developing CNS (Gui et al, 2007; Lange et al, 2009; Lobjois et al, 2008, 2004), and is inhibited by Cdkn1c (Gui et al, 2007). *CyclinD1* loss of function by shRNA reduces the number of cycling cells in the chick embryonic neural tube (Lacomme et al, 2012; Lukaszewicz and Anderson, 2011) and favors neurogenesis in the mouse cortex, possibly through a lengthening of G1 phase (Lange et al, 2009). We hypothesized that the anti-neurogenic *Cdkn1c* knock-down phenotype might be caused by a failure to inhibit CyclinD1, and that a concomitant downregulation of *CyclinD1* should therefore rescue, at least partially, this phenotype. Using a validated shRNA targeting chick *CyclinD1* (Lukaszewicz and Anderson, 2011, Fig. EV9A), we compared the effect of individual and simultaneous downregulation of *CyclinD1* and *Cdkn1c*. Remarkably, at 48 hae, whereas *Cdkn1c* shRNA and *CyclinD1* shRNA alone respectively increased and decreased the proportion of EdU incorporation in pRb-positive progenitors, the double *Cdkn1c/CyclinD1* knock-down was indistinguishable from control (Fig. 6A); in addition, the concomitant downregulation of *CyclinD1* rescued the reduction in G1 length observed upon *Cdkn1c* knock-down (Fig. 6B), as assessed in a FlashTag cohort by cumulative EdU incorporation in pRb-positive cells 6 h 30 min after mitosis. These data are consistent with the hypothesis that the effect of Cdkn1c on G1 duration is mediated by CyclinD1 inhibition.

We then analyzed the fate of pairs of sister cells born from progenitors dividing 24 hae in these four conditions. While *CyclinD1* knock-down alone did not alter the distribution of PP, PN and NN pairs compared to the control situation, in double knock-down experiments it completely counteracted the anti-neurogenic effect of *Cdkn1c* downregulation, restoring a distribution of PP, PN and NN pairs similar to the control condition

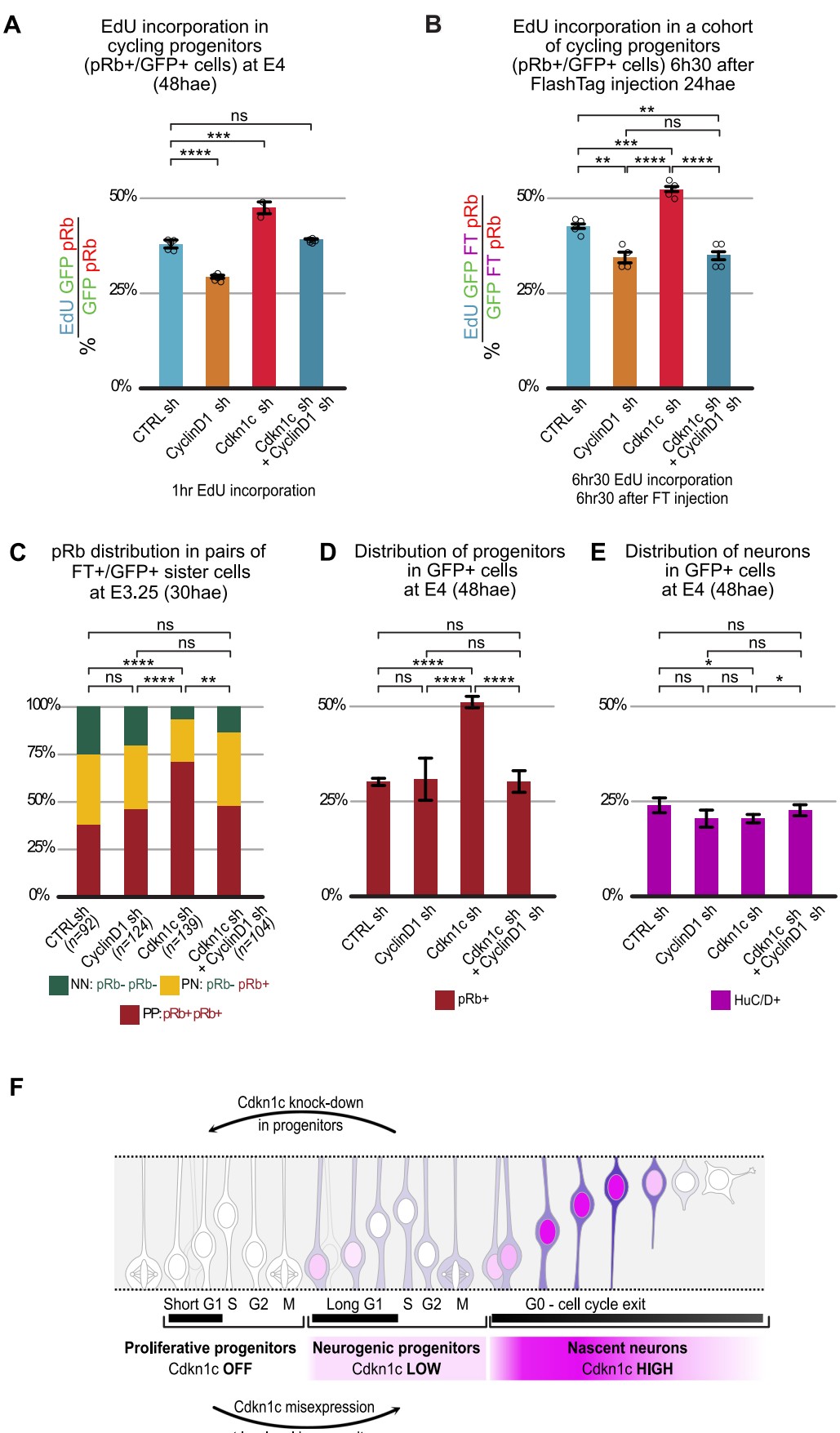

**A** EdU incorporation in cycling progenitors (pRb+/GFP+ cells) at E4 (48hae)

**B** EdU incorporation in a cohort of cycling progenitors (pRb+/GFP+ cells) 6h30 after FlashTag injection 24hae

1hr EdU incorporation

6hr30 EdU incorporation
6hr30 after FT injection

**C** pRb distribution in pairs of FT+/GFP+ sister cells at E3.25 (30hae)

NN: pRb- pRb-   PN: pRb- pRb+
PP: pRb+ pRb+

**D** Distribution of progenitors in GFP+ cells at E4 (48hae)

pRb+

**E** Distribution of neurons in GFP+ cells at E4 (48hae)

HuC/D+

**F**

Cdkn1c knock-down in progenitors

Short G1 S G2 M    Long G1 S G2 M    G0 - cell cycle exit

Proliferative progenitors   Neurogenic progenitors   Nascent neurons
Cdkn1c **OFF**              Cdkn1c **LOW**            Cdkn1c **HIGH**

Cdkn1c misexpression at low level in progenitors

◄ **Figure 6.  Cdkn1c acts on cell cycle duration and neurogenesis via regulation of CyclinD1 activity.**

(A) Diagram indicating the proportion of EdU-positive progenitors after a 1 h EdU pulse at E4. The columns represent the percentages of EdU-positive cells in electroporated progenitors (pRb/GFP double-positive) in control, single CyclinD1, single Cdkn1c, and double CyclinD1/Cdkn1c shRNA conditions. Error bars show means ± SD; ns, CTRL vs Cyclin D1: ****$P < 0.0001$; CTRL vs Cdkn1c: ***$P = 0.0002$; CTRL vs double ns, $P = 0.0864$; (unpaired Student $t$ test). Three to six embryos were used per condition. See Table EV1 for the number of pRb/GFP double-positive cells analyzed. (B) Dynamics of EdU incorporation in a FlashTag-positive cohort of electroporated progenitors 6 h 30 min after FlashTag injection. The columns represent the percentages of EdU-positive cells in FlashTag/pRb/GFP triple-positive cells in control, single CyclinD1, single Cdkn1c, and double CyclinD1/Cdkn1c shRNAs conditions 6 h 30 min after FlashTag injection. Error bars show means ± SD; CTRL vs Cyclin D1: **$P = 0.0038$; CTRL vs Cdkn1c: ***$P = 0.0006$; CTRL vs double **$P = 0.0091$; CyclinD1 vs Cdkn1c: ****$P = 0.0001$; CyclinD1 vs double: ns, $P = 0.8090$; Cdkn1c vs double: ****$P = 0.0001$; (unpaired Student $t$ test). Four embryos were used per condition (see Table EV1 for the number of FlashTag/pRb/GFP triple-positive cells analyzed). (C) Diagram indicating the percentage of PP, PN, and NN pairs of sister cells in control, single CyclinD1, single Cdkn1c and double CyclinD1/Cdkn1c shRNA conditions. PP, PN, and NN stand for divisions producing two progenitors, one progenitor and one neuron, or two neurons, respectively. The distribution of PP, PN, and NN clones between different conditions was compared using the Chi-square test. CTRL vs Cyclin D1: ns, $P = 0.47$; CTRL vs Cdkn1c: Chi-square value = 28.45, ****$P < 0.0001$; CTRL vs double ns, $P = 0.1$; CyclinD1 vs Cdkn1c: Chi-square value = 19.70, ****$P < 0.0001$; CyclinD1 vs double: ns, $P = 0.39$; Cdkn1c vs double: Chi-square value = 13.58, **$P = 0.0011$. Eight to ten embryos were used per condition. The number of pairs analyzed is indicated at the bottom of the diagram. (D) Distribution of pRb-positive progenitors at E4 in control, single CyclinD1, single Cdkn1c, and double CyclinD1/Cdkn1c shRNA conditions. Error bars show means ± SD; CTRL vs Cyclin D1: ns, $P = 0.82$; CTRL vs Cdkn1c: ****$P < 0.0001$; CTRL vs double ns, $P = 0.9839$; CyclinD1 vs Cdkn1c: ****$P < 0.0001$; CyclinD1 vs double: ns, $P = 0.9839$; Cdkn1c vs double: ****$P < 0.0001$; (unpaired Student $t$ test). Quantification from transverse vibratome sections at the thoracic level at E4 (48 hae). See Fig. EV9B for representative images. Between 2580 and 4373 electroporated cells from 5 to 6 embryos were counted for each condition. (E) Distribution of the neurons (HuC/D-positive cells) at E4 in control, single CyclinD1, single Cdkn1c, and double CyclinD1/Cdkn1c shRNAs conditions. Error bars show means ± SD; CTRL vs Cyclin D1: ns, $P = 0.05$; CTRL vs Cdkn1c: *$P = 0.014$; CTRL vs double: ns, $P = 0.36$; CyclinD1 vs Cdkn1c: ns, $P = 0.96$; CyclinD1 vs double: ns, $P = 0.118$; Cdkn1c vs double: *$P = 0.0401$; (unpaired Student $t$ test). Quantification from transverse sections at the thoracic level at E4 (48 hae). See Fig. EV9B for representative images. Between 2580 and 4373 electroporated cells from 5 to 6 embryos were counted for each condition. (F) Cartoon summarizing the dual role of Cdkn1c in the neurogenic transition of progenitors and cell cycle exit of newborn neurons via the dynamics of its expression level. hae hours after electroporation.

(Fig. 6C). Consistently, at the population level, CyclinD1 down-regulation alone did not affect the ratio of proliferating progenitors and neuron production 48 hae, but it fully rescued the *Cdkn1c* knockdown phenotype and restored the rate of neurogenesis to that of a control situation (Figs. 6D,E and EV9B).

These results demonstrate that the neurogenic activity of Cdkn1c in progenitors is largely resulting from its regulatory role on cell cycle dynamics.

## Discussion

In this report, we used pseudotime reconstitution of single-cell transcriptomics to identify regulators of the neurogenic transition in the chick spinal cord. After defining a set of genes clustering with the neurogenic progenitor marker *Tis21*, we established a "neurogenic progenitor score" that was used for differential expression analyses. We focused on one of the most significant candidates emerging from this analysis, the Cyclin/CDK inhibitor Cdkn1c. We provide evidence that *Cdkn1c* expression initiates at low levels in neurogenic progenitors before it peaks at higher levels in future neurons. Importantly, specifically altering the early phase of *Cdkn1c* expression impairs the balance between proliferative and neurogenic modes of division. This allows us to re-interpret the role of Cdkn1c during spinal neurogenesis: while previously mostly considered as a binary regulator of cell cycle exit, we demonstrate that Cdkn1c is also an intrinsic regulator of the transition from the proliferative to neurogenic status in cycling progenitors. This occurs through a change in their mode of division, and our double knock-down experiments suggest that the onset of Cdkn1c expression may promote this change by counteracting a CyclinD1/CDK6 complex-dependent mechanism.

Cell cycle regulators are key players of neurogenesis, controlling the ability of neural cells to either proliferate or exit the cell cycle and differentiate. Nonetheless, studies in a wide range of species have demonstrated that beyond this binary choice, cell cycle regulators also influence the neurogenic potential of progenitors, i.e the commitment of their progeny to differentiate or not (Bonnet et al, 2018; Calegari and Huttner, 2003; Fujita, 1962; Kicheva et al, 2014; Lange et al, 2009; Lukaszewicz and Anderson, 2011; Peco et al, 2012; Pilaz et al, 2009; Roussat et al, 2023; Smith and Schoenwolf, 1987; Takahashi et al, 1995). Among cell cycle parameters, the major change observed during the neurogenic transition is a lengthening of the G1 phase. In the developing mouse cortex, progenitors positive for the neurogenic marker Tis21 display a longer G1 phase compared to Tis21 negative progenitors (Calegari et al, 2005; Calegari and Huttner, 2003; Lange et al, 2009) and apical radial glia display a shorter G1 phase than fate restricted intermediate progenitors (Arai et al, 2011). Consistently, live imaging has documented a lengthening of the G1 phase between two consecutive cycles in chick spinal neural progenitors (Molina et al, 2022). Experimental modulations of the duration of G1 phase in the dividing mother cell have revealed its pivotal role in determining the daughter cells' capacity to self-renew or differentiate (Calegari and Huttner, 2003; Lange et al, 2009; Pilaz et al, 2009). These functional explorations have mostly focused on Cyclin/CDK complexes (Calegari et al, 2005; Calegari and Huttner, 2003; Lange et al, 2009; Pilaz et al, 2009). Intriguingly, while Cdkn1c is a key regulator of these complexes, its gain and loss of function phenotypes in the CNS have essentially been attributed to a post mitotic role in daughter cells, primarily through its function in regulating cell cycle exit (Gui et al, 2007; Mairet-Coello et al, 2012; Tury et al, 2011).

Our study combines a detailed analysis of *Cdkn1c* expression dynamics in progenitors and nascent neurons with functional approaches specifically targeting its activity in progenitors to resolve this apparent paradox. Using an innovative somatic knock-in strategy, we were able to tag the *Cdkn1c* locus with Myc epitopes and access the history of Cdkn1c transcription and translation via the Gal4-UAS reporter, and we demonstrated that the protein is expressed at low levels in neurogenic progenitors. Functional experiments dedicated to challenge specifically Cdkn1c expression

in progenitors showed that reducing its expression favors a proliferative mode of division and eventually impedes neurogenesis, whereas a premature induction of low-level Cdkn1c expression is sufficient to convert proliferative into neurogenic progenitors (Fig. 6F). In addition, we showed that Cdkn1c regulates the switch to a neurogenic division mode, possibly through an elongation of the G1 phase of the cell cycle resulting from the inhibition of the CyclinD-CDK6 complex (CDK4 is absent from the chicken genome).

Taken together, our study adds a new player to the complex panorama linking cell cycle dynamics and progenitor cell behavior. It reinforces the view that a faster or slower passage through the G1 phase of a progenitor is instructive for its proliferative versus neurogenic mitotic behavior, and therefore dictates the fate of its daughters. It also highlights the importance of tightly regulating the dynamics of expression of cell cycle regulators to ensure a proper neurogenic progression (Fig. 6F).

In the mouse spinal cord, the higher proliferation observed in *Cdkn1c* knock-out embryos was attributed to a failure to exit the cell cycle in newborn neurons (Gui et al, 2007). In light of our results in the chick spinal cord, the mouse phenotype could also in part be interpreted as a failure of proliferative progenitors to progress to the neurogenic state, although this study did not report the "low level" expression of Cdkn1c that we describe here in the neurogenic progenitor population, possibly due to a low sensitivity of the antibody. Supporting this idea, higher proliferation and a shortening of the G1 phase have been described in cortical progenitors in *Cdkn1c* knock-out mice (Mairet-Coello et al, 2012). Altogether, this suggests a possible conservation of the dual role linked to the dynamics of expression of Cdkn1c in the embryonic mammalian CNS.

What initiates *Cdkn1c* expression in progenitors? A first possibility is that *Cdkn1c* expression may be controlled by a regulatory cascade involving the *Hes* genes. In pancreatic progenitors, *Cdkn1c* is transcriptionally repressed by Hes1 downstream of the Notch pathway (Georgia et al, 2006), and the same authors report a complementary expression of *Hes1* and *Cdkn1c* in the mouse neural tube. Our scRNASeq data analysis confirms a similar complementarity between *Hes5* (which appears to be functionally equivalent to Hes1 in the chick spinal tube; Fior and Henrique, 2005) and Cdkn1c. The transcriptional activity of *Hes1* in pancreatic progenitors is repressed by Hes6. Interestingly, the onset of *Hes6* expression shortly precedes that of Cdkn1c in our transcriptomic scRNASeq dataset. This suggests that the initiation of Cdkn1c expression in neurogenic progenitors might be triggered by the antagonistic activity of Hes6 on *Hes1/Hes5*. Another candidate is the proneural bHLH transcription factor NeuroG2. Gui et al proposed that the strong expression of Cdkn1c that drives prospective neurons out of the cell cycle is under transcriptional control of the NeuroG2 transcription factor in the spinal cord. Indeed, a massive overexpression of NeuroG2 leads to a strong Cdkn1c upregulation and cell cycle exit (Gui et al, 2007). Our scRNAseq data indicate that NeuroG2 is already expressed at low levels in neurogenic progenitors (Fig. 1D) and could therefore act upstream of Cdkn1c onset in progenitors.

Once its expression is initiated, Cdkn1c may take part in a feedback loop with NeuroG2 leading to their concomitant upregulation during neurogenesis, such as what has been described

in other contexts. Indeed, studies in Xenopus primary neurogenesis and in mammalian pancreatic progenitors have shown that Cdkn1c-related CDK inhibitors favor the stability of NeuroG2 and NeuroG3 proteins. Mechanistically, this involves the inhibition of CDK/cyclin-dependent phosphorylation of NeuroG2/3, which targets them for degradation (Ali et al, 2014; Azzarelli et al, 2017; Krentz et al, 2017; Vernon et al, 2003). Additionally, in the mouse developing cortex, the closely related p27[Kip1]/Cdkn1b was shown to regulate NeuroG2 protein stability and activity in a cell cycle and CDK/Cyclin-independent manner (Nguyen et al, 2006). Possibly reinforcing the feedback loop, NeuroG2 is also involved in the downregulation of CyclinD1 and CyclinE2 in spinal progenitors (Lacomme et al, 2012). Whether such a feedback loop exists between Cdkn1c and NeuroG2 remains to be tested directly in neural progenitors, but it is tempting to speculate that it would drive the transition from their moderate expression, driving progenitors into a neurogenic state, to a peak of expression leading to cell cycle exit and differentiation in prospective neurons. This is supported by previous work in the mammalian pancreas, showing that a mild overexpression of the related Cdkn1b, which lengthens G1 without overt cell cycle exit, also results in Neurog3 stabilization and priming for endocrine differentiation (Krentz et al, 2017). Further work is needed to decipher the mutual relationships between these factors during the neurogenic transition.

Neurogenic regulators display dynamic expression during the process of neurogenesis, and different levels in their expression may correspond to specific activities during the progression through different cellular states, as illustrated in this study with Cdkn1c. These multiple functions cannot be disentangled by complete loss-of-function or massive overexpression. Here, we circumvented these limitations via a simple and efficient somatic knock-in method that allowed us to tightly control the level and duration of Cdkn1c misexpression in a subset of progenitors in the chick embryo. The knowledge of expression dynamics of large panels of genes from single-cell transcriptomics combined with our flexible method of targeted genome insertions opens the way to generalize similar customized approaches to functionally explore regulatory networks during neurogenesis. This strategy could also be extended to other contexts and animal models, enabling to bypass the lengthy process of transgenic line generation and complex crossing schemes.

One key aspect of the neurogenic transition is the reliance on asymmetric division in the early stages of neuron production. Asymmetric division of neural progenitors is an active process relying on intrinsic asymmetries in the progenitor cell involving the unequal distribution of fate determinants during mitosis (e.g. (Derivery et al, 2015; Kressmann et al, 2015; Saade et al, 2017; Tozer et al, 2017). It will be important to understand whether and how Cdkn1c and other regulators of the cell cycle are involved in setting up the intrinsic polarities necessary for this process in neural progenitors. In this context, it is interesting to note that several cell cycle regulators have already been shown to play a direct role in the machinery that establishes cellular asymmetry during the division of drosophila neuroblasts (Tio et al, 2001) and sensory organ precursors (Darnat et al, 2022); on the other hand, studies in the developing mouse cortex have shown that the *CyclinD2* mRNA is asymmetrically localized and inherited upon division of neural progenitors, and behaves as a fate determinant in

their progeny (Tsunekawa et al, 2012). Hence, cell cycle regulators are likely to be involved at multiple levels in the process of neurogenesis, from the determination of the neurogenic competence of neural progenitors to the cellular process of asymmetric division. In this context, it will be interesting to explore whether and how Cdkn1c might control the asymmetric distribution of fate determinants that have been identified over the last years (Peyre and Morin, 2012; Saade et al, 2017; Tozer et al, 2017; Tsunekawa et al, 2012).

# Methods

### Reagents and tools table

| Reagent/resource | Reference or source | Identifier or catalog number |
|---|---|---|
| **Antibodies** | | |
| Chick anti-GFP | Aves Labs | Cat#GFP-1020 RRID : AB_10000240 |
| Rabbit anti-pRb (Ser807/811) | Cell Signaling | Cat# 8516S RRID : AB_331472 |
| Mouse anti-HuC/D (clone 16A11) | ThermoFisher Scientific | Cat#A-21271 RRID : AB_221448 |
| Rabbit anti-c-Myc tag | Sigma-Aldrich | Cat# C3956 RRID : AB_439680 |
| Mouse anti-c-myc tag (clone 9E10) | Sigma-Aldrich | Cat# M5546 RRID : AB_260581 |
| Mouse anti-Pax7 | DSHB | Cat# pax7 RRID : AB_528428 |
| Rabbit anti-pH3 | Millipore | Cat# 06570 RRID : AB_310177 |
| Rabbit anti-active Caspase3 | R&D Systems | Cat#, AF835 RRID : AB_2243952 |
| Rabbit anti-RFP | Rockland | Cat# 600-401-379, RRID : AB_2209751 |
| Goat anti-Sox2 | R&D Systems | Cat#, AF2018 RRID : AB_355110 |
| **Experimental models: organisms/strains** | | |
| Chick fertilized eggs | EARL Morizeau | JA57 |
| **Recombinant DNA** | | |
| pTol2-H2B-eGFP | Peyre et al, 2011 | N/A |
| pTol2-H2B-eGFP Luciferase | Peyre et al, 2011 | N/A |
| pTol2-H2B-eGFP-Cdkn1c shRNA1 (Cdkn1c sh1) | This paper | N/A |
| pTol2-H2B-eGFP-Cdkn1c shRNA2 (Cdkn1c sh2) | This paper | N/A |
| pTol2-H2B-eGFP-Cdkn1c shRNA3 (Cdkn1c sh3) | This paper | N/A |
| pTol2-H2B-eGFP-Cdkn1c shRNA4 (Cdkn1c sh4) | This paper | N/A |
| pTol2-H2B-eGFP-Cdkn1c shRNA5 (Cdkn1c sh5) | This paper | N/A |
| pTol2-H2B-eGFP-Cdkn1c shRNA6 (Cdkn1c sh6) | This paper | N/A |
| pX330-U6-Chimeric_BB-CBh-hSpCas9 | Cong et al, 2013 | RRID: Addgene_42230 |
| Control-gRNA (X-905) | Petit-Vargas et al, 2024 | N/A |
| Uni2-dual-gRNA cloning vector (X-1251) | Petit-Vargas et al, 2024 | N/A |

| Reagent/resource | Reference or source | Identifier or catalog number |
|---|---|---|
| Uni2-control-dual-gRNA (X-1252) | Petit-Vargas et al, 2024 | N/A |
| Uni2-Cdkn1c-dual-gRNA#1 (X-1254) | This paper | N/A |
| Uni2-Cdkn1c-dual-gRNA#2 (X-1255) | This paper | N/A |
| Uni2-Cdkn1c-dual-gRNA#3 (X-1256) | This paper | N/A |
| Pax7-gRNA#2 (X-824) | Petit-Vargas et al, 2024 | N/A |
| Cdkn1c-6Myc-P2A-Gal4 targeting construct (X-1270) | This paper | N/A |
| Pax7-> P2A-Gal4 targeting construct (X-906) | Petit-Vargas et al, 2024 | N/A |
| Pax7-> P2A-Cdkn1c-3Myc-P2A-Gal4 targeting construct (X-1246) | This paper | N/A |
| Pax7-> P2A-Cdkn1c-6Myc-P2A-Gal4 targeting construct (X-1405) | This paper | N/A |
| cCyclinD1 shRNA | Lukaszewicz and Anderson, 2011 | N/A |
| pCAGGS | Hitoshi et al, 1991 | N/A |
| pCAGGS-H2B-EGFP (CX-H2B-GFP) | Gift from K. Hadjantonakis | N/A |
| pCAGGS-H2B-mRFP1 (CX-H2B-RFP) | Gift from S. Tajbakhsh | N/A |
| pUAS-nls-eGFP (UAS-nls-GFP) | Petit-Vargas et al, 2024 | N/A |
| pUAS-H2B-miRFP670 (UAS-H2B-miRFP) | Petit-Vargas et al, 2024 | N/A |
| **Oligonucleotides** | | |
| General oligonucleotide: First hairpin primer 5′ 5′-GGCGGGGCTAGCTGGAGAAGAT GCCTTCCGGAGAGGTGCTGCTGAGCG-3′ | Eurofins Genomics | N/A |
| General oligonucleotide: First hairpin primer 3′ 5′-GGGTGGACGCGTAAGAGGGGAAGAAAGCTT CTAACCCCGCTATTCACCACCACTAGGCA-3′ | Eurofins Genomics | N/A |
| Target forward sequence Cdkn1c shRNA1 5′-GAGAGGTGCTGCTGAGCGAGGCACCGT GCCCGCGTTCTATAGTGAGCCACAGATGTA-3′ | Eurofins Genomics | N/A |
| Target reverse sequence Cdkn1c shRNA1 5′-ATTCACCACCACTAGGCACGGCACCGTG CCCGCGTTCTATACATCTGTGGCTTCACT-3′ | Eurofins Genomics | N/A |
| Target forward sequence Cdkn1c shRNA2 5′-GAGAGGTGCTGCTGAGCGTACGACCCGCATC ACAGATTTTAGTGAACCACAGATGTA-3′ | Eurofins Genomics | N/A |
| Target reverse sequence Cdkn1c shRNA2 5′-ATTCACCACCACTAGGCACACGACCCGC ATCACAGATTTTACATCTGTGGCTTCACT-3′ | Eurofins Genomics | N/A |
| Target forward sequence Cdkn1c shRNA3 5′-GAGAGGTGCTGCTGAGCGGGCGCCGTCTGC AGGAGCTTATAGTGAACCACAGATGTA-3′ | Eurofins Genomics | N/A |
| Target reverse sequence Cdkn1c shRNA3 5′-ATTCACCACCACTAGGCAAGCGCCGTCTGCA GGAGCTTATACATCTGTGGCTTCACT-3′ | Eurofins Genomics | N/A |
| Target forward sequence Cdkn1c shRNA4 5′-GAGAGGTGCTGCTGAGCGTGGAGCCGGGAGA ACCGCGCCGTAGTGAAGCCACAGATGTA-3′ | Eurofins Genomics | N/A |
| Target reverse sequence Cdkn1c shRNA4 5′-ATTCACCACCACTAGGCATGAGCCGGG AGAACCGCGCCGTACATCTGTGGCTTCACT-3′ | Eurofins Genomics | N/A |

| Reagent/resource | Reference or source | Identifier or catalog number |
|---|---|---|
| Target forward sequence Cdkn1c shRNA5 5'-GAGAGGTGCTGCTGAGCGCGACCCGC ATCACAGATTTCTTAGTGAAGCCACAGATGTA-3' | Eurofins Genomics | N/A |
| Target reverse sequence Cdkn1c shRNA5 5'-ATTCACCACCACTAGGCACGACCCGCATCAC AGATTTCTTACATCTGTGGCTTCACT-3' | Eurofins Genomics | N/A |
| Target forward sequence Cdkn1c shRNA6 5'-GAGAGGTGCTGCTGAGCGCTCAATAAACAAAA CAAAAAATAGTGAAGCCACAGATGTA-3' | Eurofins Genomics | N/A |
| Target reverse sequence Cdkn1c shRNA6 5'-ATTCACCACCACTAGGCACTCAATAAAC AAAACAAAAAATACATCTGTGGCTTCACT-3' | Eurofins Genomics | N/A |
| Target forward sequence Cdkn1c gRNA1 5'-CACCGCTGAGCACACCCCCCGCA AG -3' | Eurofins Genomics | N/A |
| Target reverse sequence Cdkn1c gRNA1 5'-AAAGCTTGCGGGGGGGTGTGCTCAG C -3' | Eurofins Genomics | N/A |
| Target forward sequence Cdkn1c gRNA1 5'-CACCGCGGCTCCGCTGAGCCAGG TG -3' | Eurofins Genomics | N/A |
| Target reverse sequence Cdkn1c gRNA1 5'-AAAGCACCTGGCTCAGCGGAGC CGC-3' | Eurofins Genomics | N/A |
| Target forward sequence Cdkn1c gRNA1 5'-CACCGAGCTCCTCACCTGGCT CAG -3' | Eurofins Genomics | N/A |
| Target reverse sequence Cdkn1c gRNA1 5'-AAAGCTGAGCCAGGTGAGGA GCTC-3' | Eurofins Genomics | N/A |
| Target forward sequence Pax7 gRNA1 5'-CACCGCCTGTCTCTACTGGTA GGAG-3' | Eurofins Genomics | N/A |
| Target reverse sequence Pax7 gRNA1 5'-AAAGCTCCTACCAGTAGAGACA GGC -3' | Eurofins Genomics | N/A |
| **Software** | | |
| ImageJ/FIJI | Schneider et al, 2012 | http://imagej.net/Welcome RRID: SCR_003070 |
| Graphpad Prism | Graphpad | http://www.graphpad.com/ RRID: SCR_002798 |
| Microsoft Excel | Microsoft | RRID:SCR_016137 |
| R/RStudio | | https://posit.co/ RRID: SCR_000432 |
| MicroManager | Edelstein et al, 2010 | https://micro-manager.org/ RRID: SCR_000415 |
| Affinity Suite | Affinity | N/A |

## Transcriptomic analysis

### *Production of scRNA-seq data*

Sample preparation of chick cervical progenitors for single-cell RNA sequencing. Three chick embryos at 66 h of embryonic development (E2.75, HH stage 18) were collected and dissected in ice-cold Phosphate Buffered Saline solution (PBS), transferred into ice-cold L15 medium for further dissection to retain only the cervical spinal region (spanning the length of five somites starting from somite number eight). To generate a single-cell suspension, dissection products were then transferred in 250 μl of 37 °C pre-heated papain/L15 solution (Worthington LS003126 Papain—Stock solution = 41.6 mg/ml in 100 ml; Working solution = 50 μl of stock solution diluted in 1.5 ml of L15 medium) and incubated at 37 °C for 15 min in 2-ml Eppendorf tubes. Papain was then replaced with ice-cold L15 medium, and clusters of cells were disaggregated through gentle up-and-down pipetting. Cells were then centrifuged at $300\times g$ for 2 min at 4 °C. The supernatant was removed, and 500 μl of new ice-cold L15 medium was added. Another round of up-and-down pipetting was performed, and cells were then sieved through 30-μm filters to eliminate clumps of poorly dissociated cells. Filtered cells were then centrifuged for 4 min at $300\times g$, and the supernatant was replaced with 1 ml of PBS containing 0.04% bovine serum albumin (BSA). Cells were then centrifuged for 4 min at $300\times g$, 900 μl of supernatant was removed, and cells were then re-suspended in the remaining 100 μl of solution. Quality control was assayed by counting live *versus* dead cells using Trypan blue. Samples with >90% viability were then used for the generation of scRNAseq datasets. After viability assessment, the cell concentration of the samples was adjusted to 1000 cells/μl.

Single-cell transcriptomes generation, cDNA synthesis, and library construction. Single-cell RNA-seq and Illumina sequencing were performed at the Ecole Normale Supérieure GenomiqueENS core facility (Paris, France). The cellular suspension (4000 cells) was loaded on a 10x Chromium instrument to generate 2871 single-cell GEMs, using the manufacturer's instructions (single cell 3' v2 protocol, 10x Genomics). Library construction was performed as per the manufacturer's protocol and then sequenced on a NextSeq 500 device (Illumina) using paired-end (PE) 26/57, generating 533 million reads.

Pre-processing of chick scRNA-seq data. Primary analyses (demultiplexing, UMI processing, mapping, feature assignment, and gene quantification) were performed with Eoulsan 2 (Lehmann et al, 2021). We used as references the NCBI chick reference genome assembly galGal6.fa.gz and a dedicated GTF annotation: galGal6_embryo_spinal_NT_improved.gtf. This annotation was built on top of the NCBI galGal6.ncbiRefSeq.gtf.gz annotation (downloaded from https://hgdownload.soe.ucsc.edu/goldenPath/galGal6/bigZips/genes/). Due to the high number of poorly annotated genes' 3'UTRs in the chick genome, we developed a novel approach based on the re-annotation of the genome with single-cell RNA-seq data (10x Genomics short reads) and long-read bulk RNA-seq (Oxford Nanopore Technologies) from the same cell types in the chicken embryo. To this aim, we set up a genome re-annotation pipeline based on Nextflow. It takes as input the galGal6 chick reference genome and the long-reads RNA-seq data, and it outputs an improved annotation. (Lehmann et al, manuscript in preparation). This pipeline mainly relies on the use of the genome-based analysis tool IsoQuant (Prjibelski et al, 2023) for the transcript reconstruction step. We also added filtering and quality checks of the novel annotation based on the single-cell RNA-seq data.

### *Biological analyses of scRNA-seq data*

Data cleaning and preparation. The chick dataset was subjected to cleaning steps before proceeding with analyses. Filtering was

performed to remove unwanted cells: cells presenting UMI counts below the 0.5th percentile and above the 99.9th percentile, more than 20% UMI counts associated with mitochondrial genes, and more than 0.3% UMI counts associated with hemoglobin genes. This filtered dataset contained 2479 cells. With regard to gene filtering, we kept genes that are detected at least once in at least three cells. All filtering analyses were performed using Seurat v4.0.1. Scater 1.18.6.

Normalization and dimension reduction. Data were log-normalized with Seurat v4.0.1 function "NormalizeData", and confounding factors such as cell cycle phases and gender were then regressed out using the function "ScaleData" (Lehmann et al, 2021). To preserve differences between proliferating and non-proliferating cells, we separated cells in two groups: "cycling" (G2/M and S) and "non-cycling" (G1/G0). Dimension reduction was then performed on scaled data, and 2D representation of the dataset (PCA and UMAP plots) was obtained. After consulting the percentage of variance explained by each dimension, we chose to keep the first 30 components.

Cell classification. First, using the expression of known cell population markers, we removed all cells that were neither progenitors nor neurons such as cells of the mesoderm (expressing *Foxc1/2, Twist1/2, Meox1/2*) and neural crest (expressing *Sox10*) as performed in (Delile et al, 2019). At this stage, 1878 chick cells remained. To better characterize these neural cells, we applied self-defined progenitor (P) and neuron (N) signature scores, using the Seurat function "AddModuleScore". Scores were based on several known and newly identified markers (originating from differential analysis performed in an initial dataset exploration). A detailed list of used marker genes is provided below.

(Progenitor genes = Sox2, Notch1, Rrm2, Hmgb2, Cenpa, Ube2c, Hes5; Neuron genes = Tubb3, Stmn2, Stmn3, Nova1, Rtn1, Mapt).

Clustering and differential expression. In order to identify sub-populations of cells within the population of interest, we then performed graph-based clustering using the Louvain algorithm as implemented in Seurat v4.0.1. Clustering, coupled with differential expression results (obtained using a negative binomial test) did not bring out clusters evocative of a delineation between proliferative and neurogenic progenitor populations, as cells were mainly differentiated by the patterning factors of the dorso-ventral (DV) axis. In order to find other variation sources, we designed a "denoising" strategy based on pseudotime analysis.

Pseudotime analysis. The pseudotime analysis was performed on the whole neural population. In order to identify genes whose expression varies over time, we relied on the "DifferentialGeneTest" function of the trajectory-inference dedicated tool Monocle3, which led to hierarchical clustering of genes along a pseudotime axis. Partitioning around medoids algorithm (PAM) was then applied to cluster cells based on similar gene expression profiles along the pseudotime. We then focused on the *Btg2/Tis21*-containing cluster to look for differentially expressed genes (Fig. 1C).

## Experimental model

Fertilized eggs of JA57 chicken were purchased from EARL Morizeau (8 rue du Moulin, 28190 Dangers, France). Eggs were

incubated at 38 °C in a Sanyo MIR-253 incubator for the appropriate amount of time.

### Cryostat sections

For cryostat sections, chick embryos were collected at E2.25 (HH st13-14), E3 (HH st18), and E4 (HH st22) (Hamburger and Hamilton, 1992) in ice-cold PBS, then fixed overnight in 4% formaldehyde/PBS at 4 °C. The following day, embryos were washed three times for 5 min in PBS at room temperature (RT). Embryos were equilibrated at 4 °C in PBS/15% sucrose, then equilibrated at 42 °C in PBS/15% sucrose/7.5% gelatin solution, embedded in plastic dishes containing 1 ml of PBS/15% sucrose/7.5% gelatin solution and flash frozen in 100% ethanol at −50 °C on dry ice, before storage at −80 °C. Prior to cryostat sectioning, samples were equilibrated for 1 h at −25 °C. In all, 20 μm cryostat sections were obtained using a Leica CM3050S Cryostat and manually mounted on SuperFrost Plus microscope slides, before storage at −20 °C.

### In situ hybridization

For in situ hybridization, gelatin-mounted cryosections were first equilibrated at RT for 15 min, and de-gelatinized by washing slides in 37 °C PBS three times for 5 min. All following steps were carried out at RT unless mentioned otherwise. Slides were bathed for 20 min in RIPA buffer (150 mM NaCl, 1% NP-40, 0.5% Na deoxycholate, 0.1% SDS, 1 mM EDTA, 50 mM Tris pH 8.0), post-fixed in 4% paraformaldehyde/PBS for 10 min, and washed with PBS three times for 5 min. Slides were then bathed in triethanolamine solution (100 mM triethanolamine, acetic acid 0.25% pH 8.0) for 15 min and washed with PBS three times for 5 min. Subsequently, slides were pre-hybridized during 1 h in 69 °C pre-heated hybridization solution (50% formamide, 5× saline-sodium citrate (SSC), 5× Denhardt's, 500 μg/ml herring sperm DNA, 250 μg/ml yeast RNA) and hybridized overnight at 69 °C with the same hybridization solution in the presence of the heat-denatured (95 °C for 5 min) DIG-labeled RNA probes. The following day, slides were transferred in post-hybridization solution (50% formamide; 2× SSC; 0.1% Tween20) at 69 °C for 1 h, then washed in 69 °C pre-heated 2× SSC solution for 30 min, and finally in 0.2× SSC solution at RT for 5 min. Slides were washed with buffer 1 (100 mM maleic acid, pH 7.5, 150 mM NaCl, 0.05% Tween20) during 20 min at RT, blocked for 30 min in buffer 2 (buffer 1/10% fetal calf serum (FCS), followed by overnight incubation at 4 °C with 250 μl of the anti-DIG antibody (Merck #11093274910) diluted 1:2000 and other necessary primary antibodies (when additional immunostaining was needed) in buffer 2. Slides were covered with a coverslip to limit loss of solution during overnight incubation. The following day, coverslips were gently removed and slides were washed with buffer 1, three times for 5 min, and equilibrated for 30 min by bathing in buffer 3 (100 mM Tris pH 9.5, 100 mM NaCl, 50 mM MgCl₂). In situ hybridization signal was visualized through a color reaction by bathing slides in BM-Purple (Merck # 11442074001). The color reaction was allowed to develop in the dark at RT during the appropriate amount of time and was stopped by bathing slides in 4% paraformaldehyde/PBS for 10 min. Sections were finally washed with PBS three times for 5 min before either mounting with coverslip using Aquatex or proceeding with the subsequent immunostaining protocol steps if required (see the section "Vibratome/cryostat sections and immunostaining").

All of RNA-probes were synthesized using a DIG RNA labeling kit (Merck #11277073910) following the manufacturer's protocol. Antisense probes were prepared from the following linearized plasmids: cHes5.1 (previously described in Baek et al, 2018), Cdkn1c (a gift from Matthew Towers, described in Pickering et al, 2019), and cCyclinD1 5' (a gift from Fabienne Pituello, previously described in Lobjois et al, 2004).

### In ovo electroporation

Electroporations were performed at HH13-14 (embryonic day 2, E2) by applying 5 pulses of 25 V for 50 ms, with 100 ms in between pulses. Electroporations were performed using a square wave electroporator (Nepa Gene CUY21SC Square Wave Electroporator, or BTX ECM-830 Electro Square Porator, or Ovodyne Intracell TSS20) and a pair of 5 mm gold-plated electrodes (BTX Genetrode model 512) separated by a 4 mm interval. For bilateral electroporation of the Cdkn1c-Myc and *Pax7*-Cdkn1c-Myc knock-in constructs (Figs. 4C and EV7E), the two injections were performed at 3 h interval, the polarity of the electrode was reversed, and four pulses of 20 V were applied for each electroporation.

### Plasmids

RNA interference: Small interfering RNA sequences against the chick version of Cdkn1c were determined using siDirect: http://sidirect2.rnai.jp/. Target sequences for Cdkn1c are as follows:

Cdkn1c shRNA 1: 5'-CGGCACCGTGCCCGCGTTCTA-3';
Cdkn1c shRNA 2: 5'-CACGACCCGCATCACAGATTT-3';
Cdkn1c shRNA 3: 5'-AGCGCCGTCTGCAGGAGCTTA-3';
Cdkn1c shRNA 4: 5'-TGAGCCGGGAGAACCGCGCCG-3';
Cdkn1c shRNA 5: 5'-CGACCCGCATCACAGATTTCT-3';
Cdkn1c shRNA 6: 5'-CTCAATAAACAAAACAAAAAA-3'.

Target sequences were cloned into the first hairpin of the miR30-derived structure of the pTol2-H2B-EGFP-miRNA plasmid (Peyre et al, 2011) using the following method (Das et al, 2006): 100 ng of both general oligonucleotides (first hairpin primer 5': 5'-GGCGGGGCTAGCTGGAGAAGATGCCTTCCGGA-GAGGTGCTGCTGAGCG-3' and first hairpin primer 3': 5'-GGGTGGACGCGTAAGAGGGGAAGAAAGCTTC-TAACCCCGCTATTCACCACCACTAGGCA-3') were used together with 10 ng of both target-specific oligonucleotides (Target forward sequence: 5'-GAGAGGTGCTGCTGAGCG_**FORWARD-TARGETSEQUENCE**_TAGTGAAGCCACAGATGTA-3' and Target reverse sequence: 5'-ATTCACCACCACTAGGCA_**RE-VERSETARGETSEQUENCE**_TACATCTGTGGCTTCACT-3') in a one-step PCR reaction to generate a product containing the miR30 like hairpin and the chick miRNA flanking sequences. Obtained PCR products and the pTol2-H2B-EGFP-miRNA plasmid were submitted to NheI/MluI double enzymatic digestion, and purified digested products were then ligated to create Cdkn1c miRNA plasmids.

The cCyclinD1 (chick CyclinD1) shRNA plasmid was previously described in (Lukaszewicz and Anderson, 2011), and was a kind gift of Dr Fabienne Pituello. A plasmid coding for a combination of a shRNA against Luciferase and a GFP reporter was used as a control (described in (Peyre et al, 2011). An empty pCAGGS plasmid was used to match total DNA concentrations between experimental and control electroporation mixes when needed. All miRNA and shRNA plasmids were used at 1 µg/µl except when otherwise mentioned.

Somatic knock-ins: Somatic knock-in of a *6xMyc-P2A-Gal4-VP16* reporter at the C-terminus of the *Cdkn1c* locus was achieved via CRISPR-Cas9-based microhomology-mediated end joining (MMEJ). The *6xMyc-P2A-Gal4-VP16* cassette in the targeting vector is flanked by 37 bp 5' and 42 bp 3' arms of homology corresponding to the genomic sequence immediately upstream and downstream of the stop codon of the *Cdkn1c* locus. These arms of homology are flanked on both ends by a universal "uni2" gRNA target site that does not target any sequence in the chick genome (GGGAGGCGTTCGGGCCACAG; Welker et al, 2021, Petit-Vargas et al, 2024). An "armless" vector with the same *6xMyc-P2A-Gal4-VP16* cassette and uni2 linearization site, but without the arms of homology, was constructed as a control for the "locus-specificity" of the insertion and signal. Details of the constructs and cloning steps are available upon request. The MMEJ-based knock-in method relies on the simultaneous linearization of the target locus and of the targeting vector in cells. This is achieved by coexpression of two gRNAs, one targeting the genomic locus, the other (uni2) targeting the knock-in vector. We generated a double gRNA construct that possesses two cassettes, each expressing a chimeric gRNA under control of the human U6 promoter. This vector, derived from pX330 (Cong et al, 2013); Addgene #42230), also expressed humanized spCas9 (Cas9) protein under the CBh promoter. We chose three different gRNAs located in the vicinity of the *Cdkn1c* stop codon, using the CRISPOR website (http://crispor.tefor.net/crispor.py). The sequence targeted by gRNA#1 (CTGAGCACACCCCCCGCAAG) is located 12 bases upstream of the *Cdkn1c* stop codon in the sense direction and entirely comprised in the left arm of homology. In order to avoid targeting of the knock-in vector and of the modified locus after insertion of the knock-in cassette, the target sequence for gRNA#1 was destroyed in the left arm of homology via two conservative base changes in the last base of the recognition and in the PAM (see Fig. EV1B). Upon initial validation of the knock-in efficiency with a UAS-nls-GFP reporter, gRNA#1 yielded the strongest GFP signal of the three gRNAs and was chosen for all subsequent experiments. A gRNA that does not target any sequence in the chick genome was used as a control (GCACTGCTACGATCTACACC; (Gandhi et al, 2017)) and did not yield any GFP signal.

Somatic knock-in of the *Cdkn1c* coding sequence in the *Pax7* locus was achieved via CRISPR-Cas9-based Homology-Directed Recombination (HDR). A *Pax7-P2A-Gal4* knock-in vector and gRNAs were first generated and validated for efficient and specific targeting at the C-terminus of the *Pax7* locus (for details see Petit-Vargas et al, 2024). In this vector, the Gal4-VP16 cassette is flanked with long left (1056 bp) and right (936 bp) arms of homology to the C-terminal region of *Pax7*. This vector was then modified by inserting an 846 bp synthetic DNA fragment (IDT) coding for a P2A sequence, chick *Cdkn1c*, and three Myc tags, immediately downstream of the Gal4-VP16 sequence. This places a *P2A-Gal4-VP16-P2A-Cdkn1c-3xMyc* cassette in frame with the C-terminus of *Pax7*. The introduction of two P2A pseudocleavage sequences ensures that Pax7, Gal4-VP16, and Cdkn1c-Myc are produced as three independent proteins from the *Pax7* locus in dorsal progenitors that have undergone homologous recombination. The *Pax7* gRNA targets the GGGCTCCTACCAGTAGAGAC sequence 16 bases upstream of the *Pax7* stop codon in the sense direction, and is entirely comprised upstream of the stop codon. In order to avoid targeting of the knock-in plasmid and re-targeting of the

locus after insertion of the knock-in cassette, the gRNA target sequence was destroyed in the left arm of homology via insertion of three bases (AGA), two bases upstream of the PAM. This inserts an Arginine 5 amino acids upstream of the C-terminus of *Pax7* (see Fig. EV7C). In addition to this extra amino acid, a P2A sequence is appended at the C-terminus of the Pax7 protein. We did not attempt to monitor whether this modification of the Pax7 C-terminus modifies its activity. For consistency, control experiments for the *Pax7*-driven misexpression of *Cdkn1c-P2A-Gal4* were performed in a *Pax7-P2A-Gal4 knock-in* background using the parental Pax7-P2A-Gal4 knock-in construct described in Petit-Vargas et al, 2024, which generates the same modification of the *Pax7* C-terminal sequence.

For in ovo knock-in experiments, the homologous recombination and gRNA vectors were each used at 0.8 µg/µl. The UAS reporter plasmid (pUAS-nls-EGFP) was added to the electroporation mix at 0.3 µg/µl.

### Western blotting

For western blots, E2 embryos were electroporated with the Cdkn1c-Myc knock-in construct, the vector co-expressing Cas9 and either the control gRNA, Cdkn1c gRNA#1, or Cdkn1c gRNA#3 gRNA, and a H2B-mRFP1 electroporation reporter construct. For each condition, electroporated hemitubes from 5 embryos were dissected 30 h after electroporation and transferred to a 1.5-ml tube in 30 µl PBS. Two volumes (60 µl) of RIPA buffer and a Protease Inhibitor cocktail were added, and the tissue was pipetted up and down 50 times with a 200 µl pipette, frozen at −20 °C, thawed and pipetted up and down 50 more times. Loading buffer was added to a 1× concentration, and the tubes were boiled for 5 min at 95 °C. They were spun at maximum speed in a microcentrifuge, and the supernatant was transferred to a clean tube and frozen for storage.

Protein concentrations were determined by Bradford protein assay. For each condition, 50 µg of proteins mixed with SDS loading dye were separated by electrophoresis in 4–12% Tris-glycine SDS-PAGE gels (Life Technologies) and transferred onto 0.2-µm nitrocellulose membranes (Protran; Amersham) using Thermo Fischer Scientific Power Blotter Systems. Membranes were blocked in PBS containing 10% (w/v) milk and 0.05% Tween-20 (Euromedex) (PBS-T) for 30 min at room temperature (RT) and incubated overnight at 4 °C with anti-Myc and anti-RFP primary antibodies (1:1000 in PBS-T with 10% milk). After three washes with PBS-T, membranes were incubated with goat anti-mouse (1:5000; Promega) or goat anti-rat (1:5000) secondary antibodies. Signals were detected using SuperSignal West Femto or SuperSignal West Pico PLUS (Thermo Fischer Scientific) and imaged with the LAS-4000 mini system (GE Healthcare).

### Vibratome sections

For vibratome sections, chick embryos were collected at desired stages and roughly dissected to remove membranes in ice-cold PBS, fixed for 1 h in ice-cold 4% formaldehyde/PBS, and rinsed three times for 5 min in PBS at RT. Chick embryos were then finely dissected in PBS and subsequently embedded in 4% agarose (4 g of agarose in 100 ml of water, boiled in a microwave and cooled at 50 °C) until agarose became solid. Thereafter, 100-µm vibratome sections were realized using a ThermoScientific HM 650 V Microtome and collected in six-well plates filled with cold PBS.

### Immunostaining

Sections were permeabilized in PBS-0.3% Triton for 30 min at RT, and then incubated with the primary antibodies diluted in the blocking solution (PBS-0.1% Triton/10% FCS) at 4 °C overnight with gentle agitation. The following day, sections were washed three times for 5 min in PBS at RT, incubated 4 h in the dark at RT or ON at 4 °C with the appropriate secondary antibodies (and DAPI if needed) diluted in PBS-0,1% Triton, washed again three times for 5 min at RT with PBS and mounted with Vectashield (with or without DAPI, depending on experiment—Vector Laboratories H-1000-10 & H-1200-10).

All immunostainings on slide-mounted cryosections were performed during and after the end of in situ hybridization revelation protocol. Slides were incubated with primary antibodies during the appropriate step described above in the "Cryostat" and "In situ hybridization" sections. After in situ hybridization revelation, slides were incubated 4 h in the dark at RT with 250 µl of appropriate secondary antibodies (and DAPI, if needed) diluted in PBS-0.1% Triton, washed again three times for 5 min at RT with PBS, and mounted with Aquatex.

Primary antibodies used are: chick anti-GFP (GFP-1020—1:2000) from Aves Labs; goat anti-Sox2 (clone Y-17—1:1000) from Santa Cruz; rabbit anti-pRb (Ser807/811— 1:1000) from Cell Signaling; mouse anti-c-myc tag (Clone 9E10—1:100) from Sigma-Aldrich; mouse anti Pax7 (monoclonal, 1:100) from DSHB; rabbit anti-phospho-Histone 3 (Polyclonal—1: 250) from Millipore; rabbit anti-active Caspase3 (Polyclonal—1:300) from R&D systems; mouse anti-HuC/D (clone 16A11—1:50) from Life Technologies. Secondary antibodies coupled to Alexa Fluor 488, Cy3, or Alexa Fluor 649 were all obtained from Jackson Laboratories and all used at a 1:500 dilution.

### EdU labeling

Proliferating progenitors in the neural tube were labeled with 5-ethynyl-2′-deoxyuridine (EdU, Click-iT EdU imaging kit, Invitrogen #C10338) via in ovo incorporation. Before EdU injection, membranes surrounding the embryos were slightly opened using forceps. For 1-h pulse experiments, 100 µl of a 500 µM solution of EdU diluted in PBS was deposited directly on the embryo through the opened membranes. For cumulative EdU labeling, embryos were incubated with EdU for the appropriate amount of time before collection. In this context, 100 µl of a 500 µM solution of EdU diluted in PBS was deposited every 6 h through the previously opened space after initial injection. After collection, embryos were subsequently processed following the vibratome sections protocol. Revelation of EdU incorporated in progenitors was carried out on vibratome sections after the permeabilization step, using the Click-iT EdU imaging kit according to the manufacturer's protocol.

### FlashTag preparation and injection

A 1 mM stock solution of CellTrace Far Red (Life Technologies, #C34564 - (Baek et al, 2018)) was prepared by adding 20 µl of DMSO to a CellTrace Far red dye stock vial. A working solution of 100 µM was subsequently prepared by diluting 1 µl of stock solution in 9 µl of 37 °C pre-heated PBS, and injected directly into E3 chick neural tubes. The eggs were resealed with parafilm, and embryos were incubated at 38 °C for the appropriate time until dissection.

### Image acquisition

Transverse sections of chick embryo neural tubes after in situ hybridization and/or immunofluorescence were obtained either on a confocal microscope (model SP5; Leica) using ×40 and ×63 (Plan Neofluar NA 1.3 oil immersion) objectives and Leica LAS software, or on an inverted microscope (Nikon TiEclipse) equipped with a Yokogawa CSU-WI spinning disk confocal head, a Borealis system (Andor Technologies) and an sCMOS Camera (Orca Flash4LT, Hamamatsu) using a 40× objective (CFI Plan APO LBDA, NA 0.95, Nikon) or a ×100 oil immersion objective (APO VC, NA 1.4, Nikon) and micromanager software (Edelstein et al, 2010). For image processing, data analysis, and quantification, we used the Fiji software to adjust brightness and contrast.

### Image quantifications

In the ventral motor neuron domain of the neural tube, progenitors differentiate earlier than in any other region of the neural tube. Thus, to reason on a more homogeneous progenitor population, we restricted all our analysis to the dorsal two-thirds of the neural tube, except for the *Pax7*-Cdkn1c misexpression analysis, which was performed in the more dorsal Pax7 domain. All cell counting in this study was performed manually.

### Quantification of Cdkn1c-Myc levels

Cdkn1c-Myc levels measurements in the context of endogenous Myc-tagging of Cdkn1c or *Pax7*-driven Cdkn1c-Myc misexpression were performed on 100x spinning disk confocal images (Z-stacks, 31 z-levels at 1 μm interval) acquired from vibratome sections.

Raw stacks were first pre-treated as follows in FIJI to remove background and normalize orientation of the ventricular surface: (1) A "background" image of a similar stack acquired in an empty region of the slide was acquired, an average projection of this full stack was generated, and restacked as an "average background" stack of 31 identical z-levels, which was then subtracted for each sample stack. (2) A custom macro ("Zprogressive_background-substractor.ijm") was used, in which the user selects a ROI with no specific signal in the tissue in any z-level of the slice in the three channels of interest (GFP, Myc, pRb) to define the "tissue background", which corresponds to the mean signal in this ROI (3*31 values). This channel and z-level specific "tissue background" is then subtracted in each corresponding channel and z-level, to account for the progressive decrease of background in successive z-levels. In addition, during this step, a value of 5 was further subtracted from the GFP and Myc channels, and a value of 60 from the pRb channels, since we observed that residual background corresponding to these values was still detected in "cell-less" parts of the tissue (typically, in the lumen of the neural tube). After background subtraction, the image was rotated and cropped at the level of the apical surface of the electroporated side, such that it was aligned with the horizontal axis: as a result, the $y$ value of any pixel in the image corresponds to its distance from the apical surface.

For measurements, a second custom macro ("Macro_measure Myc-GFP-pRb-intensity in ×100 images from gRNA1.ijm") was used. Cells of interest (cells with a visible Myc and/or GFP signal) were selected by the user with the point tool at the z-level corresponding to the center of the nucleus. A circle of fixed diameter is automatically overlaid on the selected cell at the corresponding z-level, and the mean signal intensity in all three GFP, Myc, and pRb channels, as well as $x$, $y$ (distance to the apical surface) and $z$ position are recorded in a "final measurements" table.

Since the measurements are obtained from a single z-level, and from a fixed-size circle, they do not correspond to a complete measurement averaged from the whole signal obtained for each cell, but are a representative approximation of signal intensity.

In addition, graphs in Figs. 2E and EV1F show the Myc values as a function of pRb signal positivity. To determine pRb status, a threshold of 30 was applied to the pRb values measured with the macro described above, and cells with a pRb signal intensity between 0 and 30 were considered as negative, whereas cells with a pRb signal intensity higher than 30 were considered as positive.

The two macros are provided in the Source File accompanying this manuscript.

### Status of proliferation/differentiation balance at the tissue level

The status of the proliferation and differentiation in different experimental conditions was analyzed in electroporated cell populations, identified on the basis of GFP reporter expression in transverse vibratome sections of embryos one or two days after electroporation at HH13-14. Unambiguous identification of cycling progenitors and postmitotic neurons is notoriously difficult in the chick spinal cord, as there are no reliable reagents that label these populations: markers of neurons, such as HuC/D of bIII-tubulin (TujI) are not detected during the first hours of neural differentiation; on the other hand, markers of progenitors usually either do not label all the phases of the cell cycle (eg. Phospho-Rb, thereafter pRb (Molina et al, 2022), or persist transiently in newborn neurons (e.g., Sox2 (Coquand et al, 2024)). With these limitations in mind, we used antibodies against HuC/D to label neurons and phosphorylated Rb to identify the progenitor population. This leaves a population of "undetermined cells" that are negative for both markers, and that can be either progenitors in G1 phase before the restriction point and are therefore still pRb negative (Molina et al, 2022), or newborn neurons that have not yet activated HuC/ D. For conditions analyzed using a combination of GFP, pRb, and HuC/D primary antibodies, two ratios were determined. The progenitor ratio was obtained by dividing the number of shRNA-electroporated (GFP-positive) and pRb-positive cells by the total number of electroporated cells (GFP-positive). Neuron ratio was obtained by counting the number of shRNA-electroporated (GFP) HuC/D-positive cells over the total number of electroporated cells (GFP-positive).

### Cumulative EdU incorporation in Cdkn1c knockdown and control progenitors

Chick embryos were electroporated at E2 with either control or Cdkn1c shRNA vectors, and EdU injections were performed *in ovo* starting at E3 and then every 6 h to cover the whole cell cycle.

At each measured timepoint (1, 4, 7, 10, 12, 14, and 17 h after the first EdU injection), we quantified the number of EdU-positive electroporated progenitors (triple positive for EdU, pRb, and GFP) over the total population of electroporated progenitor cells (pRb and GFP-positive) (Fig. 5A). The average values of cycling progenitors obtained for each time point were then plotted to construct the graphs. The number of embryos, sections, and cells quantified for each timepoint in each condition is detailed below.

Cdkn1c shRNA condition: A minimum of 730 cells, collected from 3 to 5 embryos, were analyzed for each timepoint, from 1 to 2 different experiments.

Control shRNA condition: A minimum of 702 cells, collected from 3 to 5 embryos were analyzed for each timepoint, from 1 to 2 different experiments.

### Quantification of progenitors in S phase at a given time point (1 h EdU pulse)

Control shRNA, Cdkn1c shRNA, cCyclinD1 shRNA, or a combination of Cdkn1c and cCyclinD1 shRNAs and a reporter of electroporation (GFP), and were electroporated in HH13-14 chick embryos. Two days after shRNAs electroporation, we injected EdU in ovo 1 h before collecting the embryos. We then labeled transverse sections for EdU incorporation as described in the EdU labeling section. We quantified the proportion of progenitors in S phase in the shRNA conditions (GFP/EdU/pRb-positive cells) over the global population of electroporated progenitors (GFP/pRb-positive). For each condition, a minimum of 1594 cells collected from 3 to 6 embryos were analyzed.

### G1 analysis of neural progenitors at the cell level

We used the FlashTag (FT) technique, based on the ability of the cell-permeant dye CellTrace Far Red (Life Technologies, #C34564) to fluorescently label intracellular proteins. Previous experiments in the embryonic chick have shown that upon direct injection in the neural tube, FT dyes preferentially enter progenitor cells undergoing mitosis near the apical surface and that this incorporation only occurs during a 15–30 min time window (Baek et al, 2018). Since FT fluorescence is preserved in daughter cells after mitosis, this dye offers a convenient means to synchronously label a cohort of cells dividing at the time of injection and follow their progeny.

Using a combination of FT injection and cumulative EdU incorporation allows to monitor precisely the average length of the G1 phase. Daughter cells from FT-positive progenitors enter G0/G1 phase just after mitosis, and will start incorporating EdU only when entering S phase. For each cell in a FT cohort, the time window between FT injection and the beginning of EdU incorporation corresponds to the duration of the G1 phase. One day after electroporation, FT and EdU were injected simultaneously, and embryos were collected at different time points after injection to identify the time at which all cells in the FT cohort have exited G1 and entered S phase. For time points over 6 h 30, an additional EdU injection was performed after 6 h to cumulatively label the whole population of cycling cells.

We quantified the number of electroporated (GFP-positive) progenitors (pRb-positive) having incorporated EdU and FT (GFP/pRb/EdU/FT quadruple positive) relative to the number of FT electroporated progenitors (GFP/pRb/FT triple positive). At each time point, the percentage of quadruple-positive cells represents the proportion of progenitors having completed their G1 phase. This percentage reaches a plateau when all the progenitors in the FT cohort have entered S phase. Therefore, an experimental FT cohort that reaches the plateau faster than the control FT cohort has a shorter G1 phase duration.

A similar approach was used for the rescue experiment of G1 length reduction observed upon Cdkn1c knockdown by a concomitant downregulation of CyclinD1 (Fig. 6B), except that only the 6 h 30 min time point was generated.

### Clonal analysis of sister cell identities and mode of division

We first determined the time point after mitosis at which pRb becomes a reliable progenitor marker by monitoring the time window after which all progenitors in a synchronized cohort of cells undergoing mitosis reach the late G1 stage, as determined by pRb immunoreactivity. Using FT to label a cohort of pairs of sister cells that perform their division synchronously at E3, we counted the proportion of pairs in the cohort that contained 0, 1, or 2 cells positive for pRb at different time points after FT injection. The distribution between these three categories should reach a plateau when all the progenitors in the cohort have become positive for pRb. At E3, this plateau was reached between 4 h 30 and 6 h after FT injection, and the distribution of pRb immunoreactivity within pairs of FT-positive sister cells was stable at later time points, indicating that from 6 h after FT injection, the proportions of FT pairs with 0, 1, or 2 pRb-positive cells respectively correspond to the proportions of NN, PN and PP pairs in the cohort. We therefore choose to perform clonal analysis in embryos harvested 6 h after FT injection.

For the clonal analysis strategy to analyze the fate of pairs of sister cells born from the division of mother cells downregulated for Cdkn1c (Fig. 3H), chick embryos were electroporated at E2 with the relevant shRNAs. For clonal analysis strategy to analyze the fate of pairs of sister cells born from the division of mother cells misexpressing Cdkn1c (Fig. 4F) chick embryos were electroporated at E2 with a Myc-tagged Cdkn1c coding sequence and the Gal4-VP16 transcription factor downstream of Pax7 (Figs. 4B and EV7B–D) or a control construct that introduces the Gal4-VP16 reporter alone to the Pax7 locus (Petit-Vargas et al, 2024), and a UAS-nls-GFP reporter. One day later, FT was injected in the neural tube in order to follow the progeny of isochronically dividing neural progenitors. Cell identity of transfected GFP-positive cells was determined as follows: cells positive for pRb and FT were classified as progenitors, and cells positive for FT and negative for pRb as neurons. In addition, a similar intensity of both the GFP and FT signals within pairs of cells, and a relative position of the two cells consistent with the radial organization of clones in this tissue (Leber and Sanes, 1995; Loulier et al, 2014) were used as criteria to further ascertain sisterhood. This combination restricts the density of events fulfilling all these independent criteria, and can confidently be used to ensure a robust identification of pairs of sister cells. The mode of division used by their mother cell was determined as follows: symmetric proliferative if the two daughter cells were attributed the progenitor identity (PP); asymmetric if one of the daughters was a progenitor and the other daughter a neuron (PN); and terminal neurogenic if the two daughter cells had a neuronal identity (NN).

To monitor the number of differentiated neurons in a synchronous cohort of newborn cells born from the division of mother cells misexpressing Cdkn1c (Fig. 3H), we injected FlashTag 22 h after electroporation and collected the embryos 20 h after FlashTag injection (Figs. 4G and EV8B). Neuronal identity of transfected GFP/FT-positive cells was assessed by the detection of HuC/D signal in vibratome transverse sections.

### Statistical analyses

The number of embryos and analyzed cells or sections are indicated in the legends to the figures. All data processing and statistical analyses were performed using Excel, GraphPad Prism, or R software, and the tests are indicated in the legends to the figures.

Statistical significance of *P* values is represented on graphs as: ns, $P > 0.05$; *$P < 0.05$; **$P < 0.01$; ***$P < 0.001$; ****$P < 0.0001$. Exact *P* values for *, **, *** are provided in the legends to figures.

### Contact for reagent and resource sharing

As Lead Contact, Xavier Morin (Institut de Biologie de l'Ecole Normale Supérieure) is responsible for all reagent and resource requests. Please contact Xavier Morin at xavier.morin@bio.ens.p-sl.eu with requests and enquiries.

## Data availability

Single-cell RNA sequencing data have been deposited in NCBI's Gene Expression Omnibus (GEO) repository under the accession number GSE273710. Source data for the main figures have been deposited in the BioImages repository and are accessible at https://www.ebi.ac.uk/biostudies/bioimages/studies/S-BIAD2283.

The source data of this paper are collected in the following database record: biostudies:S-SCDT-10_1038-S44319-025-00653-9.

## Peer review information

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

## Acknowledgements

We thank our colleague Samuel Tozer for discussions and critical reading of the manuscript. We thank Fabienne Pituello and Matthew Towers for plasmids. We thank Benjamin Bunel for the original drafting of the Figures. This work was supported by grants from the Fondation pour la Recherche Medicale (FRM EQU202003010547), the Labex MEMO LIFE, the Fondation Cino del Duca to XM, the Institut Universitaire de France to MTC, and a joint grant from the Agence Nationale de la Recherche (SYMASYM ANR-18-CE16-0021-01) to MTC and XM. BM was supported by doctoral grants from the French Ministry of Higher Education and Research (MESR) and the Labex MEMO LIFE. This work has received support under the program «Investissements d'Avenir» launched by the French Government and implemented by the Agence Nationale de la Recherche, with the reference ANR-10-LABX-54 MEMO LIFE. The GenomiqueENS core facility was supported by the France Génomique national infrastructure, funded as part of the "Investissements d'Avenir" program managed by the Agence Nationale de la Recherche (contract ANR-10-INBS-09).

## Author contributions

**Baptiste Mida**: Formal analysis; Investigation; Visualization. **Nathalie Lehmann**: Data curation; Software; Formal analysis; Visualization; Methodology; Writing—review and editing. **Rosette Goïame**: Investigation. **Fanny Coulpier**: Investigation. **Kamal Bouhali**: Investigation. **Isabelle Barbosa**: Investigation. **Hervé Le Hir**: Investigation. **Morgane Thomas-Chollier**: Supervision; Funding acquisition; Methodology; Project administration; Writing—review and editing. **Evelyne Fischer**: Conceptualization; Supervision; Validation; Investigation; Visualization; Methodology; Writing—original draft; Project administration; Writing—review and editing. **Xavier Morin**: Conceptualization; Supervision; Funding acquisition; Validation; Investigation; Visualization; Methodology; Writing—original draft; Project administration; Writing—review and editing.

Source data underlying figure panels in this paper may have individual authorship assigned. Where available, figure panel/source data authorship is listed in the following database record: biostudies:S-SCDT-10_1038-S44319-025-00653-9.

## Disclosure and competing interests statement

The authors declare no competing interests.

# Expanded View Figures

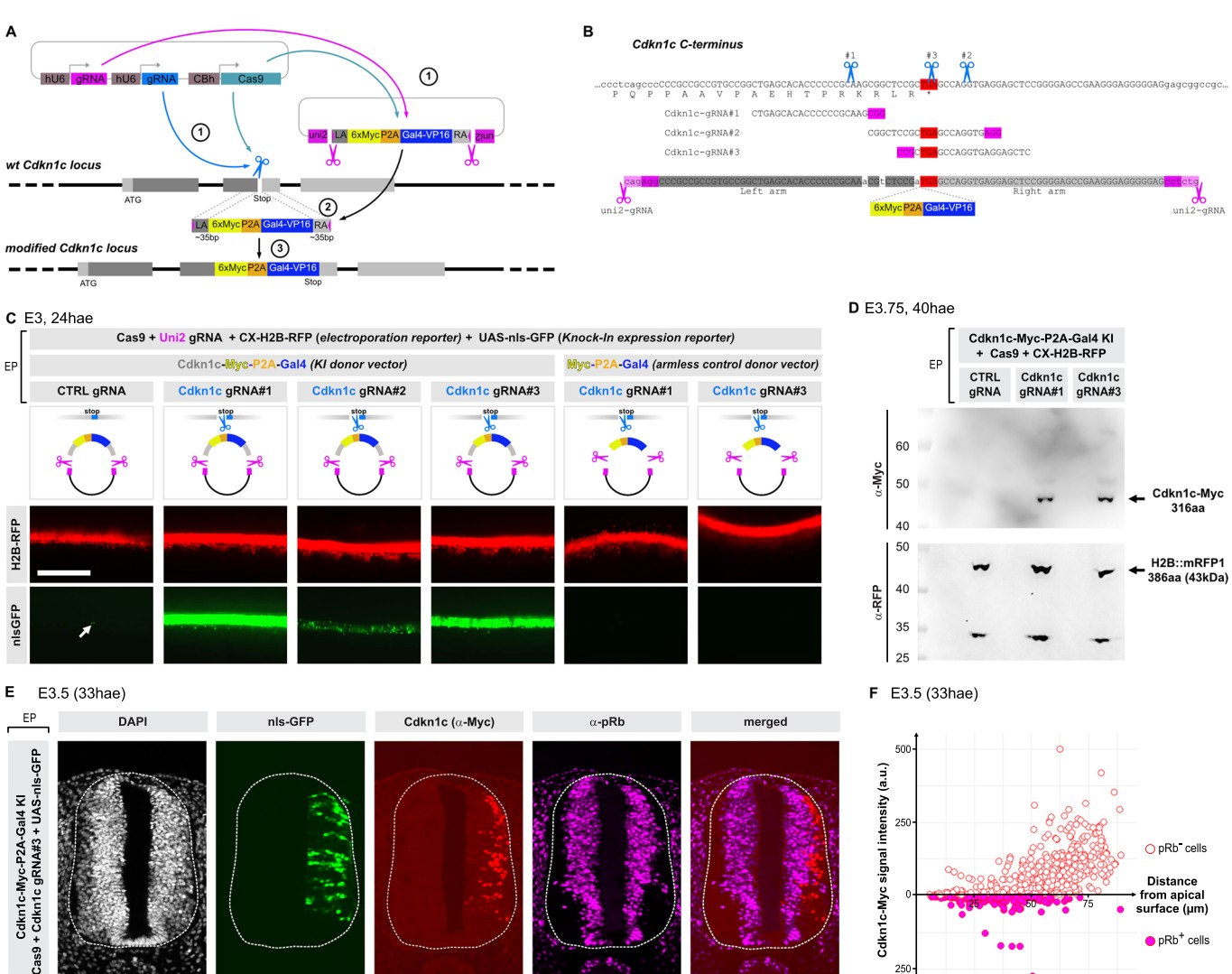

**Figure EV1. Somatic knock-in strategy to target the endogenous *Cdkn1c* locus with Myc tags and monitor the dynamic expression of the protein in spinal cord progenitors.**

(A) Microhomology Mediated End Joining (MMEJ) strategy used for the somatic knock-in. The donor plasmid carries short arms of homology ( < 35 bp) to the *Cdkn1c* locus at the level of the C-terminus, flanking a sequence consisting of 6 Myc tags in frame with *Cdkn1c* coding sequence, a P2A pseudo cleavage sequence and the Gal4-VP16 synthetic transcription factor. This donor cassette is flanked on both sides by target sites for a "universal" guide RNA (uni2 gRNA) designed to trigger linearization of the vector and to release the donor cassette as a linear double-stranded DNA fragment in the electroporated cells. Somatic knock-in is achieved by the coelectroporation of a CRISPR/Cas9 vector that expresses the Cas9 protein, the uni2 gRNA targeting the donor vector for linearization, and the gRNA targeting the *Cdkn1c* locus. (B) Details of the targeted genomic sequence for Myc tagging of the endogenous *Cdkn1c* locus. Genomic sequence at the level of *Cdkn1c* C-terminus (top), sequence of the three gRNAs (middle), and sequence of the arms of homology used in the MMEJ construct (bottom, sequence highlighted in blue). The three bases highlighted in white represent silent base changes introduced in the left arm of homology to prevent recognition and cleavage of the donor vector by gRNA#1. Arrows indicate the theoretical cut sites of the three gRNAs on the target locus. (C) Validation of the efficiency and specificity of the knock-in strategy: the donor vector was co-electroporated with CRISPR/Cas9 vectors expressing either a control gRNA (CTRL gRNA) or one of the three gRNAs targeting the *Cdkn1c* locus. A UAS-nls-GFP vector was included in the electroporation mix to report expression of the Gal4-VP16 transcription factor in addition to an electroporation reporter (CX-H2B-mRFP). One representative embryo (out of 4–6) is shown for each condition, with similar electroporation level (red). gRNA#1 led to a strong GFP signal (green), showing the greatest efficiency. gRNA#3 appeared slightly less efficient, and gRNA#2 yielded a much lower signal. An "armless" construct lacking homology to the *Cdkn1c* locus was used as an additional negative control with gRNA#1 and gRNA#3. Specificity is demonstrated by the virtual absence of background GFP signal when the control gRNA or the armless donor vector are used (white arrow points to two GFP-positive cells or clusters of cells observed in the control gRNA embryo). Scale bar: 300 μm. (D) Western blot analyses on protein extracts from neural tubes electroporated with the *Cdkn1c-Myc* knock-in construct and control (CTRL) gRNA, Cdkn1c gRNA#1 or Cdkn1c gRNA#3, and a H2B-mRFP1 electroporation reporter construct. The anti-Myc antibody (top panel) reveals a strong and specific band with both gRNA#1 and #3, and no signal at all with a control gRNA. Of note, the ~45 kDa apparent size is higher than the expected ~35 kDa for the 316 aa long Cdkn1c-6xMyc fusion protein. This is consistent with previous observations with human and mouse Cdkn1c, which display a major band at 57 kDa in western blots despite a theoretical size of 35 kDa (Lee et al, 1995; Matsuoka et al, 1995). Bottom panel: detection of H2B-mRFP1 protein with an anti-RFP antibody was used as a loading control and reveals bands of similar intensity in all three conditions. (E) Endogenous Cdkn1c-Myc protein expression pattern at E3.5 (33 hae) using guide #3 to target *Cdkn1c* locus. Somatic knock-in of Myc tags at the *Cdkn1c* locus allows the visualization of Cdkn1c protein using an anti-Myc immunofluorescence (Myc, red) on transverse vibratome sections. Inclusion of a Gal4-VP16 transcription factor in the knock-in construct identifies all the cells which express or have previously expressed Cdkn1c via the UAS-nls-GFP reporter (green). Progenitor population was visualized by an anti-phospho-Rb antibody (pRb, magenta). The neural tube contour is highlighted by a dashed line. High levels of endogenous Cdkn1c-Myc signal was observed in the mantle zone contiguous to the ventricular zone while weaker Myc signal is observed in the progenitor zone. Scale bar: 50 μm. (F) Quantification of Cdkn1c-Myc expression in relationship with pRb expression along the apico-basal axis. Quantification of the Myc signal intensity from a *Cdkn1c-Myc* knock-in insertion was performed in vibratome sections from embryos electroporated with the donor vector, gRNA#3 and a UAS-nls-GFP reporter (E). Knock-in cells that express or have expressed Cdkn1c were identified on the basis of Myc and/or GFP positivity. The Myc signal intensity in individual is plotted on the Y axis towards the upper (pRb + ) and lower (pRb-) parts of the graph as a function of their position along the apico-basal axis (X-axis). Data are from 558 cells from 12 sections from 5 embryos. hae: hours after electroporation.

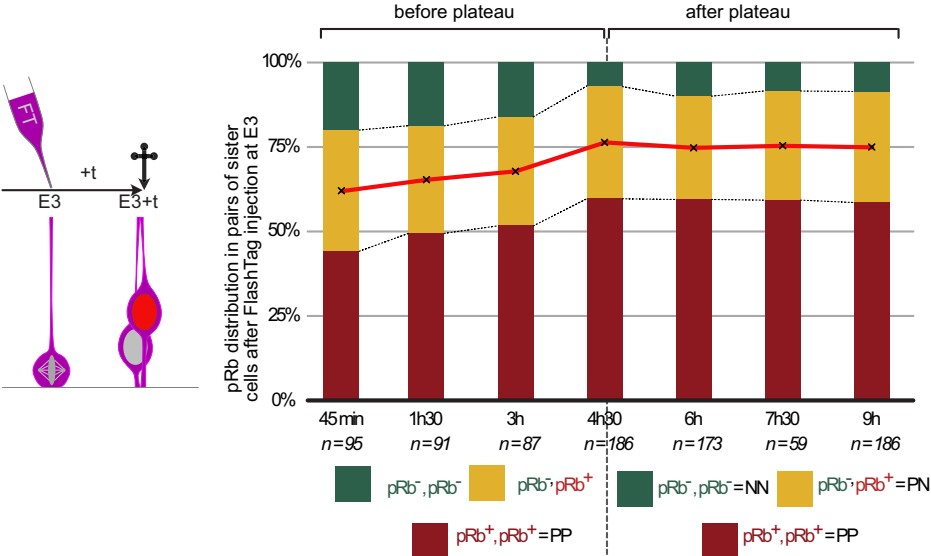

Time course of Rb phosphorylation in cohorts of pairs of sister cells labelled at E3 with FlashTag

**Figure EV2. pRb is a reliable marker of progenitor cells 6 h after mitosis.**

Left: Experimental scheme for the determination of Rb phosphorylation status in FlashTag-labeled cells. Wild-type E3 embryos are injected with FlashTag (FT), which specifically labels cells synchronously undergoing mitosis at the time of injection. Embryos are collected at different timepoints (t) after injection to determine at which timepoint pRb positivity reaches a plateau in the cohort of FlashTag-positive daughter cells. Right: Time course of pRb expression in FlashTag-positive pairs of sister cells at consecutive time points after injection. FlashTag injection was performed at E3 and embryos were harvested at the indicated timepoints after injection. Thoracic vibratome sections were immunostained with anti-pRb antibody to evaluate the pRb status in pairs of FlashTag-positive sister cells. The proportion of pairs with two pRb-positive cells (red), one pRb-positive cell (yellow) or zero pRb-positive cell (green) becomes stable after 4h30. This indicates that the proportion of individual pRb-positive cells (red line) reaches a plateau corresponding to the proportion of cycling progenitors in this cohort. Therefore, after that time point, pRb positivity becomes a reliable marker of progenitor status in FlashTag labeled cells. This allows the retrospective attribution of the mode of division (PP, PN or NN) of the mother of pairs of sister cells via pRb labeling. $N = 2$ to 5 embryos per time point. The number of pairs analyzed at each time point is indicated at the bottom of the diagram.

**Figure EV3.  Predominant expression of *Cdkn1c* transcript in the G1 phase of neural progenitors.**

The "Neurogenic Progenitor" population identified in the scRNAseq dataset (362 cells from a single dataset) was split in two populations [G1 ($n = 282$ cells) and S/G2/M ($n = 80$ cells)] according to cell cycle scores. *Cdkn1c* mRNA expression was compared between these two populations, revealing a significantly higher expression in G1 compared to the other cell cycle phases. Statistical test: Wilcoxon test, $P = 0.00014$. Black dots: median; Horizontal bars: mean; box plot bounds: 1st and 3rd quartiles; whiskers: 1.5xIQR values from median.

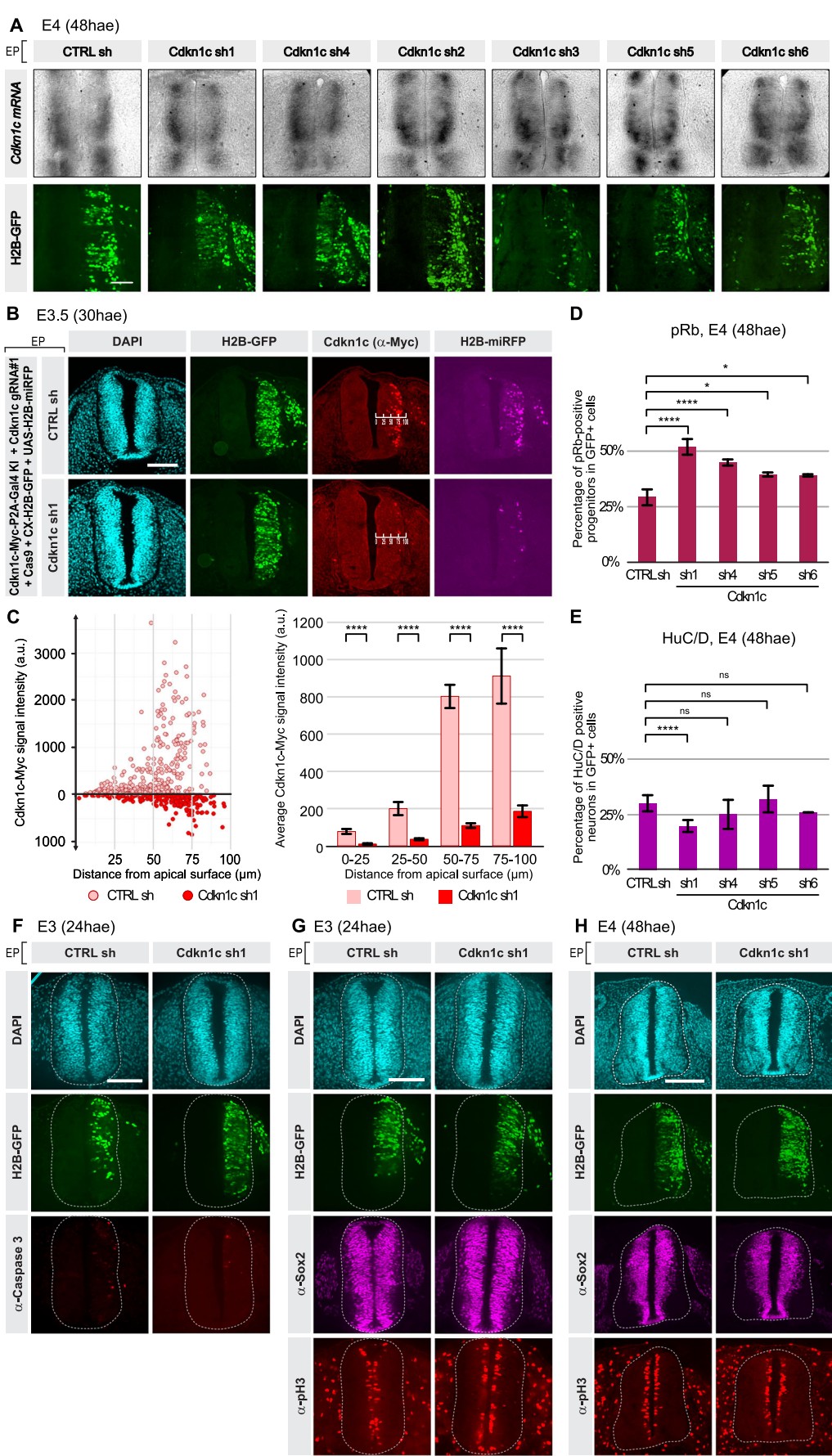

◀ **Figure EV4.  Partial knock down of *Cdkn1c* expression in spinal neural progenitors and prospective neurons via shRNA delays neurogenesis.**

(A) mRNA expression of *Cdkn1c* in chick embryonic neural tube after electroporation of each of the six shRNAs. In situ hybridization and GFP immunofluorescence on the same cryosection of thoracic region of chick embryo. Upper panel: visible downregulation of *Cdkn1c* mRNA was only observed with shRNA1 (sh1) and to a lesser extent shRNA4 (sh4) conditions, while comparable mRNA expression to the control condition (CTRL sh) was observed with the other shRNAs (sh2,3,5,6) (compare left versus right hemitubes). Lower panel: Corresponding level of electroporation for each embryo (GFP immunofluorescence). Scale bar: 50 μm. (B) Endogenous Cdkn1c-Myc protein expression pattern at E3,25 (30hae) using guide #1 to target the *Cdkn1c* locus upon downregulation of *Cdkn1c* via shRNA approach. Top row: Control sh, bottom row: Cdkn1c sh. Somatic knock-in of Myc tags at the *Cdkn1c* locus allows the visualization of Cdkn1c protein using an anti-Myc immunofluorescence (Myc, red) on E3,25 transverse vibratome sections. A H2B-GFP construct (green) was used as a control of electroporation. Cells with a knock-in event were visualized thanks to the coelectroporation of a UAS:H2B miRFP construct (magenta). Scale bar: 100 μm. The graduated scales in the third column illustrate the bins used for quantification in (C). (C) Quantification of Cdkn1c-Myc expression along the apico-basal axis upon downregulation of *Cdkn1c* via shRNA approach. Quantifications of the Cdkn1c-Myc signal intensity from knock-in insertions were performed in vibratome sections from embryos electroporated with a control shRNA or Cdkn1c shRNA1 (see representative examples in (B)). Left graph: the Myc signal intensity measured in individual cells is plotted on the Y-axis towards the upper (CTRL sh) and lower (Cdkn1c sh1) parts of the graph as a function of their position along the apico-basal axis (X-axis). Right graph: bars represent the average Myc signal intensity in 25μm-wide bins along the apico-basal axis. CTRL sh: 347 cells from 5 embryos; Cdkn1c sh1: 233 cells from 5 embryos. Error bars show means ± SD; ****$P < 0.0001$ (Kolmogorov-Smirnov test). (D) Distribution of pRb-positive progenitors (red) in Cdkn1c shRNAs 1, 4, 5, 6 or control shRNA conditions at E4, 48 hae. (values for CTRL sh and Cdkn1c sh1 are identical to Fig. 3E). Error bars show means ± SD. CTRL vs sh1 and CTRL vs sh4, ****$P < 0.0001$; CTRL vs sh5 and CTRL vs sh6, *$P = 0.0365$; (unpaired Student's t test relative to CTRL sh). Numbers of counted electroporated cells: sh4, 3111 cells from 8 embryos; sh5, 1552 from 3 embryos; sh6, 931 cells from 3 embryos were analyzed. (E) Distribution of the HuC/D-positive neurons (magenta) in shRNAs 1, 4, 5, 6 or control conditions at E4, 48 hae. (values for CTRL sh and Cdkn1c sh1 are identical to Fig. 3F). Error bars show means ± SD; CTRL vs sh1, ****$P < 0.0001$. All others: ns, $P > 0.05$; (unpaired Student's *t* test relative to CTRL sh). Numbers of counted electroporated cells: sh4, 3111 cells from 8 embryos; sh5, 1552 from 3 embryos; sh6, 931 cells from 3 embryos were analyzed. (F) Cdkn1c knock-down did not induce cell death. Transverse vibratome sections of embryos 24 hae with a CTRL sh (left column) or Cdkn1c sh1 (right column). Immunostaining with an anti-Caspase3 antibody (red, 3rd row) show a signal in a few cells on the electroporated side (H2B-GFP, 2nd row), which is similar in both conditions. Top row: DAPI staining. Scale bar: 100 μm. (G, H) Cdkn1c knock-down did not induce ectopic localization of progenitors in the mantle zone. Transverse vibratome sections of embryos 24 hae (E3, (E)) or 48 hae (E4, (F)) with a CTRL sh (left column) or Cdkn1c sh1 (right column). Electroporated cells are marked with H2B-GFP (green, 2nd row). Immunostaining with an anti-Sox2 antibody (magenta, 3rd row) was restricted to the ventricular zone, and no ectopic cells were observed in the mantle zone. Similarly, mitotic figures labeled with an anti-pH3 antibody (red, 4th row) were restricted to the apical surface. Scale bars: 100 μm. hae hours after electroporation, CTRL control, sh shRNA.

Experimental scheme for clonal analyses

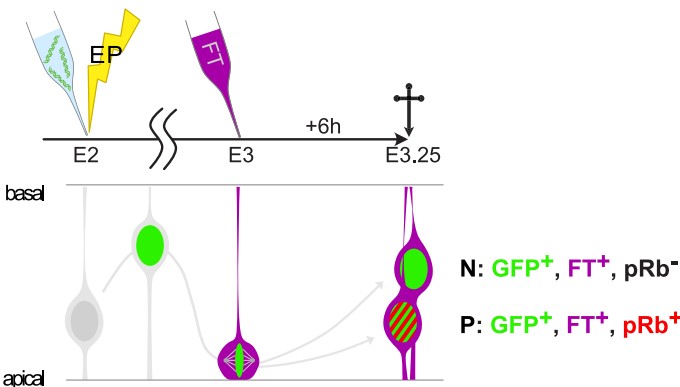

E3.25, 30hae, 6 hours after FT injection

| PP | PN | NN |
|---|---|---|

◀ **Figure EV5. FlashTag-based assay for the analysis of pairs of sister cells: principle and examples.**

Top: principle of the analysis of pairs of sister cells. Embryos are electroporated at HH13-14 (top left, yellow thunder) with Cdkn1c or control shRNA plasmids co-expressing a H2B-GFP reporter. Embryos are injected with the FlashTag dye (FT) 24 h after electroporation to label a synchronous cohort of mitotic progenitors, and collected 6 h later. At this time point after FT injection, anti-pRb immunofluorescence on thoracic vibratome sections determines the progenitor (pRb-positive) or neuron (pRb negative) status of electroporated (GFP-positive) pairs of FlashTag-positive sister cells (this panel is identical to Fig. 3G). Bottom: representative examples of two cell clones used for the quantification of modes of division depicted in Fig. 3H. Left to right panels show 3 examples each of PP pairs (column 1), PN pairs (column 2) and NN pairs (column 3). Arrows show pRb-positive (red) progenitors and asterisks show pRb negative neurons in FlashTag-positive (magenta) pairs of GFP-positive (green) sister cells. Scale bars: 25 µm.

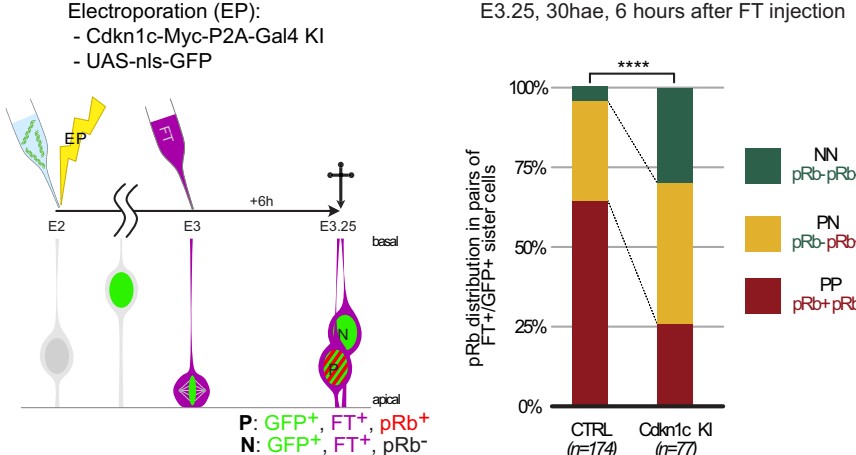

**Figure EV6.  Cdkn1c positive progenitors are more neurogenic than the overall progenitor population.**

Left: Experimental scheme for the clonal analysis. A knock-in of the Gal4 reporter in the *Cdkn1c* locus was performed via in ovo electroporation (yellow thunder) at E2 and FlashTag (FT) was injected 24 h later. Sister cells born from Cdkn1c-positive progenitors dividing at the time of FlashTag injection were identified on the basis of the expression of a UAS-nls-GFP reporter and FlashTag positivity. Right: The distribution of PP, PN, and NN pairs is significantly different between *Cdkn1c* knock-in progenitors and FT-positive pairs of sister cells on the contralateral side of the same transverse vibratome sections, indicating that the Cdkn1c-positive population of progenitors is more neurogenic than the whole population at that stage. Statistical test: Chi-square test, ****P < 0.0001; Chi-square value = 18.57. Pairs were obtained from 5 embryos. The number of pairs analyzed for both populations is indicated at the bottom of the graph.

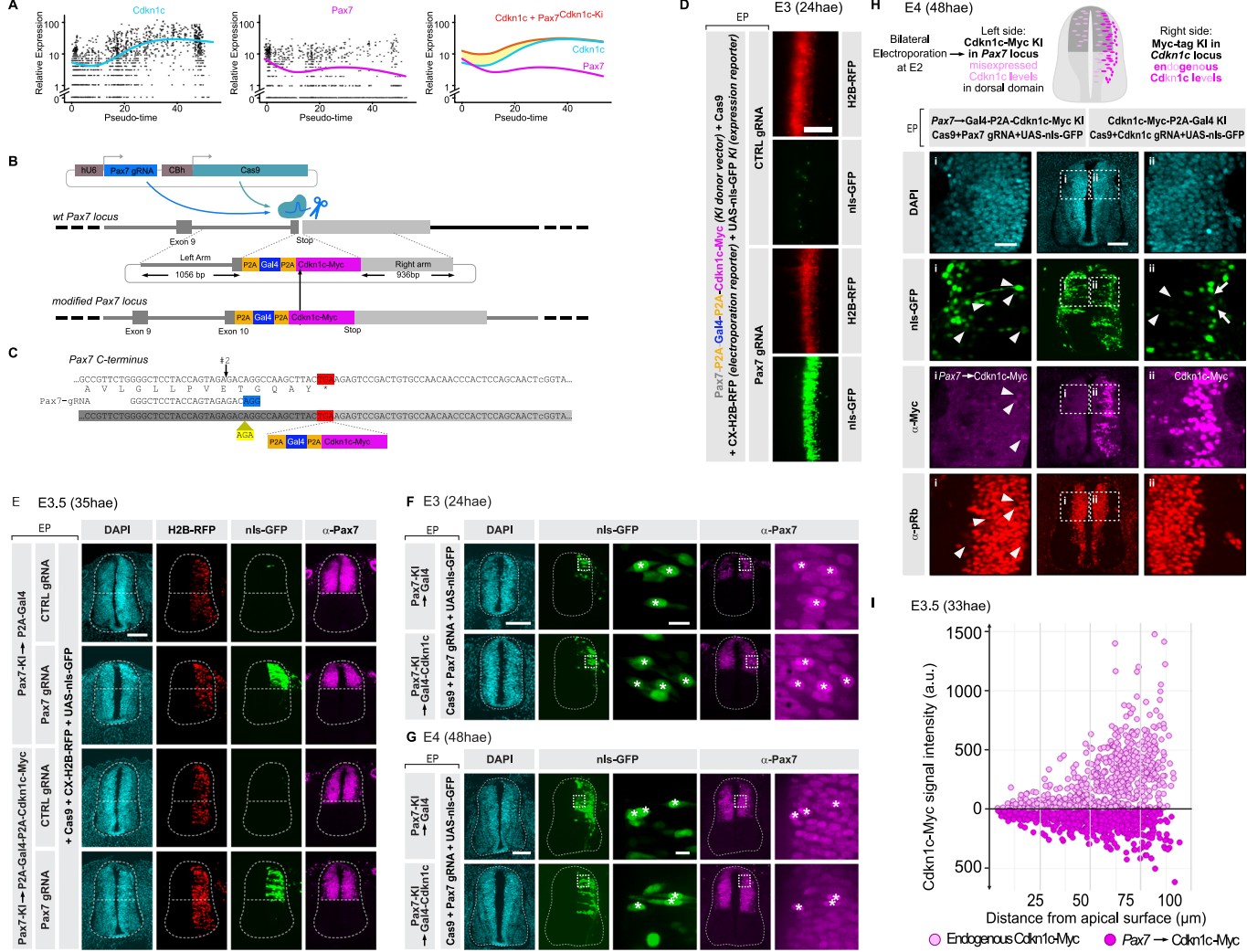

◀ **Figure EV7. *Cdkn1c* knock-in at the *Pax7* locus leads to a premature low-level expression of Cdkn1c in dorsal neural progenitors.**

(A) Expression levels of *Cdkn1c* and *Pax7* transcripts along the pseudo-time axis from the chick scRNASeq analysis. *Cdkn1c* expression has very low levels in "early" progenitors (left part of the pseudotime axis) and increases in more mature progenitors, before peaking in differentiating neurons. Pax 7 expression levels in "early" progenitors are slightly higher than those of *Cdkn1c*. The right panel shows an estimated cumulative expression (red) of endogenous (blue) and *Pax7*-driven (magenta) *Cdkn1c* levels upon knock-in of *Cdkn1c* coding sequences in the *Pax7* locus, which should result in premature expression in early progenitors, but no overexpression in newborn neurons. The scale for "Relative Expression" is logarithmic. (B) Principle of the "homology directed repair" (HDR) strategy used to drive low-level misexpression of *Cdkn1c*-Myc and Gal4-VP16 in dorsal progenitors from the *Pax7* locus. Somatic knock-in is based on a donor plasmid which carries long arms of homology (~1Kb) to the *Pax7* locus at the level of the C-terminus in exon 10. The arms of homology flank an "in-frame" knock-in cassette that consists in a P2A pseudocleavage site, a Gal4-VP16 synthetic transcription factor, a second P2A pseudocleavage site and the *Cdkn1c* coding sequence fused to three C-terminal Myc tags. This knock-in approach requires in addition the electroporation of a second vector expressing the Cas9 protein and a gRNA that targets the genomic region of *Pax7* upstream of the stop codon. Upon successful knock-in insertion, *Pax7* (gray), *Gal4-VP16* (blue) and *Cdkn1c-Myc* (magenta) coding sequences will be transcribed from the *Pax7* locus and co-translated. The insertion of P2A pseudocleavage sites (orange) between the three sequences will ensure that all three proteins are present as independent proteins. (C) Details of the targeted genomic sequence at the C-terminus of the *Pax7* locus. Genomic sequence at the level of *Pax7* C-terminus (top), sequence of the validated gRNA (Petit-Vargas et al, 2024) targeting this region (middle), and sequence of the arms of homology surrounding the knock-in cassette (bottom). To avoid possible targeting of the donor arms by the Pax7 gRNA, three bases (highlighted in yellow) were inserted 5 amino acids upstream of the *Pax7* stop codon. Note that this introduces an Arginine residue (AGA) in the Pax7 sequence. (D) Validation of the efficiency and specificity of the knock-in strategy. Imaging of the neural tube directly *in ovo*. The donor vector was co-electroporated together with a dual vector expressing the Cas9 nuclease and either a control gRNA (CTRL gRNA) or the gRNA targeting the *Pax7* locus. A UAS-nls-GFP vector was included in the electroporation mix to report expression of the Gal4-VP16 transcription factor. Finally, an electroporation reporter (CX-H2B-mRFP) was added to monitor the quality of electroporation. One representative embryo is shown for each condition, with similar electroporation level (red). Specificity is demonstrated by the virtual absence of background GFP signal when the control gRNA is used (compared to the massive GFP signal with Pax7 gRNA, only few GFP-positive cells are observed in the control embryo). Scale bar, 300 μm. (E) Transverse vibratome sections of embryos after knock-in of P2A-Gal4-VP16 somatic knock-in (top row) and P2A-Cdkn1c-P2A-Gal4-VP16 knock-in (bottom row) donor constructs at the *Pax7* locus at E3.5 (35hae). The UAS-nls-GFP signal (green) reports knock-in events and expression of Gal4-VP16 from the *Pax7* locus, and accordingly is restricted to the Pax7-domain (α-Pax7, magenta) in the dorsal half of the neural tube, and only in the presence of the Pax7-specific gRNA, despite electroporation all along the dorso-ventral axis, as revealed by the electroporation reporter H2B-RFP. The dashed line outlines the neural tube and the ventral limit of the Pax7 expression domain. Scale bar: 50 μm. (F, G) Characterization of Pax7 protein expression after knock-in at the *Pax7* locus. Immunostainings of Pax7 protein after somatic Knock-in of P2A-Gal4-VP16 (top row) or P2A-Gal4-VP16-P2A-Cdkn1c (bottom row) coding sequences at the *Pax7* locus on transverse vibratome sections of embryos 24 hae (E3, (A)) and 48 hae (E4, (B)). Knock-in events are identified via the co-electroporated UAS-nls-GFP reporter (green, second and third columns). First row: DAPI staining. Third and fifth rows of the panel show a close-up of the region highlighted by a dashed rectangle in the second and fourth row. A similar modest increase in Pax7 immunofluorescence is observed in knock-in cells (green, asterisks) in both conditions (compare top and bottom rows). Since this increase occurs irrespective of whether Cdkn1c is overexpressed or not, it does not affect the interpretation of phenotypes comparing the two conditions. The contour of the neural tube is underlined by a dashed line. Scale bars: 100 μm for first, second and fourth rows, 10 μm for close-ups. (H) *Pax7*-driven exogenous expression of Cdkn1c in dorsal progenitors mimics the levels of Cdkn1c expression in neurogenic progenitors at E4 (48 hae). A bilateral electroporation scheme was used to compare *Pax7*-driven levels of Cdkn1c expression (electroporation 1, left side hemi-tube, knock-in of Cdkn1c-Myc in the *Pax7* locus) with endogenous Cdkn1c levels (electroporation 2, right side hemi-tube, knock-in of a Myc tag in the *Cdkn1c* locus). The level of Cdkn1c-Myc (magenta) expression driven by *Pax7* is low and restricted to the ventricular region, where it is comparable to the endogenous levels of Cdkn1c-Myc expression in the contralateral side (i and ii, arrowheads). Note that although very few cells with a detectable Cdkn1c-Myc expression are observed in the misexpressed condition, the UAS-nls-GFP reporter is widely expressed, indicating strong electroporation and knock-in efficiency. This strong GFP signal, compared to the weak Myc signal, is explained by a differential stability and posttranslational regulation between Cdkn1c-Myc and Gal4-VP16, and by amplification of the GFP fluorescence via the Gal4/UAS system. Scale bars, 100 μm in central column and 30 μm in close-ups. (I) Quantification of endogenous versus *Pax7*-driven Cdkn1c-Myc expression along the apico-basal axis. The same vector combinations used in (F, G) were electroporated in separate sets of embryos to generate either a knock-in of the Myc reporter in the Cdkn1c locus reporting the endogenous expression level, or a knock-in of the Cdkn1c-Myc fusion in the *Pax7* locus, reporting the misexpression level. Quantifications of the Cdkn1c-Myc signal intensity from both conditions were performed 33 h after electroporation in vibratome sections. The Myc signal intensity measured in individual cells is plotted on the Y-axis towards the upper (endogenous Cdkn1c-Myc, light plum) and lower (*Pax7*-driven misexpressed Cdkn1c-Myc, magenta) parts of the graph as a function of their position along the apico-basal axis (X-axis). Endogenous Cdkn1c-Myc: 697 cells from 5 embryos; *Pax7*-driven Cdkn1c-Myc: 761 cells from 4 embryos. hae: hours after electroporation.

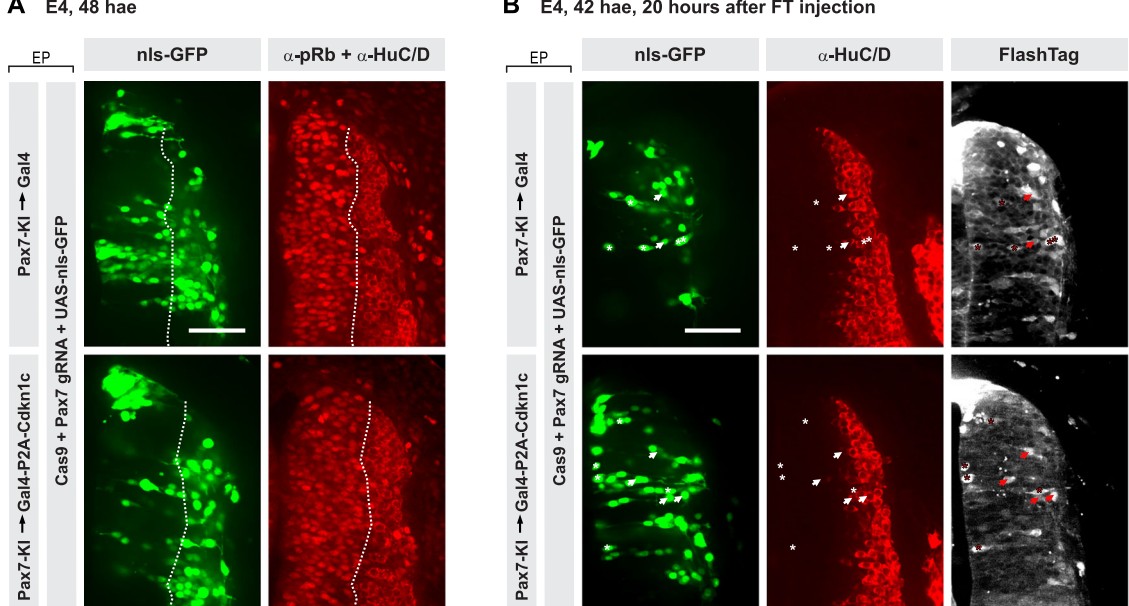

**Figure EV8. Premature low-level expression of Cdkn1c in dorsal neural progenitors leads to accelerated neurogenesis.**

(A) Transverse vibratome sections of embryos 48 h after electroporation of P2A-Gal4-VP16 somatic knock-in (top row) or P2A-Gal4-VP16-P2A-Cdkn1c knock-in (bottom row) constructs at *Pax7* locus. pRb and HuC/D immunofluorescence can be used to identify the nature of the cells (progenitor, pRb-positive nuclear signal in the ventricular zone or neuron, HuC/D-positive cytoplasmic signal in the mantle zone) that have undergone a knock-in event (UAS-nls-GFP, green signal). The dashed line marks the ventricular zone. Scale bar: 50 µm. (B) Transverse vibratome sections of embryos 42 h after electroporation of P2A-Gal4-VP16 somatic knock-in (top row) or P2A-Cdkn1c-P2A-Gal4-VP16 knock-in (bottom row) constructs at *Pax7* locus. FlashTag (gray) was injected 22 h after electroporation to label a cohort of mitotic cells, and embryos were collected 20 h later. HuC/D immunofluorescence (red) was used to identify neurons. Arrows point to triple positive (FlashTag +, GFP +, HuC/D +) cells and asterisks highlight double-positive (FlashTag +, GFP +, HuC/D−) cells. Scale bar: 50 µm. hae hours after electroporation.

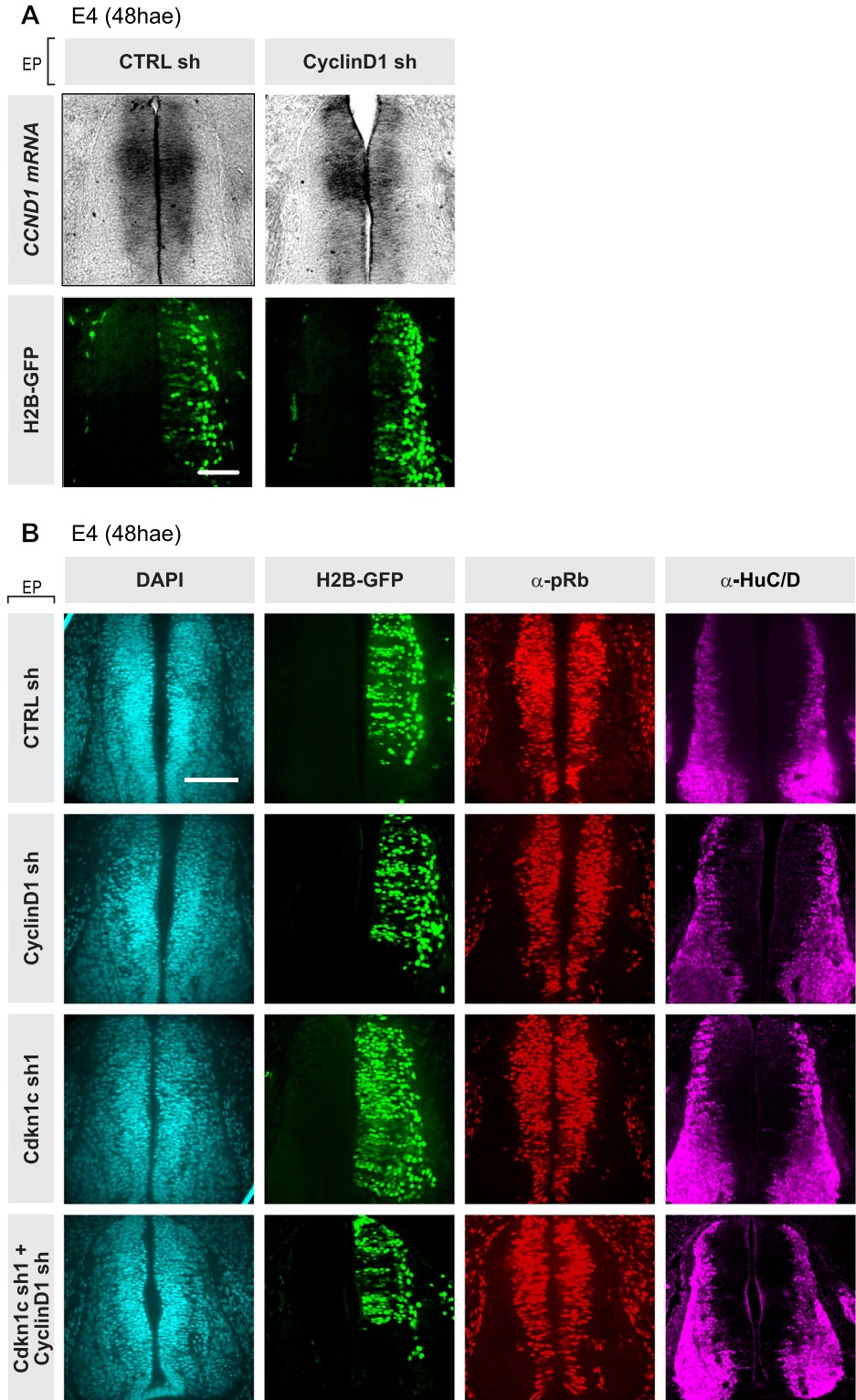

**Figure EV9.  Knock-down of *CyclinD1* mRNA rescues the Cdkn1c anti-neurogenic phenotype in the chick embryonic neural tube.**

(A) In situ hybridization of a *CyclinD1* antisense probe on transverse cryosections of chick embryonic neural tube followed by an anti-GFP immunostaining to reveal electroporated cells two days (E4, 48 hae) after CyclinD1 shRNA (CyclinD1sh) electroporation. Scale bars: 100 μm. (B) Transverse vibratome sections of the chick neural tube (thoracic level) at E4 (HH st22) stained with HuC/D antibody (magenta) to label neurons and anti-pRb (red) antibody to label progenitors in control, single CyclinD1, single Cdkn1c or double CyclinD1/Cdkn1c shRNAs (sh) conditions. Representative images related to the quantification of the percentage of pRb-positive and HuC/D-positive cells within the electroporated population (GFP-positive) shown in Fig. 6D,E. Scale bar: 100 μm. hae: hours after electroporation.

