## [Peer Review File · EMBO Reports]

A low-level CDKN1c/p57kip2 expression drives spinal progenitors towards neurogenic modes of division

Baptiste Mida, Nathalie Lehmann, Rosette Goïame, Fanny Coulpier, Kamal Bouhali, Isabelle Barbosa, Hervé le Hir, Morgane Thomas-Chollier, Evelyne Fischer, and Xavier Morin

Corresponding author(s): *Xavier Morin (xmorin@biologie.ens.fr)*, *Evelyne Fischer (evelyne.fischer@bio.ens.psl.eu)*

Review Timeline:

Transfer Date:	14th Apr 25
Editorial Decision:	24th Apr 25
Revision Received:	15th Sep 25
Editorial Decision:	17th Oct 25
Revision Received:	28th Oct 25
Accepted:	13th Nov 25

Editor: *Esther Schnapp*

**Transaction Report: This manuscript was transferred to
EMBO reports following peer review at Review Commons.**

**Review
COMMONS**

Review #1

1. Evidence, reproducibility and clarity:

Evidence, reproducibility and clarity (Required)

Summary

This study utilizes the developing chicken neural tube to assess the regulation of the balance between proliferative and neurogenic divisions in the vertebrate CNS. Using single-cell RNAseq and endogenous protein tagging, the authors identify Cdkn1c as a potential regulator of the transition towards neurogenic divisions. Cdkn1c knockdown and overexpression experiments suggest that low Cdkn1c expression enhances neurogenic divisions. Using a combination of clonal analysis and sequential knockdown, the authors find that Cdkn1c lengthens the G1 phase of the cell cycle via inhibition of cyclinD1. This study represents a significant advance in understanding how cells can transition between proliferative and asymmetric modes of division, the complex and varying roles of cycle regulators, and provides technical advance through innovative combination of existing tools.

Major and Minor Comments:

Overall

- Sample numbers are missing or unclear throughout for all imaging experiments. The authors should add numbers of cells analysed and/or numbers of embryos for their results to be appropriately convincing.
- Values and error bars on graphs must be defined throughout. Are the values means and error bars SD or SEM?

Results 2

- A reference should be provided for cell type distribution in spinal neural tube, where the authors state that cell bodies of progenitors reside within the ventricular zone.
- The authors state that Cdkn1c "was expressed at low levels in a salt and pepper fashion in the ventricular zone, where the cell bodies of neural progenitors reside, and markedly increased in a domain immediately adjacent to this zone which is enriched in nascent neurons on their way to the mantle zone. In contrast, the transcript was completely

excluded from the mantle zone, where HuC/D positive mature neurons accumulate." It is not clear if this is referring only to E4 or also to E3 embryos. Indeed, Cdkn1c expression appears to be much more salt and pepper at E3 and only resolves into a clear domain of high expression adjacent to the mantle zone at E4. It may be helpful if this expression pattern could be described in a bit more detail highlighting the changes that occur between E3 and E4.

- It would be useful to annotate the ISH images in Fig 2A to show the ventricular and mantle zones as defined by immunofluorescence.
- Reference should be included for pRb expression dynamics.
- Could the Myc tag insertion approach disrupt protein function or turnover?
- Why was the insertion target site at the C terminus chosen?
- OPTIONAL Could a similar approach be used to tag Cdkn1c with a fluorescent protein to enable live imaging of dynamics?
- In suppl Fig 1C nlsGFP-positive cells are shown in the control shRNA condition. How can this be explained and does it impact the interpretation of the findings?
- In Fig 2B, there are a number of Myc labelled cells in the mantle zone, whereas the in situ images show no appreciable transcript expression. Is this because the protein but not the transcript is present in these cells? Could the authors comment on this?

Results 3

- It should be mentioned how mRNA expression levels were quantified in the shRNA validation experiment (supp Fig 2A).
- Figure panels are not currently cited in order. Citation or figure order could be changed.
- The authors should provide representative images for the graphs shown in Fig 3A and 3B. These could go into supplementary if the authors prefer.
- A supplementary figure showing the Caspase3 experiment should be added.
- OPTIONAL. Identification of sister cells in the clonal analysis experiments is based on static images and cannot be guaranteed. Could live imaging be used to watch divisions followed by fixation and immunostaining to confirm identity?

Results 4

- How did the authors quantify the intensity of endogenous Myc-tagged Cdkn1c to confirm the validity of the Pax7 locus knock in? Can they show that the expression level was consistently lower than the endogenous expression in neurons? Quantification and sample numbers should be shown.
- In Fig 4B, the brightness of row 2 column 1 is lower than the same image in row 2 column

2, which is slightly misleading, since it makes the misexpressed expression level look lower than it is compared with endogenous in column 3. Is this because only a single z-section is being displayed in the zoomed in image? If so, this should be stated in the figure legend.

- In Fig 4D, the increase in neurogenic divisions is mainly because of the rise in terminal NN divisions according to the graph, but no clear increase in PN divisions. Could the authors comment on the significance of this?

Results 5

- The proportion of pRb-positive progenitors having entered S phase was stated to be higher at all time points; however, it is not significantly higher until 6h30 and is actually trending lower at 2h30.

- OPTIONAL Could CyclinD1 activity be directly assessed?

General

- Scale bars missing fig s1c s4d.

- OPTIONAL Some of the main findings be replicated in another species, for example, mouse or human to examine whether the mechanism is conserved.

- OPTIONAL Could use approaches other than image analysis be used to reinforce findings, for example biochemical methods, RNAseq or FACS?

- A model cartoon to summarise outcomes would be useful.

- Unclear how cells were determined to be positive or negative for a label. Was this decided by eye? If so, how did the authors ensure that this was unbiased?

2. Significance:

Significance (Required)

Strengths:

This manuscript investigates the mechanisms regulating the switch from symmetric proliferative divisions to neurogenic division during vertebrate neuronal differentiation. This is a question of fundamental importance, the answer to which has eluded us so far. As such, the findings presented here are of significant value to the neurogenesis community and will be of broad interest to those interested in cell divisions and asymmetric cell fate acquisition. Specific strengths include:

- Variety of approaches used to manipulate and observe individual cell behaviour within a

physiological context.

- A limitation of using the chicken embryo is the lack of available antibodies for immunostaining. The authors take advantage of recent advances in chicken embryo CRISPR strategy to endogenously tag the target protein with Myc, to facilitate immunostaining.
- Innovative combination of genetic and labelling tools to target cells, for example, use of FlashTag and EdU in combination to more accurately assess G1 length than the more commonly used method.
- Premature misexpression demonstrates that the previously observed dynamics indeed regulate cell fate.
- Mechanistic insight by examining downstream target CyclinD1.
- Clearly presented with useful illustrations throughout.
- Logic is clear and examination thorough.
- Conclusions are warranted on the basis of their findings.

Limitations

- This study primarily used visual analysis of fixed tissue images to assess the main outcomes. To reinforce the conclusions, these could be supplemented with live imaging to appreciate dynamics, or biochemical techniques to look at protein expression levels.
- Some aspects of quantification require explanation in order for the experiments to be replicated.
- It is imperative that precise sample sizes are included for all experiments presented.

Advance:

- First functional demonstration role for Cdkn1c in regulating neurogenic transition in progenitors.
- Conceptual advance suggesting Cdkn1c has dual roles in driving neurogenesis: promoting neurogenic divisions of progenitors and the established role of mediating cell cycle exit previously reported.
- Technical advances in the form of G1 signposting and endogenous Myc tagging using CRISPR in chicken embryonic tissue.

Audience:

Of broad interest to developmental biologists. Could be relevant to cancer, since Cdkn1c is implicated.

♣ Please define your field of expertise with a few keywords to help the authors contextualize your point

Developmental biology, vertebrate embryonic development, neuronal differentiation, imaging. Please note that we have not commented on RNAseq experiments as these are outside of our area of expertise.

3. How much time do you estimate the authors will need to complete the suggested revisions:

Estimated time to Complete Revisions (Required)

(Decision Recommendation)

Less than 1 month

4. Review Commons values the work of reviewers and encourages them to get credit for their work. Select 'Yes' below to register your reviewing activity at Web of Science Reviewer Recognition Service (formerly Publons); note that the content of your review will not be visible on Web of Science.

Yes

Review #2

1. Evidence, reproducibility and clarity:

Evidence, reproducibility and clarity (Required)

The work by Mida and colleagues addresses important questions about neurogenesis in the embryo, using the chicken neural tube as their model system. The authors investigate the mechanisms involved in the transition from stem cell self-renewal to neurogenic progenitor divisions, using a combination of single cell, gene functional and tracing studies.

The authors generated a new single cell data set from the embryonic chicken spinal cord and identify a transitory cell population undergoing neuronal differentiation, which expresses Tis21, Neurog2 and Cdkn1c amongst other genes. They then study the role of

Cdkn1c and investigate the hypothesis that it plays a dual role in spinal cord neurogenesis: low levels favour transition from proliferative to neurogenic divisions and high levels drive cell cycle exit and neuronal differentiation.

****Major comments****

I have only a general comment related to the main point of the paper. The authors claim that Cdkn1c onset in cycling progenitor drives transition towards neurogenic modes of division, which is different from its role in cell cycle exit and differentiation. Figures 3F and 4D are key figures where the authors analysed PP, PN and NN mode of divisions via flash tag followed by analysis of sister cell fate. If their assumption is correct, shouldn't they also see, for example in Fig. 4D, an increase in PN or is this too transient to be observed or is it bypassed? At the moment, the calculations of PN and NN frequencies are merged in the text, so perhaps describing PN and NN numbers separately will help better understand the dynamics of this gradual process (especially since there is little to no difference in PN). Could the increase in NN be compatible also with a role in cell cycle exit and differentiation, for example from cells that have been targeted and are still undergoing the last division (hence marked by flash tag) or there won't be any GFP cells marked by flash tag a day after expression of high levels of Cdkn1c? Basically, what would the effect of expressing higher levels of Cdkn1c be? I guess this will really help them distinguish between transition to neurogenic division rather than neuronal differentiation. If not experimentally, any further comments on this would be appreciated.

****Minor comments****

Fig 3C my understanding is that HuC/D should be nuclear, but in fig 3C it seems more cytoplasmic (any comment?)

Fig Suppl 3E (and related 4B), immuno for Cdkn1c-Myc: to help the reader understand the difference between the immuno signals when looking at the figure, I would suggest writing on the panel i) Pax7-Cdkn1c-Myc and ii) endogenous Cdkn1c-Myc, rather than 'misexpressed' and 'endogenous', which is slightly confusing (especially because what it is called endogenous expression is higher).

Literature citing: Introduction and discussion are very nicely written, although they could benefit from some more recent literature on the topic. For example, Cdkn1c role as a gatekeeper of stem cell reserve in the stomach, gut, (Lee et al, CellStemCell 2022 PMID: 35523142) or some other work on symmetric/asymmetric divisions and clonal analysis in

zebrafish (Hevia et al, CellRep 2022 PMID: 35675784, Alexandre et al, NatNeur PMID: 20453852), mammals (Royal et al, Elife 2023 37882444, Appiah et al, EMBO rep 2023 PMID: 37382163). Also, similar work has been performed in the developing pancreatic epithelium, where mild expression of Cdkn1a under Sox9rtTa control was used to lengthen G1 without overt cell cycle exit and this resulted in Neurog3 stabilization and priming for endocrine differentiation (Krentz et al, DevCell 2017 PMID: 28441528), so similar mechanisms might be in place to gradually shift progenitor towards stable decision to differentiate.

Moreover, in the discussion, alongside Neurog2 control of Cdkn1c, it could be mentioned that the feedback loop between Cdk inhibitors and neurogenic factor is usually established via Cdk inhibitor-mediated inhibition of proneural bHLHs phosphorylation by CDKs (Krentz et al, DevCell 2017 PMID: 28441528, Ali et al, 24821983, Azzarelli et al 2017 - PMID: 28457793; 2024 - PMID:39575884).

Further, in the discussion, could they mention anything about the following open questions: is there evidence for Cdkn1c low/high expression in mammalian spinal cord? Or maybe of other Cdk inhibitors? Is Cdkn1c also involved in cell cycle exit during gliogenesis or is there another Cdk inhibitor expressed at later developmental stages, hence linking this with specific cell fate decisions?

2. Significance:

Significance (Required)

The work here presented has important implications on neural development and its disorders. The authors used the most advanced technologies to perform gene functional studies, such as CRISPR-HDR insertion of Myc-tag to follow endogenous expression, or expression under endogenous Pax7 promoter, often followed by flash tag experiments to trace sister cell fate, and all of this in an in vivo system. They then tested cell cycle parameters, clonal behaviour and modes of cell division in a very accurate way. Overall data are convincing and beautifully presented. The limitation is potentially in the resolution between the events of switching to neurogenic division versus neuronal differentiation, which might just warrant further discussion.

This work advances our knowledge on vertebrate neurogenesis, by investigating a key player in proliferation and differentiation.

I believe this work will be of general interest to developmental and cellular biologists in different fields. Because it addresses fundamental questions about the coordination between cell cycle and differentiation and fate decision making, some basic concepts can be translated to other tissues and other species, thus increasing the potential interested

audience.

My work focuses on stem cell fate decisions in mammalian systems, and I am familiar with the molecular underpinnings of the work here presented. However, I am not an expert in the chicken spinal cord as a model and yet the manuscript was interesting. I am also not sufficiently expert in the bioinformatic analysis, so cannot comment on the technical aspects of Figure 1 and the way they decided to annotate their data.

3. How much time do you estimate the authors will need to complete the suggested revisions:

Estimated time to Complete Revisions (Required)

(Decision Recommendation)

Between 1 and 3 months

Yes

Review #3

1. Evidence, reproducibility and clarity:

Evidence, reproducibility and clarity (Required)

****Summary:****

In this study, Mida et al. analyze large-scale single-cell RNA-seq data from the chick embryonic neural tube and identify Cdkn1c as a key molecular regulator of the transition from proliferative to neurogenic cell divisions, marking the onset of neurogenesis in the developing CNS. To confirm this hypothesis, they employed classical techniques, including the quantification of neural cell-specific markers combined with the flashTAG label, to track and isolate isochronic cohorts of newborn cells in different division modes. Their

findings reveal that Cdkn1c expression begins at low levels in neurogenic progenitors and becomes highly expressed in nascent neurons.

Using a classical knockdown strategy based on short hairpin RNA (shRNA) interference, they demonstrate that Cdkn1c suppression promotes proliferative divisions, reducing neuron formation. Conversely, novel genetic manipulation techniques inducing low-level CDKN1c misexpression drive progenitors into neurogenic divisions prematurely.

By employing cumulative EdU incorporation assays and shRNA-based loss-of-function approaches, Mida et al. further show that Cdkn1c extends the G1 phase by inhibiting cyclin D, ultimately concluding that Cdkn1c plays a dual role: first facilitating the transition of progenitors into neurogenic divisions at low expression levels, and later promoting cell cycle exit to ensure proper neural development.

This study presents several ambiguities and lacks precision in its analytical methodologies and quantification approaches, which contribute to confusion and potential bias. To enhance the reliability of the conclusions, a more rigorous validation of the methods employed is essential.

This study introduces a novel approach to tracking the fate of sister cells from neural progenitor divisions to infer the division modes. While previous methods for analyzing the division mode of neural progenitor cells have been implemented, rigorous validation of the approach introduced by Mida et al. is necessary. Furthermore, the concept of cell cycle regulators interacting to control the duration of specific cell cycle stages and influencing progenitor cell division modes has been explored before, potentially limiting the novelty of these findings.

****Majors comments:****

1. The study presents ambiguity and lacks precision in quantifying neural precursor division modes. The authors use phosphorylated retinoblastoma protein (pRb) as a marker for neurogenic progenitors, claiming its reliability in identifying neurogenic divisions. However, they do not provide a thorough characterization of pRb expression in the developing chick neural tube, leaving its suitability as a neurogenic division marker unverified. Furthermore, retinoblastoma protein (Rb) and cyclin D interact crucially to regulate the G1/S phase transition of the cell cycle, with cyclin D/CDK complexes phosphorylating Rb. Since the authors conclude that CDKN1c primarily acts by inhibiting the cyclin D/CDK6 complex, it is likely that CDKN1c influences pRb expression or phosphorylation state. This raises the possibility that pRb could be a direct target of CDKN1c, whose expression and

phosphorylation would be altered in gain-of-function (GOF) and loss-of-function (LOF) analyses of CDKN1c. In light of this, it would be more appropriate to consider pRb as a CDKN1c target and discuss the molecular mechanisms regulating cell cycle components. A more precise approach would involve using other markers or targets to quantify neural precursor division modes at earlier stages of neurogenesis.

2. Furthermore, the study employs FlashTag labeling to track daughter cells post-division, but the 16-hour post-injection window may result in misidentification of sister cells due to the potential presence of FlashTagged cells that did not originate from the same division. This introduces a risk of bias in quantification, data misinterpretation, and potential errors in defining division modes. A more rigorous validation of the FlashTag strategy and its specificity in tracking division pairs is necessary to ensure the reliability of their conclusions.

3. The knock-in strategy used to tag the endogenous CDKN1c protein in Figure 2 is an elegant tool to infer protein dynamics in vivo. However, since strong conclusions regarding CDKN1c dynamics during the cell cycle are drawn from this section, it would be advisable to strengthen the results by including quantification with adequate replication and proper statistical analysis, as the current findings are preliminary and somewhat speculative.

- "Although pRb is specific for cycling cells, it is only detected once cells have passed the point of restriction during the G1 phase." Please provide literary reference confirming this observation. Given that pRb immunoreactivity is used as a marker for cycling progenitors to base many of the results of this study, it would be very valuable to characterize the dynamics of pRb in cycling cells in the studied tissue, for instance combined with the cell cycle reporter used by Molina et al. (Development 2022).

- The characterization of dynamics is performed only with one of the gRNAs (#1) on the basis that it produces the strongest NLS-GFP signal, as a proxy for guide efficiency. It would be nice if the authors could validate guide cutting efficiency via sequencing (e.g. using a Cas9-T2A-GFP plasmid and sorting for positive cells).

- In order to make sure that the dynamics inferred from Myc-tag immunoreactivity do reflect the cell cycle dynamics of CDKN1c-myc, it would be advisable to confirm in-frame insertion of the myc-tag sequence.

- It would be valuable to analyse the dynamics of Myc immunoreactivity in combination of pRb in all three gRNAs (highlighted in Supplementary Figure 1), as it would be a strong point in favour that the dynamics reflect the endogenous CDKN1c dynamics.

- It would be very valuable to provide a quantification of said dynamics (e.g. plotting myc intensity / pRb immunoreactivity along the apicobasal axis of the tissue).

4. In Figure 3, the authors use a short-hairpin-mediated knock-down strategy to decrease the levels of Cdkn1c, and show that this manipulation leads to an increase percentage of cycling progenitors and a decrease in the number of neurons in electroporated cells.

The authors claim that their shRNA-based knockdown strategy aims to reduce low-level Cdkn1c expression in neurogenic progenitors while minimally affecting the higher expression in newborn neurons required for cell cycle exit. However, several factors need consideration. Electroporation introduces variability in shRNA delivery, making it difficult to achieve consistent gene inhibition across all cells, especially for dose-dependent genes like Cdkn1c. Additionally, Cdkn1c generates multiple isoforms, which may not be fully annotated in the chick genome, raising the possibility that the shRNA targets specific isoforms, potentially explaining the observed low expression. A more rigorous approach, such as qPCR analysis of sorted electroporated cells, would better validate the expression levels, rather than relying on in situ hybridization, presenting electroporated and non-electroporated cells in the same section (Supp. Figure 2).

- As the authors note, "Unambiguous identification of cycling progenitors and postmitotic neurons is notoriously difficult in the chick spinal cord". "markers of progenitors usually either do not label all the phases of the cell cycle (eg. Phospho-Rb, thereafter pRb), or persist transiently in newborn neurons (eg. Sox2)." Given that pRb immunoreactivity is used as the basis for a lot of the conclusions in this study, it would be valuable to add a characterization of its dynamics as mentioned in Figure 2, as well as provide literary references/proof that Sox2 expression persists in newborn neurons.

- The undefined population (pRb-/HuCD-) introduces an unknown that assumes that the percentage of progenitors in G1 phase before the restriction point and the number of newborn neurons are equal for both conditions in an experiment. Can the authors provide explanation for this assumption?

- In Gui et al. (Dev Biol 2006), authors showed that a knockdown of Cdkn1c leads to a failure of nascent neurons to exit the cell cycle and causes them to re-entry the cell cycle, shown by ectopic mitoses. In that study, cells born from those ectopic mitoses eventually leave the cell cycle leading to an increase in the number of neurons. Can the authors check for ectopic mitoses at 24hpe and 48hpe?

- The authors then address the question of whether the decrease in neuron number is due to the failure of newborn neurons to exit the cell cycle or to a delay in the transition from proliferative to neurogenic divisions. For that, they implement a strategy to label a synchronized cohort of progenitors based of incorporation of a FlashTag dye. - Given that this strategy is the basis of many of the experiments in this article, it would be very valuable to expand on the validation of this technique as cited in major comment #2. In figure 3E, the close proximity of cell pairs in PP and PN clones shown in the pictures makes their sibling status apparent. However, this is not the case for the NN clone. Can the authors further explain with what criteria they determined the clonal status of two FlashTag labelled cells? Can they provide further image examples of different types of clones?

- Can the authors show that the plateau reached in Sup Figure 3 for pRb immunoreactivity corresponds to a similar dynamic for HuC/D immunoreactivity?

- In order to further validate the strategy, could the authors use it at different stages to validate if they can replicate the different percentages of PP/PN/NN reported in the literature (e.g. Saade Cell Rep 2013)?

5. In Figure 4, the strategy used to induce a low-dose overexpression of CDKN1c is an elegant method to introduce CDKN1c-Myc expression under the control of the endogenous Pax7 promoter, active in proliferative progenitors. The main point to address is:

- Please provide proof that Pax7 expression is not altered in guides with a successful knock-in event (e.g. sorting and WB against the Pax7 protein) or the immunohistochemistry as performed in the Pax7-P2A-Gal4 tagging in Petit-Vargas et al., 2024.

- Given the cell cycle regulated expression and activity of CDKN1c, can the authors elaborate on whether this is regulated at the promoter level? If so, how does this differ from the promoter activity of Pax7?

- It would be advisable to characterize the dynamics along the cell cycle for the overexpressed form of CDKN1c-Myc relative to pRb, similarly to what was done in Figure 2B.

6. In figure 5, the authors use a double knock-down strategy to test the hypothesis that the effect of Cdkn1c in G1 length is partially at least through its inhibition of CyclinD1. Results show that double shRNA-mediated knock-down of CyclinD1 and Cdkn1c counteracts the effects of Cdkn1c-sh alone on EdU incorporation, PP/PN/NN cell divisions and overall ratios of progenitors and neurons.

- In the measurement of progenitor cell cycle length in Figure 5A, it would be more appropriate to present the nonlinear regression method described by Nowakowski et al. (1989), as has been commonly used in the field (Saade et al., 2013, PMID: 23891002, Le Dreau et al., 2014, PMID: 24515346, Arai et al., 2011, PMID: 21224845).

- Cumulative EdU incorporation in spinal progenitors (pRb-positive) at E3 (24 hours after injection) showed that the proportion of EdU-positive progenitors reached a plateau at 14 hours in control conditions, which is later than what has been reported in Le Dreau et al., 2014 (PMID: 24515346). Can you explain why?

- It would be interesting to measure G1 length as in Figure 5D for the double cdkn1c-sh - ccnd1-sh knock down condition, to see if it rescues G1 length. As well as in the Ccnd1 knock down condition alone to see if it increases G1 length in this context as well.

Minor comments

Introduction:

- The introduction should include references of studies of the role of Cdkn1c in cortical development (Imaizumi et al. Sci Rep 2020, Colasante et al. Cereb Cortex 2015, Laukoter et al. Nature Communications 2020).

1. Transcriptional signature of the neurogenic transition (Figure 1).

- In the result section, it would be informative to include the genes used to determine the progenitor and neuron score (instead of in Methods).

- Figure 1A. It would be informative to add in the diagram what "filtering" means (eg. Neural crest cells).

- In the result section, "However, while Tis21 expression is switched off in neurons, Cdkn1c transiently peaks at high levels in nascent neurons before fading off in more mature cells." Missing literary reference or data to clearly demonstrate this point.

- "Interestingly, the gene cluster that contained Tis21 also contained genes encoding proteins with known expression and/or functions at the transition from proliferation to differentiation, such as the Notch ligand Dll1, the bHLH transcription factors Hes6, NeuroG1 and NeuroG2, and the coactivator Gadd45g." Missing references.

- There is an error in the color code in Cell Clusters in Figure 1C (cluster 4 yellow in the legend but ocre in the figure)

It would be valuable to assign cell cycle stage to neural progenitor cells (based on cell cycle score) and determine whether cdkn1c at the transcript level also shows enrichment in G1 cells considered to be progenitors.

2. Progressive increase in Cdkn1c/p57kip2 expression underlie different cellular states in the embryonic spinal neural tube (Figure 2).

- Figure 2A. Scale bar is missing in E3 and E4. It is important to consider the growth of the developing spinal cord and present it accordingly (E3 transverse section, Figure 2).

- Figure 2 could use a diagram of the knock-in strategy used, similar as the one in Figure 4A.

- Indicate hours post-electroporation. Indicate which guide is used in the main text.

3. Downregulation of Cdkn1c in neural progenitors delays the transition from proliferative to neurogenic modes of division (Figure 3).

- In methods: "Thus, to reason on a more homogeneous progenitor population, we restricted all our analysis to the dorsal one half or two thirds of the neural tube." Indicate when and depending on what one half or two thirds of the neural tube were analysed.

- Figure 3. Would have a better flow if 3C preceded 3A and 3B.

- Figure 3C. it would be informative to show pictures of the electroporated NT at both 24hpe

and 48hpe, as well as highlighting the dorsal part of the neural tube that was used for quantification.

- Are the clonal analysis experiments (Fig 3D, E and F) also restricted to the dorsal region?

- Figure Sup3B colour code is switched (green for PP and red for NN) compared to the rest of the paper.

- In methods "At each measured timepoint (1h, 4h, 7h, 10h, 12h, 14 and 17h after the first EdU injection), we quantified the number of EdU positive electroporated progenitors (triple positive for EdU, pRb and GFP) over the total population of electroporated progenitor cells (pRb and GFP positive) (Figure 3B)." Explanation does not correspond to Figure 3B.

4. Inducing a premature expression of Cdkn1c in progenitors triggers the transition to neurogenic modes of division (Figure 4.).

- "We took advantage of the Pax7 locus, which is expressed in progenitors in the dorsal domain at a level similar to that observed for Cdkn1c in neurogenic precursors (Supplementary Figure 4A)". Missing reference or data showing that Pax7 is restricted to the dorsal domain.

- "its intensity was similar to the one observed for endogenous Myc-tagged Cdkn1c in progenitors (Figure 4B and Supplementary Figure 4E), and remained below the endogenous level of Myc-tagged Cdkn1c observed in nascent neurons, confirming the validity of our strategy". It would be valuable to add a quantification to demonstrate this point, either by fluorescence levels or WB of nls-GFP cells.

- For Figure 4C and D, it would be valuable to add images to illustrate the quantification.

- "At the population level, at E4, Cdkn1c expression from the Pax7 locus resulted in a strong reduction in the number of progenitors (pRb positive cells)". Indicate in the main text that this is 48hpe.

- Legend of figure 4D should indicate that the quantification has been done 24hpe.

- "To circumvent the cell cycle arrest that is triggered in progenitors by strong overexpression of Cdkn1c (Gui et al., 2007)". It would be advisable to expand on this reference on the text, or ideally to include a simple Cdkn1c overexpression experiment.

- "We observed a massive increase in the proportion of neurogenic (PN and NN) divisions rising from 57% to 84% at the expense of proliferative pairs (43% PP pairs in controls versus 16% in misexpressing cells, Figure 4D)." adding the percentages in the main text is a bit inconsistent with how the rest of the data is presented in the rest of the sections.

- Figure sup 4C includes references to 3 gRNAs even when only one is used in the study.

5. The proneurogenic activity of Cdkn1c in progenitors is mediated by modulation of cell cycle dynamics (Figure 5)

- "we targeted the CyclinD1/CDK4-6 complex, which promotes cell cycle progression and proliferation, and is inhibited by Cdkn1c." reference missing

- It would be valuable to add an image to illustrate what is quantified in Figure 5D, Figure F

and Figure G.

- It would be informative to include experimental set-up information (e.g. hae) in Figures 5A, 5B, 5F and 5G.
- Clarify if analysis is restricted to the dorsal progenitors or the whole dorsoventral length of the tube.

Discussion:

- "Nonetheless, studies in a wide range of species have demonstrated that beyond this binary choice, cell cycle regulators also influence the neurogenic potential of progenitors, i.e the commitment of their progeny to differentiate or not (Calegari and Huttner, 2003; FUJITA, 1962; Kicheva et al., 2014; Lange et al., 2009; Lukaszewicz and Anderson, 2011a; Pilaz et al., 2009; Smith and Schoenwolf, 1987; Takahashi et al., 1995)." Should include maybe references to Peco et al. Development 2012, Roussat et al. J Neurosci. 2023).
- "This occurs through a change in the mode of division of progenitors, acting primarily via the inhibition of the CyclinD1/CDK6 complex." The data shown in the paper does not demonstrate that Cdkn1c is inhibiting CyclinD1, only that knocking down both mRNAs counteracts the effect of knocking down Cdkn1c alone at the general tissue level and in the percentage of PP/PN/NN clones. This statement should be qualified.

Other comments:

- There is a general lack of consistency in indicating the timing of the experiments, both in terms of embryonic stage/day and in terms of hours-post-electroporation.
- To improve clarity for the reader, it would help if electroporation was shown consistently on the same side of the neural tube. If electroporation has been performed at different sides and this is reflected in the figures, it would be advisable to explain on the figure legend.
- Figure legends should include the number of embryos/tissue sections analysed for each experiment, as well as information on whether the sections were cryostat or vibratome.
- Overall, there is a lack of consistency in the figures regarding how much information is available to the reader (e.g. Sup Figure 2A, in the panel mRNA in situ hybridisation of Cdkn1c is referred to only as Cdkn1c whereas in Sup figure 5 the in situ reads as CCND1 mRNA). Readability would improve a lot if figures included information on what is an electroporated fluorescent tag or an immunostaining (similar to the label in sup 4D) as well as the exact stage and hours after electroporation where relevant.
- "Primary antibodies used are: chick anti-GFP (GFP-1020 - 1:2000) from Aves Labs; goat

antiSox2 (clone Y-17 - 1:1000) from Santa Cruz". There is no Sox2 immunostaining in the article.

2. Significance:

Significance (Required)

In neural development, there is a progressive switch in competence in neural progenitor cells, that transition from a proliferative (able to expand the neural progenitor pool) to neurogenic (able to produce neurons). Several factors are known to influence the transition of neural progenitor cells from a proliferative to a neurogenic state, including the activity of extracellular signalling pathways (e.g. SHH) (Saade et al. 2013, Tozer et al. 2017). In this study, the authors perform scRNA-seq of the cervical neural tube of chick at a stage of both proliferative and neurogenic progenitors are present, and identify transcriptional differences between the two populations. Among the differently expressed transcripts, they identify Cdkn1c (p57-Kip2) as enriched in neurogenic progenitors. Initially characterized as a driver of cell cycle exit in newborn neurons, the authors investigate the role of Cdkn1c in cycling progenitors.

The authors find that knock-down of Cdkn1c leads to an increase in proliferative divisions at the expense of neurogenic divisions. Conversely, misexpression of Cdkn1c in proliferative progenitors leads to a switch to neurogenic divisions. Furthermore, they find that knock-down of Cdkn1c shortens G1 phase of the cell cycle, suggesting a link between G1 length and neurogenic competence in neural progenitor cells. Cell cycle length has previously been linked to competence of neural progenitors, and it has been described that longer G1 duration is linked to neurogenic competence (e.g. Calegari F, Huttner WB. 2003).

The strengths of the study include:

The identification of a subset of genes enriched in neurogenic vs. proliferative progenitors. Since the transition from proliferative to neurogenic competence is a gradual process at the tissue level, the classification of proliferative vs. neurogenic progenitors based on a score of transcripts and the identification of a subset of transcripts that are enriched in neurogenic progenitors is a valuable contribution to the neurodevelopmental field.

- The somatic knock-in strategy used to induce low-level overexpression of Cdkn1c in proliferative progenitors is an elegant strategy to induce overexpression in a subset of cells in a controlled manner and is a valuable technical advance.

- The characterization of a specific role of Cdkn1c in regulating cell cycle length in cycling progenitors is novel and valuable knowledge contributing to our understanding of how

regulation of cell cycle length impacts competence of neural progenitors.

The aspects to improve:

- The sc-RNAseq isolated genes enriched in neurogenic versus proliferative progenitors, providing valuable insight into the gradual transition from proliferative to neurogenic competence at the tissue level. However, this gene subset requires clearer representation and detailed characterization. Additionally, the full scRNA-seq dataset should be made publicly available to support further research in neurodevelopment.

- The characterization of Cdkn1c dynamics in cycling progenitors using endogenous tagging of the Cdkn1c transcript with a Myc tag is an elegant way to investigate the dynamics of Cdkn1c-myc along the cell cycle. However, it would be much more powerful if combined with a careful characterization of pRb immunostaining along the cell cycle in this tissue, as well as the quantifications and controls proposed.

- Retinoblastoma protein (Rb) and cyclin D play a key role in regulating the G1/S transition, with cyclin D/CDK complexes phosphorylating Rb. Given that CDKN1c primarily inhibits the cyclin D/CDK6 complex, it likely affects pRb expression or phosphorylation. This suggests pRb may be a direct target of CDKN1c, making it an unreliable marker for tracking and quantifying neurogenic progenitors through CDKN1c modulation. In light of this, it would be more appropriate to consider pRb as a CDKN1c target and discuss the molecular mechanisms regulating cell cycle components. A more precise approach would involve using other markers or targets to quantify neural precursor division modes at earlier stages of neurogenesis.

- Many of the conclusions of the study are based on experiments performed using the FlashTag dye in order to perform clonal analysis of proliferative vs. neurogenic divisions. It would be very valuable to further characterize the reliability of this tool as well as to provide more information on the criteria used to determine the fate of the pairs of sister cells.

- The somatic knock-in strategy used to induce low-level overexpression of Cdkn1c in proliferative progenitors is an elegant strategy to induce overexpression in a subset of cells in a controlled manner. It would be valuable to further characterize the dynamics of Cdkn1c expression using this tool and to provide proof that Pax7 expression is not altered in guides with the knock-in event.

- The presentation of the existing literature could be more up to date.

- The presentation of the data in the figures could be improved for readability.

The sc-RNA seq data and the technical advances could be of interest for an audience of researchers using chick as a model organism, and working on neurodevelopment in general. Furthermore, the characterization of Cdkn1c as a regulator of G1 length in cycling progenitors and its implications for neurogenic competence could be of general interest for

people working on basic research in the neurodevelopmental field.

Field of expertise of the reviewer: neural development, cell biology, embryology.

3. How much time do you estimate the authors will need to complete the suggested revisions:

Estimated time to Complete Revisions (Required)

(Decision Recommendation)

Between 1 and 3 months

Yes

Revision Plan

Manuscript number: RC-2025-02891

Corresponding author(s): Evelyne Fischer, Xavier Morin

1. General Statements

The main message of our study is that a single “cell cycle regulator”, the CDK inhibitor CDKN1c, can contribute successively to different cellular functions and behaviors in a developmental process depending on its expression level. This type of dual function is generally difficult to study in an integrated context *in vivo*, where classical gain and loss-of-function approaches do not easily discriminate between distinct roles. In this work, we have designed novel experimental strategies to study the role played by low level expression of CDKN1c in the “neurogenic” competence of neural progenitors, and disentangle this role from its previously characterized activity in cell cycle exit in newborn neurons.

We were pleased that all three reviewers appreciated the importance of our findings, and that they noted the relevance and elegance of the methodology that we have developed to reach our conclusions.

Here, we would like to address two comments from reviewer#3.

The first one is a “general” comment from this reviewer, who stated that **“the concept of cell cycle regulators interacting to control the duration of specific cell cycle stages and influencing progenitor cell division modes has been explored before”**. We agree with the reviewer that the link existing between cell cycle regulation/duration and modes of division has already been explored, and indeed it is largely stated in our discussion. From this perspective, our study adds Cdkn1c as a new player in the expanding list of regulators involved in this process. However, as stated above, the main novelty of our observation resides in the demonstration of distinct roles for the same cell cycle regulator in successive cellular decisions, first in cycling progenitors (controlling their neurogenic competence) and later in newborn neurons (controlling their cell cycle exit). The functional exploration of this dual role was made possible by the specific partial gain and loss-of-function strategies that we have designed.

The second comment from reviewer#3 is more technical: **“This study presents several ambiguities and lacks precision in its analytical methodologies and quantification approaches, which contribute to confusion and potential bias. To enhance the reliability of the conclusions, a more rigorous validation of the methods employed is essential. This study introduces a novel approach to tracking the fate of sister cells from neural progenitor divisions to infer the division modes. While previous methods for analyzing the division mode of neural progenitor cells have been implemented, rigorous validation of the approach introduced by Mida et al. is necessary.”**

The reviewer refers in particular to the novel method of clonal analysis of sister cells pairs that we use for the retrospective identification of the mode of division of the mother cell based on the identity of their daughters. Regarding the validation of the method, we are convinced of its robustness and validity, but we agree that our description in the original version of the manuscript may not have been sufficiently clear and detailed. We have now modified the manuscript accordingly, and also offer additional explanations in our detailed response to reviewers, as well as in the “Description of the revisions that have already been incorporated” section below.

Regarding the “***previous methods for analyzing the division mode of neural progenitor cells***”, we think that they do have some drawbacks, and this was precisely the reason why we have developed an alternative approach:

In the chick spinal cord, one method consists in the electroporation of fluorescent reporters whose combination of expression in progenitors was described to discriminate between proliferative, neurogenic asymmetric and neurogenic terminal modes of division (Saade et al 2013, le Dreau et al, 2014, Bonnet et al, 2018). One drawback of this approach lies in its assumption that the expression of these reporters is a downstream target of the mechanisms that we affect experimentally.

Another approach consists in labeling clones with a genetically encoded lineage tracer, such as Cre recombinase dependent “brainbow” markers. Although we and others have already used this retrospective approach in the past (eg. Morin et al, 2007; Tozer et al, 2017, Bonnet et al, 2018), it retains ambiguities because the time of birth of the analyzed cells is unknown, and we lack reliable markers of identity in the first few hours after mitosis. Our new FlashTag based approach solves this issue by analyzing a synchronous cohort of cells whose birthdate is known. This allows to select a timepoint for analyses at which we can attribute an identity to pairs of sister cells using pRb as a discriminating marker between progenitor (pRb+) and neuronal (pRb-) identities. Ideally, including a “positive” marker of the neuronal fate could comfort this type of analysis, but unfortunately the best available markers of neuronal differentiation in the chick are expressed late in the differentiation process. We propose, however, to introduce one such positive marker in a modified version of our strategy for the revised version of the manuscript.

Another approach, mentioned by Reviewer #1, would consist in live monitoring of dividing progenitors and fate acquisition in their progeny. This is the gold standard for direct correlation of progenitor division and daughter cells fate and we have used this approach to link asymmetric subcellular events during mitosis and asymmetric fate in the progeny (Tozer et al, 2017, Bunel et al, 2024). However, it remains a technical challenge, and we do not think that the added value would be significant in comparison to our approach of retrospective labelling in the context of the current study, where we compare the different proportions of modes of division between two experimental conditions, but do not monitor any potential asymmetries during mitosis.

2. Description of the planned revisions

Insert here a point-by-point reply that explains what revisions, additional experimentations and analyses are planned to address the points raised by the referees.

Additional experiments requested by reviewers

1) As requested by Referees 1 and 3, we will perform additional experiments to provide some quantitation of the shRNA effect.

We agree with these two reviewers that the evaluation of the level of mRNA reduction by our in situ hybridization approach was not quantitative. Rather than qRT-PCR on sorted cells, that would present the advantage of focusing solely on electroporated cells, but would result in an

averaged quantification of Cdkn1c depletion, we propose an alternative approach consisting in monitoring the level of Cdkn1c protein, assessed through Cdkn1c-Myc signal in knock-in cells, which offers a direct read-out of the protein levels in the ventricular and mantle zones. Since our shRNA strategy of “partial knock-down” is based on the idea that the shRNA effect should be more complete in progenitors expressing Cdkn1c at low levels than in newborn progenitors that express the protein at a higher level, validating the shRNA in the Cdkn1c-Myc knock-in background will allow to compare the Myc signal intensity in both the ventricular and mantle zones between control and Cdkn1c shRNA conditions.

2) To answer Referee #3’s suggestion, **“It would be nice if the authors could validate guide cutting efficiency”** we will validate guide cutting efficiency using the Tide method (Brinkman et al, 2014). In addition, as requested, we will perform genomic PCR experiments to confirm in-frame insertion of the Myc tags at the Cdkn1c locus

3) To explore the endogenous level of Cdkn1c expression in our somatic knock-in strategy, Referee 3 proposed **“an analysis of the dynamics of Myc immunoreactivity in combination of pRb in all three gRNAs (highlighted in Supplementary Figure 1)”**. In addition they asked, **“to provide a quantification of said dynamics”**

These are two interesting suggestions. To complement our data with guide #1, we have already performed a first set of Myc-immunostaining experiments on transverse sections in the context of guide #3, showing exactly the same pattern of Myc signal (see New supplementary Figure 1).

We will perform the suggested quantifications (e.g. plotting myc intensity / pRb immunoreactivity along the apicobasal axis of the tissue) using guides #1 and #3, which both show a good KI efficiency. We do not think it is useful to do these experiments with guide #2, whose efficiency is much lower, and which would lead to a very sparse signal.

4) Concerning the Cdkn1c overexpression strategy, Referees #1 and #3 raise a similar concern about the **“quantification of the intensity of endogenous Myc-tagged Cdkn1c to confirm the validity of the Pax7 locus knock in”**. We will include a more precise quantification of the “misexpressed” compared to “endogenous” CDKN1c-Myc levels. We will perform a quantification of the signal intensity in the ventricular and mantle zones as described above for the revised version of the manuscript.

5) Reviewer #3 proposed to use **“a more precise approach using other markers or targets to quantify neural precursor division modes at earlier stages of neurogenesis”**.

To complement our analyses of the modes of division, we propose to use a positive marker to assess neural identity in parallel to the absence of pRb within pairs of cells. This approach may be the most meaningful in the gain of function context (Pax7 driven expression of Cdkn1c) because in this context, the time-point to reach the plateau of Rb phosphorylation used in our FT-based assay may indeed be delayed. On the opposite, in the context of loss of functions, the plateau may be reached earlier, which would have no effect on this assay.

6) Reviewer #3 proposed to ***“measure G1 length as in Figure 5D for the double *cdkn1c-sh - ccnd1-sh* knock down condition, to see if it rescues G1 length, as well as in the *Ccnd1* knock down condition alone to see if it increases G1 length in this context as well.***

We will perform cumulative EDU incorporation experiments similar to that shown in Figure 5D to measure G1 length for the Cdkn1c-sh - CCND1-sh double knock down conditions, as well as in the CCND1 knock down condition alone.

7) As suggested by Reviewer #3, ***“It would be valuable to assign cell cycle stage to neural progenitor cells (based on cell cycle score) and determine whether *cdkn1c* at the transcript level also shows enrichment in G1 cells considered to be progenitors”.***

We have so far refrained from performing the suggested combined analysis based on cell cycle and cell type scores, as the “neurogenic progenitor population” (based on neurogenic progenitor score values) in which Cdkn1c expression is initiated represents a small number of cells in our scRNAseq, and felt that the significance of such an analysis is uncertain. We will perform this analysis in the revised version.

Manuscript modifications suggested by reviewers

1) We will modify the introduction and discussion in several instances, in order to address suggestions by Reviewers 2 and 3 to:

- expand the description of the role of Cdkn1c during cortical development
- add references to its role in other contexts and/or species.
- expand the discussion on the cross talk between neurogenic factors and CDK inhibitors in other cellular contexts.
- add a dedicated paragraph in the discussion to answer reviewer#2's questions: is there evidence for Cdkn1c low/high expression in mammalian spinal cord? Or maybe of other Cdk inhibitors? Is Cdkn1c also involved in cell cycle exit during gliogenesis or is there another Cdk inhibitor expressed at later developmental stages?

2) Reviewer #1 suggested that ***“A model cartoon to summaries outcomes would be useful.”***

We thank the reviewer for the suggestion. We will propose a summary cartoon for the revised version of the manuscript.

3. Description of the revisions that have already been incorporated in the transferred manuscript

Response to specific comments from the reviewers and related modifications that have already been implemented in the main text or Methods sections

1) One of the major comments raised by Reviewer #3 concerns **“the characterization of pRb dynamics throughout the cell cycle in the chick spinal cord.”**

We have taken this comment into account and expanded on pRb characteristics in the first occurrence of pRb as a marker of cycling cells in the manuscript. The modifications rely on:

- the quotation of several studies showing that phosphorylation of Rb is regulated during the cell cycle, as also requested by Reviewer 1 in the main text. These studies showed that pRb is not detectable during a period of variable length in early G1 in several cell types (Moser et al, 2018, Spencer et al, 2013, Gookin et al, 2017), including neural progenitors in the developing chick spinal cord (Molina et al, 2022). Apart from this absence in early G1, pRb is detected throughout the rest of the cell cycle until mitosis.

- an entire remodeling of the section that describes the expression of Myc-tagged Cdkn1c relative to pRb as follow (page 8, line 171): *“To ascertain that Cdkn1c is translated in neural progenitors, we used an anti-pRb antibody, recognizing a phosphorylated form of the Retinoblastoma (Rb) protein that is specifically detected in cycling cells (Gookin et al., 2017; Moser et al., 2018; Spencer et al., 2013) , including neural progenitors of the developing chick spinal cord (Molina et al., 2022). In the ventricular zone of transverse sections at E4 (48hae), we detected triple Cdkn1c-Myc/GFP/pRb positive cells (arrowheads in Figure 2B), providing direct evidence for the Cdkn1c protein in cycling progenitors. We also observed many double GFP/pRb positive cells that were Myc negative (arrowheads in Figure 2B). The observation of UAS-driven GFP in these pRb-positive cells is evidence for the translation of Gal4 and therefore provides a complementary demonstration that the Cdkn1c transcript is translated in progenitors. The absence of Myc detection in these double GFP/pRb positive cells also suggests that Cdkn1c/Cdkn1c-Myc stability is regulated during the cell cycle.*

Finally, we observed double Myc/GFP-positive cells that were pRb-negative (Figure 2B; asterisks). One characteristic of Rb phosphorylation as a marker of cycling cells is a period in early G1 during which it is not detectable, as described in several cell types (Gookin et al., 2017; Moser et al., 2018; Spencer et al., 2013) including chick spinal cord neural progenitors (Molina et al., 2022). Using a method that specifically labels a synchronous cohort of dividing cells in the neural tube, we similarly observed a period in early G1 during which pRb is not detectable in some progenitors at E3 (See Supplementary Figure 2 and Methods). Hence, the double Myc/GFP positive and pRb negative cells may correspond to progenitors in early G1. Alternatively, they may be nascent neurons whose cell body has not yet translocated basally (see Figure 2C). Finally, we observed a pool of GFP positive/pRb negative nuclei with a strong Myc signal in the region of the mantle zone that is in direct contact with the ventricular zone (VZ), corresponding to the region where the transcript is most strongly detected (see Figure 2A). This pool of cells with a high Cdkn1c expression likely corresponds to immature neurons exiting the cell cycle and on their way to differentiation (Figure 2B; double asterisks). In addition, a few Myc positive cells were located

deeper in the mantle zone, where the transcript is no more present, suggesting that the protein is more stable than the transcript.

In summary, our dual Myc and Gal4 knock-in strategy which reveals the history of Cdkn1c transcription and translation confirms that Cdkn1c is expressed at low level in a subset of progenitors in the chick spinal neural tube, as previously suggested (Gui et al., 2007; Mairet-Coello et al., 2012). In addition, the restricted overlap of Cdkn1c-Myc detection with Rb phosphorylation suggests that in progenitors, Cdkn1c is degraded during or after G1 completion. This may explain why a classical immunohistochemistry approach with an anti-Cdkn1c antibody only detected the protein in very few progenitors in the developing mouse CNS (Gui et al., 2007; Mairet-Coello et al., 2012)."

- a more detailed description of our own characterization of pRb dynamics in a synchronous cohort of cycling cells, which reveals a similar heterogeneity in the timing of the onset of Rb phosphorylation after mitosis. This description was initially shown in Supplementary Figure 3 and has been transferred to a new Supplementary Figure 2 to account for the fact that it is now cited earlier in the manuscript.

Regarding the specific question of the **“suitability (of pRb) as a neurogenic division marker”**: we do not directly “use phosphorylated retinoblastoma protein (pRb) as a marker for neurogenic progenitors”, but we use Rb phosphorylation to discriminate between progenitors (pRb positive) and neurons (pRb negative) identity in pairs of sister cells to retrospectively identify the mode of division of their mother.

Given that Rb is unphosphorylated during a period of variable length after mitosis (see references above), pRb is not a reliable marker of ALL cycling progenitors. We developed a time course of pRb detection in cohorts of FlashTag-positive pairs of sister cells (FlashTag injection in the neural tube labels specifically cells undergoing mitosis within 30 minutes after injection; Baek et al, 2018) to identify the timepoint corresponding to the plateau, and therefore after which Rb is phosphorylated in all cycling progenitors (see the new Supplementary Figure 2). From this timepoint, Rb phosphorylation becomes a discriminating factor between cycling progenitors (pRb positive) and non-cycling neurons (pRb negative).

We are confident that this provides a solid foundation for the determination of the identity of pairs of sister cells in all our Flash-Tag based assays, which retrospectively identify the mode of division of a progenitor on the basis of the phosphorylation status of its daughter cells 6 hours after division.

We propose to modify the main text to describe the strategy and protocol more explicitly, by introducing the sentence highlighted in yellow in the following paragraph where the paired-cell analysis is first introduced (in the Results section 3 on Cdkn1c downregulation, page 10, line 253):

“This approach allows to retrospectively deduce the mode of division used by the mother progenitor cell. We injected the cell permeant dye “FlashTag” (FT) at E3 to specifically label a cohort of progenitors that undergoes mitosis synchronously (Baek et al., 2018; Telley et al., 2016 and see Methods), and analyzed the fate of their progeny six hours later (Figure 3D). Our characterization of pRb immunoreactivity in the tissue had established beforehand that 6 hours after mitosis, all progenitors can reliably be detected with this marker (Supplementary Figure 2,

Methods). *Therefore, at this timepoint after FT injection, two-cell clones selected on the basis of FT incorporation and similar GFP signal intensity can be categorized as PP, PN, or NN based on pRb positivity (P) or not (N) (see Methods, Figure 3G and Supplementary Figures 2 and 4)."*

We also modified accordingly the legend to Supplementary Figure 2 (previously Supplementary Figure 3) which describes the identification of the plateau of pRb:

"Supplementary Figure 2: pRb is a reliable marker of progenitor cells six hours after mitosis

Left: Experimental scheme for the determination of Rb phosphorylation status in FlashTag-labelled cells. Wild type E3 embryos are injected with FlashTag (FT), which specifically labels cells synchronously undergoing mitosis at the time of injection. Embryos are collected at different timepoints (t) after injection to determine at which timepoint pRb positivity reaches a plateau in the cohort of FlashTag-positive daughter cells.

Right: Time course of pRb expression in FlashTag-positive pairs of sister cells at consecutive time points after injection. FlashTag injection was performed at E3 and embryos were harvested at the indicated timepoints after injection. Thoracic vibratome sections were immunostained with anti-pRb antibody to evaluate the pRb status in pairs of FlashTag positive sister cells.

The proportion of pairs with two pRb positive cells (green), one pRb positive cell (yellow) or zero pRb positive cell (red) becomes stable after 4h30. This indicates that the proportion of individual pRb-positive cells (red line) reaches a plateau corresponding to the proportion of cycling progenitors in this cohort. Therefore, after that time point, pRb positivity becomes a reliable marker of progenitor status in FlashTag labelled cells. This allows the retrospective attribution of the mode of division (PP, PN or NN) of the mother of pairs of sister cells via pRb labelling. N=2 to 5 embryos per time point. The number of pairs analyzed at each time point is indicated at the bottom of the diagram."

2) Another major comment from Reviewer 3 concerns the use of FlashTag to perform our clonal analysis: **"Furthermore, the study employs FlashTag labeling to track daughter cells post-division, but the 16-hour post-injection window may result in misidentification of sister cells due to the potential presence of FlashTagged cells that did not originate from the same division.**

This introduces a risk of bias in quantification, data misinterpretation, and potential errors in defining division modes. A more rigorous validation of the FlashTag strategy and its specificity in tracking division pairs is necessary to ensure the reliability of their conclusions".

The reviewer probably mistyped and meant 6-hour post injection, which is the duration that we use for paired cell tracking. In addition to the FlashTag label, the identification of pairs of sister cells rely on the electroporation reporter to assess clonality. Altogether, we combine 5 criteria, already described in the Methods section to define a clonal relationship:

- 2 cells are positive for Flash Tag
- The Flash Tag intensity is similar between the 2 cells
- The 2 cells are positive for the electroporation reporter
- The electroporation reporter intensity is similar between the two cells

- the position of the two cells is consistent with the radial organization of clones in this tissue (Leber and Sanes, 1995; Loulier et al, 2014): they are found on a shared line along the apico-basal axis, and share the same Dorso-Ventral and Antero-Posterior position.

To clarify this point, we propose to modify the paragraph describing these criteria in the Methods section (Clonal analysis of sister cell identities and mode of division, page 32, line 863) to include the sentence highlighted in yellow:

“Cell identity of transfected GFP positive cells was determined as follows: cells positive for pRb and FT were classified as progenitors and cells positive for FT and negative for pRb as neurons. In addition, a similar intensity of both the GFP and FT signals within pairs of cells, and a relative position of the two cells consistent with the radial organization of clones in this tissue (Leber and Sanes, 1995; Loulier et al., 2014) were used as criteria to further ascertain sisterhood. This combination restricts the density of events fulfilling all these independent criteria, and can confidently be used to ensure a robust identification of pairs of sister cells.”

3) Reviewer #1: ***“The authors state that Cdkn1c “was expressed at low levels in a salt and pepper fashion in the ventricular zone, where the cell bodies of neural progenitors reside, and markedly increased in a domain immediately adjacent to this zone which is enriched in nascent neurons on their way to the mantle zone. In contrast, the transcript was completely excluded from the mantle zone, where HuC/D positive mature neurons accumulate.” It may be helpful if this expression pattern could be described in a bit more detail highlighting the changes that occur between E3 and E4”.***

We have now reformulated this paragraph in the Results section 2 (page 7, line 142) as follows: *“At E3, the transcript was expressed at low levels in a salt and pepper fashion in the ventricular zone, where the cell bodies of neural progenitors reside (Saade and Marti, 2025). One day later, at E4, this salt and pepper expression was still detected in the ventricular zone, while it markedly increased in the region of the mantle zone that is immediately adjacent to the ventricular zone. This region is enriched in nascent neurons on their way to differentiation that are still HuC/D negative. In contrast, the transcript was completely excluded from the more basal region of the mantle zone, where mature HuC/D positive neurons accumulate.”*

We have also outlined the border between the ventricular and mantle zones in the relevant panels in Figure 2A to facilitate the interpretation of the expression pattern.

4) Reviewer #1 pointed that ***“In Fig 4D, the increase in neurogenic divisions is mainly because of the rise in terminal NN divisions according to the graph, but no clear increase in PN divisions. Could the authors comment on the significance of this?”*** and Reviewer # 2 had a comment on ***“the calculations of PN and NN frequencies are merged in the text, so perhaps describing PN and NN numbers separately will help better understand the dynamics of this gradual process (especially since there is little to no difference in PN)”.***

We have modified the manuscript in the Results section 4 (page12 line 297) to elaborate on our interpretation of this result: *“We observed an increase in the proportion of terminal neurogenic (NN) divisions and a decrease in proliferative (PP) divisions (Figure 4D). This*

suggests that Cdkn1c premature expression in PP progenitors converts them to the PN mode of division, while the combined endogenous and Pax7-driven expression of Cdkn1c converts PN progenitors to the NN mode of division. Coincidentally, at the stage analyzed, PP to PN conversions are balanced by PN to NN conversions, leaving the PN proportion artificially unchanged. The alternative interpretation of a direct conversion of symmetric PP into symmetric NN divisions is less likely, because the PN compartment was affected in the reciprocal Cdkn1c shRNA approach (see Figure 3F, now 3H)."

5) Reviewer #1: **"The proportion of pRb-positive progenitors having entered S phase was stated to be higher at all time points; however, it is not significantly higher until 6h30 and is actually trending lower at 2h30"**.

We thank the reviewer # 1 for having pointed this out. We have modified the sentence in the main text (Results section 5, page13 line 327) as follow: *"We found that the proportion of pRb positive progenitors having entered S phase (EdU positive cells) was significantly higher at all time points examined more than 4h30 after FT injection in the Cdkn1c knock-down condition compared to the control population (Figure 5D)"*

6) To answer Reviewer #2's comment: **"what would the effect of expressing higher levels of Cdkn1c be?"** we now cite more explicitly in the main text the study by Gui et al., 2007, which used a strong overexpression of Cdkn1c from the CAGGS promoter, and better explain the difference of our approach. We propose the following modification (Results section 4, page11 line 268):

"We next explored whether low Cdkn1c activity is sufficient to induce the transition to neurogenic modes of division. A previous study has shown that overexpression of Cdkn1c driven by the strong CAGGS promoter triggers cell cycle exit of chick spinal cord progenitors, revealed by a drastic loss of BrdU incorporation 1 day after electroporation (Gui et al., 2007). As this precludes the exploration of our hypothesis, we developed an alternative approach designed to prematurely induce a pulse of Cdkn1c in progenitors, with the aim to emulate in proliferative progenitors the modest level of expression observed in neurogenic progenitors. We took advantage of the Pax7 locus, which is expressed in progenitors in the dorsal domain at a level similar to that observed for Cdkn1c in neurogenic precursors (Supplementary Figure 6A)."

7) To answer Reviewer #3's question, **"Can the authors check for ectopic mitoses at 24hpe and 48hpe?"**, we provide a new Supplementary Figure 3E and F and have modified the main text to include these data (page 10, line 238):

"In the context of a full knock-out of Cdkn1c in the mouse spinal cord, a reduction in neurogenesis was also observed, which was attributed to a failure of prospective neurons to exit the cell cycle, resulting in the observation of ectopic mitoses in the mantle zone (Gui et al, 2007). In contrast with this phenotype, using an anti phospho-Histone3 antibody, we did not observe any ectopic mitoses 24 or 48 hours after electroporation in our knock-down condition (Supplementary Figure 3E-F). This is consistent with the fact that we also do not observe ectopic cycling cells with pRb (Figure 3A and D) and Sox2 (Supplementary Figure 3E-F) antibodies. We therefore postulated that the reduced neurogenesis that we observe upon a partial Cdkn1c knock-down"

may result from a delayed transition of progenitors from the proliferative to neurogenic modes of division”

We also include some minor modifications in the text, in response to Reviewer 's comments:

- page 5, line 94: the list of the genes used to define the Progenitor, and Neuron score: (Progenitor genes = Sox2, Notch1, Rrm2, Hmgb2, Cenpa, Ube2c, Hes5; Neuron genes = Tubb3, Stmn2, Stmn3, Nova1, Rtn1, Mapt).
- page 8 line 196: “ ***In Fig 2B, there are a number of Myc labelled cells in the mantle zone, whereas the in situ images show no appreciable transcript expression. Is this because the protein but not the transcript is present in these cells? Could the authors comment on this?***” We have added a sentence: *In addition, a few Myc positive cells were located deeper in the mantle zone, where the transcript is no more present, suggesting that the protein is more stable than the transcript.*

Modifications that have already been introduced to Figures/New figures

We also provide representative images illustrating our analyses and some new experimental data requested by the reviewers, presented in new panels of the preexisting figures or in new additional figures.

- 1) We provide the requested **representative images** of experiments that were used for quantifications for the:
 - Graph in Figure 3C (now 3C and 3F): images are added in Figure 3A-B and 3D-E;
 - Graph in Figure 4C (now 4C and 4D): images are added in supplementary Figure 7C;
 - Graph in Figure 5D: images are added in Figure 5E
 - Graph in Figure 5G (now 5H and 5I): images are added in Supplementary figure 8B
 - We now provide additional examples of different types of clones in a new supplementary Figure 4

- 2) New experimental data
 - Caspase3 experiments that were only mentioned in the original version are now presented in Supplementary Figure 3D, in response to Reviewer #1's request.
 - Myc-immunostaining experiments in the context of Cdkn1c gRNA#3 to complement and validate our data with gRNA#1. They are presented in a revised version of Supplementary Figure 1. *This experiment shows the same pattern of Myc signal, confirming the specificity of the spatial distribution of the Cdkn1c-Myc signal.*
 - anti phospho-Histone3 and anti-Sox2 immunostainings, at 24 and 48 hours post electroporation **to check for ectopic mitoses upon Cdkn1c knock-down**. These data have been added in the new Supplementary Figure 3E and F.
 - Pax7 immunostaining to document the expression of Pax7 upon a successful knock-in event at its locus. These data are presented in Supplementary Figure 7. When embryos were electroporated with either the Pax7-Cdkn1c-Gal4 or with the Pax7-Gal4 control constructs, we observed a modest increase in Pax7 signal intensity in these cells. This does not affect the interpretation of the Cdkn1c overexpression phenotype, because we used the Pax7-Gal4 construct that shows the same modification of Pax7 stability as a control for this experiment. We have introduced this comment in the legend of Supplementary Figure 7. *A similar modest increase in Pax7 immunofluorescence is observed in knock-in cells (green, asterisks) in both*

conditions (compare top and bottom rows). Since this increase occurs irrespective of whether Cdkn1c is overexpressed or not, it does not affect the interpretation of phenotypes comparing the two conditions.”

3) The reviewers pointed some missing information in **the Figures**. We have addressed all these points and modified the Figures. We provide below a list of these modifications:

a) Informations in the Figures

Sample numbers are now provided in the figure legends (numbers of cells analyzed and/or numbers of embryos), except for data in Figure 5, for which cell numbers are presented in a new Supplementary Table 1.

Information on **whether the sections were cryostat or vibratome is now provided in the** relevant figure legends (highlighted in yellow in the corresponding Legends).

We have clarified that analyses were carried out on two thirds of the neural tube (dorsal 2/3^d), excluding the ventral zone, except for the Pax7-Cdkn1c misexpression analysis. We have modified this description in the Methods (section “Image quantification”) as follows: *“Thus, to reason on a more homogeneous progenitor population, we restricted all our analysis to the dorsal two thirds of the neural tube, except for the Pax7-Cdkn1c misexpression analysis, which was performed in the more dorsal Pax7 domain.”* This is valid both for the whole population and clonal analyses.

To improve clarity for the reader, we have modified the figures to systematically show the electroporated side of the neural tube on the same side of the image. We have also homogenized the nomenclature in the figures and information on the electroporated constructs as well as the stage and timing after electroporation. We have also replace 'misexpressed' and 'endogenous' by i) *Pax7-Cdkn1c-Myc* and ii) *endogenous Cdkn1c-Myc*, in the Figure 4B, as suggested by Referee #2.

-We understand from Reviewer #1’s comment that depicting an “undefined” population on our graphs may cause some confusion. We therefore propose to present the data on pRb and HuC/D quantifications in different graphs, rather than on a combined plot, and to not reference undefined cells in Figure 3, as well as in Figures 4 and 5 depicting the gain of function and double knock-down experiments. We have implemented these changes in updated versions of the Figures.

b) Specific Modifications of the Figures:

Figure 1

Panel 1A. We have now added the detail of what ‘filtering’ means in the diagram

Panel 1C: We have corrected the color code errors in Cell Clusters in Figure 1C (cluster 4 was yellow in the legend but ocre in the figure)

Figure 2

Panel 2A: “Scale bar is missing in E3 and E4”. The scale bar is actually valid for the whole panel A. The E2 section in the original figure appeared as “large” as the E3 section along the DV

axis probably because the cutting angle was not perfectly transverse at E2, artificially lengthening the section. In a new version of the figure, we have replaced the E2 images with another section from the same experiment. The scale bar remains valid for the whole panel.

In addition, we **annotate the ISH images with a dotted line** that separates the ventricular zone from the mantle zone at E3 and E4 to show the ventricular and mantle zones as defined by immunofluorescence.

Panel 2B: We have now added as requested a diagram for the knock-in strategy, and modified the legend of the figure accordingly. We have also added the post-electroporation timing and guide used in the main text (modification highlighted in yellow)

Figure 3

Panel 3C now precedes 3A and 3B, as requested by reviewer 3. In addition, in order to cite **Figure panels in order**, we have now added a common citation of the panels in the main text referring to analyses at 24 and 48 hours after electroporation (now Figure 3A-F, highlighted in yellow). Finally, we now provide pictures of the electroporated NT at both 24hpe and 48hpe, and highlighted the dorsal part of the neural tube that was used for quantification.

Figure 4B: Reviewer 1 pointed that **the brightness of row 2 column 1 is lower than the same image in row 2 column 2**. All *images in the figure are single Z* confocal images. Images in Column 2 were acquired with a 20x objective. *The insets shown in Columns 1 and 3 are not magnified crops from the 20x image, they are 100x confocal images acquired in the same section. Importantly, the 100x close ups in columns 1 and 3 are presented with the same display parameters*. This is now indicated in the legend of the Figure.

Figure 5: We have added the experimental set-up information (hae, stage of embryos) in panels 5A, 5B, 5F and 5G.

Supplementary Figure 1C and Supplementary Figure 4D: Scale bars have been added in the figures and corresponding legends

Supplementary Figure 3

Panel E (and related 4B): to help the reader understand the difference between the immuno signals when looking at the figure, we modified as requested the writing on the panel i) Pax7-Cdkn1c-Myc and ii) endogenous Cdkn1c-Myc,

Panel 3B (now transferred to Supplementary Figure 5 in the new version): we have corrected the color code errors which was switched (green for PP and red for NN) compared to the rest of the paper.

Additional references

The reviewers proposed to include additional references. These references have now been added in the main text (highlighted in yellow), and are listed below in order of appearance in the text:

1) page 5-6, line 107-109: references to the expression profile of the genes used to determine the progenitor and neuron score.

2) page 6 line 130: a reference indicating that Tis21 mRNA expression is switched off in neurons (Iacopetti et al, 1999)

3) A recent review on spinal cord development to illustrate the cell type distribution in spinal neural tube: page 7 line 144 (Saade and E. Marti, 2025) in the introduction section.

4) page 11 line 276: references to the expression profile of Pax7 in the dorsal neural tube (Jostes et al, 1990).

5) page 29 line 78: a reference showing a persistence of Sox2 protein in differentiating neurons of the human neocortex (Coquand et al, 2024) in the Methods section.

4. Description of analyses that authors prefer not to carry out

The following request from Reviewer 3 will not be addressed in our revision plan

“Furthermore, retinoblastoma protein (Rb) and cyclin D interact crucially to regulate the G1/S phase transition of the cell cycle, with cyclin D/CDK complexes phosphorylating Rb. Since the authors conclude that CDKN1c primarily acts by inhibiting the cyclin D/CDK6 complex, it is likely that CDKN1c influences pRb expression or phosphorylation state. This raises the possibility that pRb could be a direct target of CDKN1c, whose expression and phosphorylation would be altered in gain-of-function (GOF) and loss-of-function (LOF) analyses of CDKN1c. In light of this, it would be more appropriate to consider pRb as a CDKN1c target and discuss the molecular mechanisms regulating cell cycle components”.

We agree with the reviewer that Rb phosphorylation may be a direct or indirect target of CDKN1c activity. However, exploring the molecular aspects of the cellular and developmental phenomena that we describe in our manuscript is out of the scope of our study, although it would certainly represent an interesting follow up study.

We also consider that a few other minor points suggested by the same reviewer are dispensable or not appropriate, and we will not include them in the revision

1) “In the measurement of progenitor cell cycle length in Figure 5A, it would be more appropriate to present the nonlinear regression method described by Nowakowski et al. (1989), as has been commonly used in the field (Saade et al., 2013, PMID: 23891002, Le Dreau et al., 2014, PMID: 24515346, Arai et al., 2011, PMID: 21224845).”

The Nowakowski non linear regression method has been used often in the literature in the same tissue, and is generally used to calculate fixed values for Tc, Ts, etc... This method is based on several selective criteria, and in particular the assumption that “all of the cells have the same cycle times”. Yet, many studies have documented that cell cycle parameters change during the transition from proliferative to neurogenic modes of division during which our analysis is performed; live imaging data in the chick spinal cord have illustrated very different cell cycle durations at a given time point (see Molina et al, 2022). We therefore think that the proposed formulas do not reflect the heterogenous reality of neural progenitors of the embryonic spinal cord.

However, the cumulative approach described by Nowakowski is useful to show qualitative differences between populations (e.g. a global decrease of the cycle length, like in our comparison between control and shRNA conditions). For these reasons, we prefer to display only the raw measurements rather than the regression curves.

2) “Can the authors show that the plateau reached in Sup Figure 3 for pRb immunoreactivity corresponds to a similar dynamic for HuC/D immunoreactivity?”

The plateau for Rb phosphorylation in progenitors is reached before 6 hours post mitosis at E3. At the same age, we have previously shown (Baek et al, PLoS Biology 2018) in a similar time course experiment in FT+ cells that the HuC/D signal is not detected in newborn neurons 8 hours after mitosis. HuC/D only starts to appear between 8 and 12 hours, and still increases between 8 and 16 hours. The plateau would therefore be very delayed for HuC/D compared to pRb. This long delay in the appearance of this « positive » marker of neural differentiation is the main reason why we chose to use Rb phosphorylation status for the analysis of synchronous cohorts of pairs of sister cells, because pRb becomes a discriminating factor much earlier than HuC/D after mitosis.

3) “It would be valuable to add an image to illustrate what is quantified in Figure 5D, Figure F and Figure G”.

Regarding the requested images for Figures 4D and 5F, they correspond to the same types of images already shown in Figure 3E. Since we have now added several additional examples of representative pairs of each type of mode of division in the new Supplementary Figure 4, we do not think that adding more of these images in figures 4 and 5 would strengthen the result of the quantifications.

Some suggestions from reviewer 1 were explicitly labelled as optional, and we detail below why we will not perform them. These suggestions concern:

1) “whether a similar approach could be used to tag Cdkn1c with a fluorescent protein to enable live imaging of dynamics?”

Although it could be done, we have not attempted to do this for CDKN1c because our current experience of endogenous tagging of several genes with a similar expression level (based on our scRNAseq data) and nuclear localization (Hes5, Pax7) with a fluorescent reporter shows that the fluorescent signal is extremely low or undetectable in live conditions; Therefore we favored the multi-Myc tagging approach, and indeed we find that the Myc signal in progenitors is also very low even though it is amplified by the immunohistology method; this suggests that most likely, the only signal that would be detected -if any- with a fluorescent approach would be the peak of expression in newborn neurons.

2) “Whether live imaging could be used to watch divisions followed by fixation and immunostaining to confirm identity?”

We agree with the reviewer that direct tracking is the most direct method for the identification of pairs of sister cells. However, it remains technically challenging, and the added value compared to the retrospective identification would be limited, while requiring a great

workload, especially considering the many different experimental conditions that we have explored in this study.

3) “Whether CyclinD1 Could activity be directly assessed?”

This is an interesting suggestion. For example, using the fluorescent CDK4/6 sensor developed by Yang et al (eLife, 2020) in a Cdkn1c shRNA condition would represent an elegant experimental alternative to complement our rescue experiments with the double Cdkn1c/CCDN1 shRNA. However, we fear that setting up and calibrating such a tool for in vivo usage in the chick embryo represents too much of a challenge for incorporation in this study.

4) “Whether some of the main findings could be replicated in another species, for example, mouse or human to examine whether the mechanism is conserved? Could approaches other than image analysis be used to reinforce findings, for example biochemical methods, RNAseq or FACS?”

We agree that it will be interesting and important that our findings are replicated in other species, experimental systems, and even tissues, or by alternative experimental approaches. Nevertheless, it is probably beyond the scope of this study.

Dear Dr. Morin,

Thank you for the submission of your manuscript and your point-by-point response and revision plan to EMBO reports.

I have looked at all files now and think that your study is interesting and well done. It is my pleasure to invite you to revise it along the lines you and the referees suggest for publication by EMBO reports.

Please note that the referee concerns must be fully addressed and their suggestions taken on board. Please address all referee concerns in a complete point-by-point response. Acceptance of the manuscript will depend on a positive outcome of a second round of review. It is EMBO reports policy to allow a single round of major revision only and acceptance or rejection of the manuscript will therefore depend on the completeness of your responses included in the next, final version of the manuscript.

We realize that it is difficult to revise to a specific deadline. In the interest of protecting the conceptual advance provided by the work, we recommend a revision within 3 months (25th Jul 2025). Please discuss the revision progress ahead of this time with the editor if you require more time to complete the revisions.

- 1) A data availability section providing access to data deposited in public databases is missing. If you have not deposited any data, please add a sentence to the data availability section that explains that.
- 2) Your manuscript contains statistics and error bars based on $n=2$. Please use scatter blots in these cases. No statistics should be calculated if $n=2$.

5) a complete author checklist, which you can download from our author guidelines . Please insert information in the checklist that is also reflected in the manuscript. The completed author checklist will also be part of the RPF.

6) Please note that all corresponding authors are required to supply an ORCID ID for their name upon submission of a revised manuscript (). Please find instructions on how to link your ORCID ID to your account in our manuscript tracking system in our Author guidelines

- the name of the statistical test used to generate error bars and P values,
- the number (n) of independent experiments (please specify technical or biological replicates) underlying each data point,
- the nature of the bars and error bars (s.d., s.e.m.),
- If the data are obtained from n Program fragment delivered error ``Can't locate object method "less" via package "than" (perhaps you forgot to load "than"?) at //ejpvfs23/sites23b/embor_www/letters/embor_decision_rc_revise_and_rereview.txt line 56.' 2, use scatter blots showing the individual data points.

12) All Materials and Methods need to be described in the main text using our 'Structured Methods' format, which is required for all research articles. According to this format, the Methods section includes a Reagents and Tools Table (listing key reagents, experimental models, software and relevant equipment and including their sources and relevant identifiers) followed by a Methods and Protocols section describing the methods using a step-by-step protocol format. The aim is to facilitate adoption of the methodologies across labs. More information on how to adhere to this format as well as a downloadable template (.docx) for the Reagents and Tools Table can be found in our author guidelines:

An example of a Method paper with Structured Methods can be found here: <https://www.embopress.org/doi/full/10.1038/s44320-024-00037-6#sec-4>

I look forward to seeing a revised form of your manuscript when it is ready.

Kind regards,
Esther

Reviewer#1 (Evidence, reproducibility and clarity (Required)):

SUMMARY

This study utilizes the developing chicken neural tube to assess the regulation of the balance between proliferative and neurogenic divisions in the vertebrate CNS. Using single-cell RNAseq and endogenous protein tagging, the authors identify Cdkn1c as a potential regulator of the transition towards neurogenic divisions. Cdkn1c knockdown and overexpression experiments suggest that low Cdkn1c expression enhances neurogenic divisions. Using a combination of clonal analysis and sequential knockdown, the authors find that Cdkn1c lengthens the G1 phase of the cell cycle via inhibition of cyclinD1. This study represents a significant advance in understanding how cells can transition between proliferative and asymmetric modes of division, the complex and varying roles of cycle regulators, and provides technical advance through innovative combination of existing tools.

MAJOR AND MINOR COMMENTS

Overall

♣ Sample numbers are missing or unclear throughout for all imaging experiments. The authors should add numbers of cells analysed and/or numbers of embryos for their results to be appropriately convincing.

This information is now provided in the figure legends (numbers of cells analysed and/or numbers of embryos) except for data in Figure 5A, 5D, 6B and 6C, for which this information is provided in a new Supplementary Table 1

♣ Values and error bars on graphs must be defined throughout. Are the values means and error bars SD or SEM?

We have used SD throughout the study. This information has now been added in figure legends.

Results 2

♣ A reference should be provided for cell type distribution in spinal neural tube, where the authors state that cell bodies of progenitors reside within the ventricular zone.

We now cite a recent review on spinal cord development (Saade and E. Marti, Nature Reviews Neuroscience, 2025) to illustrate this point (p.7, line 171).

♣ The authors state that Cdkn1c "was expressed at low levels in a salt and pepper fashion in the ventricular zone, where the cell bodies of neural progenitors reside, and markedly increased in a domain immediately adjacent to this zone which is enriched in nascent neurons on their way to the mantle zone. In contrast, the transcript was completely excluded from the mantle zone, where HuC/D positive mature neurons accumulate." It is not clear if this is referring only to E4 or also to E3 embryos. Indeed, Cdkn1c expression appears to be much more salt and pepper at E3 and only resolves into a clear domain of high expression adjacent to the mantle zone at E4. It may be helpful

if this expression pattern could be described in a bit more detail highlighting the changes that occur between E3 and E4.

We have now reformulated this paragraph as follows (p7, lines 169-176):

“At E3, the transcript was expressed at low levels in a salt and pepper fashion in the ventricular zone, where the cell bodies of neural progenitors reside (Saade & Martí, 2025). One day later, at E4, this salt and pepper expression was still detected in the ventricular zone, while it markedly increased in the region of the mantle zone that is immediately adjacent to the ventricular zone. This region is enriched in nascent neurons on their way to differentiation that are still HuC/D negative. In contrast, the transcript was completely excluded from the more basal region of the mantle zone, where mature HuC/D positive neurons accumulate”

♣ *It would be useful to annotate the ISH images in Fig 2A to show the ventricular and mantle zones as defined by immunofluorescence.*

Thank you for the suggestion. We have now added a dotted line that separates the ventricular zone from the mantle zone at E3 and E4 in Figure 2A.

♣ *Reference should be included for pRb expression dynamics.*

This section has been rewritten and now contains several references regarding pRb expression dynamics (p8, lines 202-205).

“To ascertain that Cdkn1c is translated in neural progenitors, we used an anti-pRb antibody, recognizing a phosphorylated form of the Retinoblastoma (Rb) protein that is specifically detected in cycling cells (Gookin et al, 2017; Moser et al, 2018; Spencer et al, 2013), including neural progenitors of the developing chick spinal cord (Molina et al, 2022). »

♣ *Could the Myc tag insertion approach disrupt protein function or turnover?*

♣ *Why was the insertion target site at the C terminus chosen?*

The first reason was practical: at the time when we decided to generate a KI in Cdkn1c, we had already generated several successful KIs at C-termini of other genes, in particular using the P2A-Gal4 approach (see Petit-Vargas et al, 2024), and had not yet experimented with N-terminal Gal4-P2A. We therefore decided to use the same approach for Cdkn1c.

We also chose to target the C-terminus to avoid affecting the active CKI domain which is located at the N-terminus.

Nevertheless, the C-terminal targeting may have an impact on the turnover: it has been described that CDK2 phosphorylation of a Threonine close to the C-terminus of Cdkn1c leads to its targeting for degradation by the proteasome from late G1 (Kamura et al, PNAS, 2003; doi: 10.1073/pnas.1831009100). We can therefore not rule out that the addition of the Myc tags close to this phosphorylation site modulates the dynamics of Cdkn1c degradation. We note, however, that we observed little overlap between the Cdkn1c-Myc and pRb signals in Gal4/UAS-GFP-positive cycling progenitors, as shown in our new quantification of the Myc signal (New Figure 2E and Figure EV 1F). This suggests that Cdkn1c is effectively degraded from late G1.

♣ OPTIONAL Could a similar approach be used to tag *Cdkn1c* with a fluorescent protein to enable live imaging of dynamics?

Although it could be done, we have not attempted to do this for *Cdkn1c* because our current experience of endogenous tagging of several genes with a similar expression level (based on our scRNAseq data) and nuclear localization (e.g. Pax7) with a fluorescent reporter shows that the fluorescent signal is extremely low or undetectable in live conditions; therefore we favored the multi-Myc tagging approach, and indeed we find that the Myc signal in progenitors is also very low even though it is amplified by the immunohistology method; this suggests that most likely, the only signal that would be detected -if any- with a fluorescent approach would be the peak of expression in newborn neurons.

♣ In suppl Fig 1C nlsGFP-positive cells are shown in the control shRNA condition. How can this be explained and does it impact the interpretation of the findings?

The reviewer refers to the **control gRNA** condition in panel C, that shows that two small patches of GFP-positive cells are visible in the whole spinal cord of this particular embryo.

Technically, the origin of these “background” cells could be multiple. A spontaneous legitimate insertion at the *Cdkn1c* locus by homologous recombination is possible, although we tend to think it is unlikely, given the extremely short length of the arms of homology; illegitimate insertions and expression of the Myc-P2A-Gal4 cassette at an off-target site of the control gRNA is a possibility. Alternatively, a low-level leakage of Gal4 expression from the donor vector could lead to a detectable nls-GFP expression in a few cells via Gal4-UAS amplification.

In any case, these cells are observed at a very low frequency (1 or 2 patches of cells/embryo) relative to the signal obtained in presence of the *Cdkn1c* gRNA#1 and gRNA#3 (probably several thousand positive cells per embryo). This suggests that if similar “background” cells are also present in presence of the *Cdkn1c* gRNAs, they would not significantly contribute to the signal, and would not impact the interpretation.

♣ In Fig 2B, there are a number of Myc labelled cells in the mantle zone, whereas the in situ images show no appreciable transcript expression. Is this because the protein but not the transcript is present in these cells? Could the authors comment on this?

It is indeed possible that the *Cdkn1c* protein is more stable than the transcript in newborn neurons and remains detectable in the mantle zone after the mRNA disappears. In Gui et al, 2006 (Figure 1B) where they use an anti- *Cdkn1c* antibody to label the protein in **mouse** spinal cord transverse sections at E11.5, a few positive cells are also visible basally. They could correspond to neurons that have not yet degraded *Cdkn1c*, although it is unclear in the picture whether these cells are really in the mantle zone or in the adjacent dorsal root ganglion; we note that a similar differential expression dynamics between mRNA and protein has been described for *Tis21/Btg2* in the developing mouse cortex, where the protein, but not the mRNA, is detected in some differentiated bIII-tubulin-positive neurons (Iacopetti et al, 1999). However, related to our response above to a previous comment from the same reviewer, we cannot entirely rule out the possibility that the Myc tags modulate the turnover of *Cdkn1c* protein and slow down the dynamics of its degradation in differentiating neurons.

We have added a sentence to indicate the presence of these cells (p9, lines 227-229):

“In addition, a few Myc-positive cells were located deeper in the mantle zone, where the transcript is no more present, suggesting that the protein is more stable than the transcript.”

Results 3

♣ It should be mentioned how mRNA expression levels were quantified in the shRNA validation experiment (supp Fig 2A).

In the original manuscript, we had not quantified the level of mRNA reduction, it was just evaluated by eye. The reason for choosing shRNA1 for the whole study was dictated by 1) the fact that we more consistently saw (by eye) a reduction in the signal on the electroporated side with this construct than with the other shRNAs, and 2) that the effect on neurogenesis was also more consistent.

We have now performed additional experiments to provide some quantitation of the shRNA effect, as this is also requested by Reviewer#3.

As our Cdkn1c KI approach offers a direct read-out of the protein levels in the ventricular and mantle zones, and since our shRNA strategy of “partial knock-down” is based on the idea that the shRNA effect should be more complete in progenitors expressing Cdkn1c at low levels than in newborn progenitors that express the protein at a higher level, we have now performed complementary experiments to validate the downregulation mediated by the shRNA in the Cdkn1c-Myc knock-in background. We compared the Myc signal intensity in Cdkn1c-expressing knock-in cells (revealed via Gal4/UAS-iRFP driver/reporter positivity) between control and Cdkn1c shRNA conditions. We pooled individual cells’ intensity in bins of 25µm width along the apical-basal axis, and find that the Myc signal is virtually completely abolished in the apical-most 0-25 and 25-50µm bins, where low-expression progenitors reside, whereas it is strongly reduced, but still highly expressed in the more basal regions, where newborn neurons are located.

These experiments are illustrated in Figure EV 4B-C and described in a dedicated paragraph in the main text (p10, lines 254-258):

“We investigated the effect of shRNA1 at a cellular resolution by combining it with the knock-in approach described above. Quantification of Cdkn1c-Myc signal in comparison to a control shRNA confirmed a total silencing in progenitors and reduced but persistent expression in nascent neurons (Figure EV 4B-C). Of note, this approach further validates the specificity of the Myc signal in our knock-in approach.”

♣ Figure panels are not currently cited in order. Citation or figure order could be changed.

We have now added a common citation of the panels referring to analyses at 24 and 48 hours after electroporation (now Figure 3A-F), allowing us to display the experimental data on the figure according to the timing post electroporation, while the text details the phenotype at the later time point first (p10, line 264).

“Neuron and progenitor populations were evaluated 24 or 48 hours after electroporation (hae) via immunofluorescence (Figures 3A-F, see Methods for the choice of the markers of these populations)”.

♣ The authors should provide representative images for the graphs shown in Fig 3A and 3B. These could go into supplementary if the authors prefer.

We have added images in a revised version of the Figure 3, as requested. In addition, in response to a comment by reviewer# 3, we have split the graphs to present the pRb and HuC/D data separately.

♣ A supplementary figure showing the Caspase3 experiment should be added.

We have added data showing Caspase3 experiments in Figure EV 4F.

♣ OPTIONAL. Identification of sister cells in the clonal analysis experiments is based on static images and cannot be guaranteed. Could live imaging be used to watch divisions followed by fixation and immunostaining to confirm identity?

We agree with the reviewer that direct tracking is the most direct method for the identification of pairs of sister cells. However, it remains technically challenging and extremely labor-intensive, especially considering the many different experimental conditions that we have explored in this study. Direct tracking is especially valuable when trying to correlate a mitotic event (e.g. an asymmetry) with the daughter cell fate, but the added value compared to the retrospective identification would be limited in the context of this analysis.

Results 4

♣ How did the authors quantify the intensity of endogenous Myc-tagged Cdkn1c to confirm the validity of the Pax7 locus knock in? Can they show that the expression level was consistently lower than the endogenous expression in neurons? Quantification and sample numbers should be shown.

We have added a quantification of the signal intensity in the ventricular and mantle zones in the revised version of the manuscript, as also requested by reviewer#3 (p12, 325-328)

“Importantly, Myc signal intensity was similar to the one observed for endogenous Myc-tagged Cdkn1c in progenitors, and remained below the endogenous level of Myc-tagged Cdkn1c observed in nascent neurons, confirming the validity of our strategy (Figure 4C and Figure EV 7H-I).”

♣ In Fig 4B, the brightness of row 2 column 1 is lower than the same image in row 2 column 2, which is slightly misleading, since it makes the misexpressed expression level look lower than it is compared with endogenous in column 3. Is this because only a single z-section is being displayed in the zoomed in image? If so, this should be stated in the figure legend.

All images in the figure are single Z confocal images. Images in Column 2 (showing both electroporated sides of the same tube) were acquired with a 20x objective, whereas the insets shown in Columns 1 and 3 are 100x confocal images. 100x images on both sides were acquired with the same acquisition parameters, and the display parameters are the same for both images in the figure. The signal intensity can therefore be compared directly between columns 1 and 3.

We have modified the legend of the figure (now Figure 4C) to indicate these points:

“The insets shown in Columns 1 and 3 are 100x confocal images acquired in the same section and are presented with the same display parameters”.

♣ ***In Fig 4D, the increase in neurogenic divisions is mainly because of the rise in terminal NN divisions according to the graph, but no clear increase in PN divisions. Could the authors comment on the significance of this?***

Our interpretation is that Pax7-Cdkn1c misexpression experiments cause both PP to PN and PN to NN conversions. This is coherent with the classical idea of a progressive transition between these three modes of division in the spinal cord. Coincidentally, in our experimental conditions (timing of analysis and level of overexpression), the increase in PN resulting from PP to PN conversions is perfectly balanced by a decrease resulting from PN to NN conversions, giving the artificial impression that the PN compartment is unaffected. A less likely hypothesis would be that misexpression directly transforms symmetric PP into symmetric NN divisions, and that asymmetric PN divisions are insensitive to Cdkn1c levels. We do not favor this hypothesis, because one would expect, in that case, that the shRNA approach would also not affect the PN compartment, and it is not what we have observed (see Figure 3H – previously 3F).

We have modified the manuscript to elaborate on this result (p13, lines 346-353)

“Overall, this suggests that Cdkn1c premature expression in PP progenitors converts them to the PN mode of division, while the combined endogenous and Pax7-driven expression of Cdkn1c converts PN progenitors to the NN mode of division. Coincidentally, at the stage analyzed, PP to PN conversions are balanced by PN to NN conversions, leaving the PN proportion artificially unchanged in our quantification (Figure 4F). The alternative interpretation of a direct conversion of symmetric PP into symmetric NN divisions is less likely, because the PN compartment was affected in the reciprocal Cdkn1c shRNA approach (see Figure 3H)”.

Results 5

♣ ***The proportion of pRb-positive progenitors having entered S phase was stated to be higher at all time points; however, it is not significantly higher until 6h30 and is actually trending lower at 2h30.***

Thank you for pointing this out. We have modified the sentence in the main text:

“We found that the proportion of pRb positive progenitors having entered S phase (EdU positive cells) was significantly higher at all time points examined more than 4h30 after FT injection in the Cdkn1c knock-down condition compared to the control population (Figure 5D)”

♣ ***OPTIONAL Could CyclinD1 activity be directly assessed?***

This is an interesting suggestion. For example, using the fluorescent CDK4/6 sensor developed by Yang et al (eLife, 2020; <https://doi.org/10.7554/eLife.44571>) in a Cdkn1c shRNA condition would have been an elegant experimental alternative to complement our rescue experiments with the double Cdkn1c/CyclinD1 shRNA. However, setting up and calibrating such a tool for in vivo usage in the chick embryo represents too much of a challenge for incorporation in this study.

General

♣ ***Scale bars missing fig s1c s4d.***

Thanks for pointing this out. Scale bars have been added in the figures and corresponding legends

♣ **OPTIONAL** *Some of the main findings be replicated in another species, for example, mouse or human to examine whether the mechanism is conserved.*

♣ **OPTIONAL** *Could use approaches other than image analysis be used to reinforce findings, for example biochemical methods, RNAseq or FACS?*

We agree that it will be interesting and important that our findings are replicated in other species, experimental systems, and even tissues, or by alternative experimental approaches. Nevertheless, it is probably beyond the scope of this study.

♣ **A model cartoon to summarise outcomes would be useful.**

We thank the reviewer for the suggestion. We have added a summary cartoon in Figure 6F.

♣ **Unclear how cells were determined to be positive or negative for a label. Was this decided by eye? If so, how did the authors ensure that this was unbiased?**

For most experiments, positivity or negativity for markers was decided by eye. However, for each experiment, we ensured that all images of perturbed conditions and the relevant controls were analyzed with the same display parameters and by the same experimenter to guarantee that the criteria to determine positivity or negativity were constant.

One exception is the new characterization of Myc expression in the Cdkn1c knock-in relationship with pRb status with gRNA#1 (Figure 2E) and gRNA#3 (Figure EV 1F), where pRb signal intensity was first measured in all cells, and a threshold was used a posteriori to define pRb+ (above threshold) or pRb- (below threshold) status. This is detailed in the Methods section.

Reviewer#1 (Significance (Required)):

SIGNIFICANCE

Strengths:

This manuscript investigates the mechanisms regulating the switch from symmetric proliferative divisions to neurogenic division during vertebrate neuronal differentiation. This is a question of fundamental importance, the answer to which has eluded us so far. As such, the findings presented here are of significant value to the neurogenesis community and will be of broad interest to those interested in cell divisions and asymmetric cell fate acquisition. Specific strengths include:

♣ *Variety of approaches used to manipulate and observe individual cell behaviour within a physiological context.*

♣ *A limitation of using the chicken embryo is the lack of available antibodies for immunostaining. The authors take advantage of recent advances in chicken embryo CRISPR strategy to endogenously tag the target protein with Myc, to facilitate immunostaining.*

♣ *Innovative combination of genetic and labelling tools to target cells, for example, use of*

FlashTag and EdU in combination to more accurately assess G1 length than the more commonly used method.

♣ *Premature misexpression demonstrates that the previously observed dynamics indeed regulate cell fate.*

♣ *Mechanistic insight by examining downstream target CyclinD1.*

♣ *Clearly presented with useful illustrations throughout.*

♣ *Logic is clear and examination thorough.*

♣ *Conclusions are warranted on the basis of their findings.*

Limitations

♣ *This study primarily used visual analysis of fixed tissue images to assess the main outcomes. To reinforce the conclusions, these could be supplemented with live imaging to appreciate dynamics, or biochemical techniques to look at protein expression levels.*

♣ *Some aspects of quantification require explanation in order for the experiments to be replicated.*

♣ *It is imperative that precise sample sizes are included for all experiments presented.*

Advance:

♣ *First functional demonstration role for Cdkn1c in regulating neurogenic transition in progenitors.*

♣ *Conceptual advance suggesting Cdkn1c has dual roles in driving neurogenesis: promoting neurogenic divisions of progenitors and the established role of mediating cell cycle exit previously reported.*

♣ *Technical advances in the form of G1 signposting and endogenous Myc tagging using CRISPR in chicken embryonic tissue.*

Audience:

Of broad interest to developmental biologists. Could be relevant to cancer, since Cdkn1c is implicated.

♣ *Please define your field of expertise with a few keywords to help the authors contextualize your point*

Developmental biology, vertebrate embryonic development, neuronal differentiation, imaging. Please note that we have not commented on RNAseq experiments as these are outside of our area of expertise

Reviewer#2 (Evidence, reproducibility and clarity (Required)):

The work by Mida and colleagues addresses important questions about neurogenesis in the embryo, using the chicken neural tube as their model system. The authors investigate the mechanisms involved in the transition from stem cell self-renewal to neurogenic progenitor divisions, using a combination of single cell, gene functional and tracing studies.

The authors generated a new single cell data set from the embryonic chicken spinal cord and identify a transitory cell population undergoing neuronal differentiation, which expresses Tis21, Neurog2 and Cdkn1c amongst other genes. They then study the role of Cdkn1c and investigate the hypothesis that it plays a dual role in spinal cord neurogenesis: low levels favour transition from proliferative to neurogenic divisions and high levels drive cell cycle exit and neuronal differentiation.

Major comments

I have only a general comment related to the main point of the paper. The authors claim that Cdkn1c onset in cycling progenitor drives transition towards neurogenic modes of division, which is different from its role in cell cycle exit and differentiation. Figures 3F and 4D are key figures where the authors analysed PP, PN and NN mode of divisions via flash tag followed by analysis of sister cell fate. If their assumption is correct, shouldn't they also see, for example in Fig. 4D, an increase in PN or is this too transient to be observed or is it bypassed?

This is indeed an important point, also raised by reviewer#1. Our interpretation is that Pax7-Cdkn1c misexpression experiments cause both PP to PN and PN to NN conversions. This is coherent with the classical idea of a progressive transition between these three modes of division in the spinal cord. Coincidentally, in our experimental conditions (timing of analysis and level of overexpression), the increase in PN resulting from PP to PN conversions is perfectly balanced by a decrease resulting from PN to NN conversions, giving the artificial impression that the PN compartment is unaffected. A less likely hypothesis would be that misexpression directly transforms symmetric PP into symmetric NN divisions, and that asymmetric PN divisions are insensitive to Cdkn1c levels. We do not favor this hypothesis, because one would expect, in that case, that the shRNA approach would also not affect the PN compartment, and it is not what we have observed (see Figure 3H - previously 3F).

At the moment, the calculations of PN and NN frequencies are merged in the text, so perhaps describing PN and NN numbers separately will help better understand the dynamics of this gradual process (especially since there is little to no difference in PN).

Regarding the results of Pax7-driven Cdkn1c overexpression presented in Figure 4D (now Figure 4E in the revised version), we had made the choice to merge PN and NN values in the main text to focus on the neurogenic transition from PP to PN/NN collectively. We agree with this reviewer, as well as with reviewer#1, that it should be more detailed and better discussed.

We have modified the manuscript to elaborate on this result (and incorporate new results in response to a comment by reviewer#3) (p13, lines 337-353):

***“We observed an increase in the proportion of terminal neurogenic (NN) divisions and a decrease in proliferative (PP) divisions (Figure 4F). As above, the P and N identities in this analysis are based on our model of a plateau of pRb 6 hours after mitosis at that stage (Figure Figure EV 2). In parallel, to rule out a possible misinterpretation of daughter cells' identity resulting from a delay in reaching the Rb phosphorylation plateau upon Cdkn1c overexpression, we monitored HuC/D+ expression in a synchronous cohort of newborn cells 20hrs after FlashTag injection (Figure 4G, Figure EV 8B). The observed increase in HuC/D positive neurons confirms an increase in neurogenic divisions.*”**

Overall, this suggests that *Cdkn1c* premature expression in PP progenitors converts them to the PN mode of division, while the combined endogenous and Pax7-driven expression of *Cdkn1c* converts PN progenitors to the NN mode of division. Coincidentally, at the stage analyzed, PP to PN conversions are balanced by PN to NN conversions, leaving the PN proportion artificially unchanged in our quantification (Figure 4F). The alternative interpretation of a direct conversion of symmetric PP into symmetric NN divisions is less likely, because the PN compartment was affected in the reciprocal *Cdkn1c* shRNA approach (see Figure 3H).”

Could the increase in NN be compatible also with a role in cell cycle exit and differentiation, for example from cells that have been targeted and are still undergoing the last division (hence marked by flash tag) or there won't be any GFP cells marked by flash tag a day after expression of high levels of *Cdkn1c*?

It is likely that a proportion of cells that would normally have done a NN division are pushed to a direct differentiation that bypasses their last division in the Pax7-Cdkn1c condition, and that they contribute to the general increase in neuron production observed in our quantification 48hae (Figure 3F -previously 3C). However, these cases would not contribute to the increase in the NN quantification in pairs of sister cells 6 hours after division at 24hae (Figure 4F – previously 4D), because by design they would not incorporate FlashTag. The rise in NN is therefore the result of a PN to NN conversion.

Basically, what would the effect of expressing higher levels of *Cdkn1c* be? I guess this will really help them distinguish between transition to neurogenic division rather than neuronal differentiation. If not experimentally, any further comments on this would be appreciated.

These experiments have been performed and presented in the study by Gui et al., 2007, which we cite in the paper. Using a strong overexpression of *Cdkn1c* from the CAGGS promoter, they showed a massive decrease in proliferation, assessed by BrdU incorporation, 24hours after electroporation. We now cite this result more explicitly in the main text, and better explain the difference of our approach. We propose the following modification (p12, lines 307-316):

« We next explored whether low *Cdkn1c* activity is sufficient to induce the transition to neurogenic modes of division. A previous study has shown that a massive overexpression of *Cdkn1c* driven by the strong CAGGS promoter triggers cell cycle exit of chick spinal cord progenitors, revealed by a drastic loss of BrdU incorporation 1 day after electroporation (Gui et al, 2007). As this precludes the exploration of our hypothesis, we developed an alternative approach designed to prematurely induce a pulse of *Cdkn1c* in progenitors, with the aim to emulate in proliferative progenitors the modest level of expression observed in neurogenic progenitors (Figure 4A). We took advantage of the Pax7 locus, which is expressed in progenitors in the dorsal domain (Jostes et al, 1990) at a level similar to that observed for *Cdkn1c* in neurogenic precursors (Figure EV 7A). »

Minor comments

Fig 3C my understanding is that HuC/D should be nuclear, but in fig 3C it seems more cytoplasmic (any comment?)

Some studies suggest that HuC/D can, under certain conditions, be observed in the nucleus of neurons. However, HuC/D is a RNA binding protein whose localization is mainly expected to be cytoplasmic. In our experience (Tozer et al, 2017), and in other publications using the same antibody in the chick spinal cord (see, for example, le Dreau et al, 2014), it is observed in the cell body of differentiated neurons, as in the current manuscript.

Fig Suppl 3E (and related 4B), immuno for Cdkn1c-Myc: to help the reader understand the difference between the immuno signals when looking at the figure, I would suggest writing on the panel i) Pax7-Cdkn1c-Myc and ii) endogenous Cdkn1c-Myc, rather than 'misexpressed' and 'endogenous', which is slightly confusing (especially because what it is called endogenous expression is higher).

This has now been modified in the figures.

Literature citing: Introduction and discussion are very nicely written, although they could benefit from some more recent literature on the topic. For example, Cdkn1c role as a gatekeeper of stem cell reserve in the stomach, gut, (Lee et al, CellStemCell 2022 PMID: 35523142) or some other work on symmetric/asymmetric divisions and clonal analysis in zebrafish (Hevia et al, CellRep 2022 PMID: 35675784, Alexandre et al, NatNeur PMID: 20453852), mammals (Royal et al, Elife 2023 37882444, Appiah et al, EMBO rep 2023 PMID: 37382163). Also, similar work has been performed in the developing pancreatic epithelium, where mild expression of Cdkn1a under Sox9rtTa control was used to lengthen G1 without overt cell cycle exit and this resulted in Neurog3 stabilization and priming for endocrine differentiation (Krentz et al, DevCell 2017 PMID: 28441528), so similar mechanisms might be in place to gradually shift progenitor towards stable decision to differentiate.

Moreover, in the discussion, alongside Neurog2 control of Cdkn1c, it could be mentioned that the feedback loop between Cdk inhibitors and neurogenic factor is usually established via Cdk inhibitor-mediated inhibition of proneural bHLHs phosphorylation by CDKs (Krentz et al, DevCell 2017 PMID: 28441528, Ali et al, 24821983, Azzarelli et al 2017 - PMID: 28457793; 2024 - PMID:39575884).

We thank the reviewer for these suggestions, and for pointing out several important studies that were not mentioned in the original manuscript.

We have now modified several paragraphs in the introduction and discussion sections to include most of the suggested references:

In the introduction (p3, lines 57-66):

“By analysing the fate of pairs of sister cells, clonal analysis approaches in various models have helped to identify and quantify these different modes of division in different wild type and mutant contexts (Alexandre et al, 2010; Appiah et al, 2023; Hevia et al, 2022; Morin et al, 2007; Royall et al, 2023; Tozer et al, 2017). These studies have contributed to establish that the balance between these different modes of division is a crucial regulator of the pace of neurogenesis and of the relative size of different neuronal pools. In addition, analysis of the content of clones over longer periods of time in the mouse and rat embryonic cortices indicate that progenitors that have undergone a neurogenic division do not normally reenter a proliferative state, suggesting an irreversible switch in competence (Gao et al, 2014; Noctor et al, 2004). »

In the discussion, which has been largely reorganized, we have modified the paragraph concerning the feedback loop to incorporate the suggested references (p18, lines 494-504):

“Once its expression is initiated, Cdkn1c may take part in a feedback loop with NeuroG2 leading to their concomitant upregulation during neurogenesis, such as what has been described in other contexts. Indeed, studies in Xenopus primary neurogenesis and in mammalian pancreatic progenitors have shown that Cdkn1c-related CDK inhibitors favor the stability of NeuroG2 and NeuroG3 proteins. Mechanistically, this involves their inhibition of CDK/cyclin dependent phosphorylation of NeuroG2/3, which targets them for degradation (Ali et al, 2014; Azzarelli et al, 2017; Krentz et al, 2017; Vernon et al, 2003). Additionally, in the mouse developing cortex, the closely related p27^{Kip1}/Cdkn1b was shown to regulate NeuroG2 protein stability and activity in a cell cycle and CDK/Cyclin independent manner (Nguyen et al, 2006). Possibly reinforcing the feedback loop, NeuroG2 is also involved in the downregulation of cyclinD1 and cyclinE2 in spinal progenitors (Lacomme et al, 2012). »

Further, in the discussion, could they mention anything about the following open questions: is there evidence for Cdkn1c low/high expression in mammalian spinal cord? Or maybe of other Cdk inhibitors?

In the discussion, we suggest that the low-high expression in the spinal cord may be conserved in the mouse, but that it could have been missed due to insufficient sensitivity of the detection with the Cdkn1c antibody (in the study by Gui et al); the loss of function phenotypes would be compatible with this hypothesis (p 17, lines 467-476):

“In the mouse spinal cord, the higher proliferation observed in Cdkn1c knock-out embryos was attributed to a failure to exit the cell cycle in newborn neurons (Gui et al, 2007). In the light of our results in the chick spinal cord, the mouse phenotype could also in part be interpreted as a failure of proliferative progenitors to progress to the neurogenic state, although this study did not report the “low level” expression of Cdkn1c that we describe here in the neurogenic progenitor population, possibly due to a low sensitivity of the antibody. Supporting this idea, higher proliferation and a shortening of G1 phase have been described in cortical progenitors in Cdkn1c knock-out mice (Mairet-Coello et al, 2012). Altogether, this suggests a possible conservation of the dual role linked to the dynamics of expression of Cdkn1c in the embryonic mammalian CNS. “

Is Cdkn1c also involved in cell cycle exit during gliogenesis or is there another Cdk inhibitor expressed at later developmental stages, hence linking this with specific cell fate decisions?

We mention in the introduction that Cdkn1c is also involved in gliogenesis at later stages. In addition, p27 and, more recently, p18 (a member of the INK family of CDK inhibitors), have also been involved in fate choices during gliogenesis (e.g. PMID: **39668249**), but we do not expand on this subject in the manuscript (p4, lines 77-87):

“Cdkn1c (Cyclin-dependent kinase inhibitor 1c/p57^{Kip2}), a member of the CDK inhibitor (CKI) family that also comprises Cdkn1a (p21cip1) and Cdkn1b (p27kip1), emerged as one of the most differentially expressed between proliferative and neurogenic progenitors. In the mammalian cortex, p57^{Kip2} plays a key role in regulation of the proliferation and differentiation of embryonic and adult neural stem cells (Colasante et al, 2015; Furutachi et al, 2013; Imaizumi et al, 2020; Itoh et al, 2007; Joseph et al, 2009, 2003; Mairet-Coello et al, 2012; Tury et al, 2011) and it was shown to promote gliogenesis during late embryogenesis and early postnatal stages (Tury et al, 2011). While in the spinal cord, Cdkn1c was previously described as a regulator of cell cycle exit via its expression in newborn neurons (Gui et al., 2007; Mairet-Coello et al., 2012; Tury et al., 2011), our results showed that it is already expressed at low level in neurogenic progenitors.”

Reviewer#2 (Significance (Required)):

The work here presented has important implications on neural development and its disorders. The authors used the most advanced technologies to perform gene functional studies, such as CRISPR-HDR insertion of Myc-tag to follow endogenous expression, or expression under endogenous Pax7 promoter, often followed by flash tag experiments to trace sister cell fate, and all of this in an in vivo system. They then tested cell cycle parameters, clonal behaviour and modes of cell division in a very accurate way. Overall data are convincing and beautifully presented. The limitation is potentially in the resolution between the events of switching to neurogenic division versus neuronal differentiation, which might just warrant further discussion.

This work advances our knowledge on vertebrate neurogenesis, by investigating a key player in proliferation and differentiation.

I believe this work will be of general interest to developmental and cellular biologists in different fields. Because it addresses fundamental questions about the coordination between cell cycle and differentiation and fate decision making, some basic concepts can be translated to other tissues and other species, thus increasing the potential interested audience.

My work focuses on stem cell fate decisions in mammalian systems, and I am familiar with the molecular underpinnings of the work here presented. However, I am not an expert in the chicken spinal cord as a model and yet the manuscript was interesting. I am also not sufficiently expert in the bioinformatic analysis, so cannot comment on the technical aspects of Figure 1 and the way they decided to annotate their data.

Reviewer#3 (Evidence, reproducibility and clarity (Required)):

Summary:

In this study, Mida et al. analyze large-scale single-cell RNA-seq data from the chick embryonic neural tube and identify Cdkn1c as a key molecular regulator of the transition from proliferative to neurogenic cell divisions, marking the onset of neurogenesis in the developing CNS. To confirm this hypothesis, they employed classical techniques, including the quantification of neural cell-specific markers combined with the flashTAG label, to track and isolate isochronic cohorts of newborn cells in different division modes. Their findings reveal that Cdkn1c expression begins at low levels in neurogenic progenitors and becomes highly expressed in nascent neurons. Using a classical knockdown strategy based on short hairpin RNA (shRNA) interference, they demonstrate that Cdkn1c suppression promotes proliferative divisions, reducing neuron formation. Conversely, novel genetic manipulation techniques inducing low-level Cdkn1c misexpression drive progenitors into neurogenic divisions prematurely.

By employing cumulative EdU incorporation assays and shRNA-based loss-of-function approaches, Mida et al. further show that Cdkn1c extends the G1 phase by inhibiting cyclin D, ultimately concluding that Cdkn1c plays a dual role: first facilitating the transition of progenitors into neurogenic divisions at low expression levels, and later promoting cell cycle exit to ensure proper neural development.

This study presents several ambiguities and lacks precision in its analytical methodologies and quantification approaches, which contribute to confusion and potential bias. To enhance the reliability of the conclusions, a more rigorous validation of the methods employed is essential. This study introduces a novel approach to tracking the fate of sister cells from neural progenitor divisions to infer the division modes. While previous methods for analyzing the division mode of neural progenitor cells have been implemented, rigorous validation of the approach introduced by Mida et al. is necessary. Furthermore, the concept of cell cycle regulators interacting to control the duration of specific cell cycle stages and influencing progenitor cell division modes has been explored before, potentially limiting the novelty of these findings.

Majors comments:

1.-The study presents ambiguity and lacks precision in quantifying neural precursor division modes. The authors use phosphorylated retinoblastoma protein (pRb) as a marker for neurogenic progenitors, claiming its reliability in identifying neurogenic divisions.

However, they do not provide a thorough characterization of pRb expression in the developing chick neural tube, leaving its suitability as a neurogenic division marker unverified.

Throughout their comments on the manuscript, this reviewer raises several points regarding the characterization of pRb expression in our model and of our use of this marker in our study. We take these comments into account and propose to expand on pRb characteristics in the first occurrence of pRb as a marker of cycling cells in the manuscript. The modifications rely on:

- the quotation of several studies showing that phosphorylation of Rb is regulated during the cell cycle, in a revised version of the whole paragraph (p8-9, lines 202-218):

“To ascertain that Cdkn1c is translated in neural progenitors, we used an anti-pRb antibody, recognizing a phosphorylated form of the Retinoblastoma (Rb) protein that is specifically detected in cycling cells (Gookin et al, 2017; Moser et al, 2018; Spencer et al, 2013), including neural progenitors of the developing chick spinal cord (Molina et al, 2022). In the ventricular zone of transverse sections at E3.5 (33hae) (Figure 2D) and E4 (48hae) (Figure 2C and Figure EV1E), we detected triple Cdkn1c-Myc/GFP/pRb positive cells (arrowheads in Figure 2C-D), providing direct evidence for the Cdkn1c protein in cycling progenitors. We also observed many double GFP/pRb positive cells that were Myc negative (arrows in Figure 2C-D). The observation of UAS-driven GFP in these pRb-positive cells is evidence for the translation of Gal4 and therefore provides a complementary demonstration that the Cdkn1c transcript is translated in progenitors. The absence of Myc detection in these double GFP/pRb positive cells also suggests that Cdkn1c/Cdkn1c-Myc stability is regulated during the cell cycle.

Finally, we observed double Myc/GFP-positive cells that were pRb-negative (Figure 2C and Figure EV 1G; asterisks). One characteristic of Rb phosphorylation as a marker of cycling cells is a period in early G1 during which it is not detectable, as described in several cell types (Gookin et al, 2017; Moser et al, 2018; Spencer et al, 2013) including chick spinal cord neural progenitors (Molina et al, 2022).”

- a more detailed description of our own characterization of pRb dynamics in a synchronous cohort of cycling cells, which reveals a similar heterogeneity in the expression of pRb after mitosis. This description was initially shown in Supplementary Figure 3 and has now been transferred to a new Figure EV 2 to account for the fact that it is now presented earlier in the manuscript.

Regarding the specific question the **“suitability (of pRb) as a neurogenic division marker”**: we do not directly **“use phosphorylated retinoblastoma protein (pRb) as a marker for neurogenic progenitors”**, but we use the status of Rb phosphorylation to discriminate between progenitors (pRb+) and neurons (pRb-) identity in pairs of sister cells to retrospectively identify the mode of division of their mother.

Given that Rb is unphosphorylated during a period of variable length after mitosis (see references above), pRb is not a reliable marker of ALL cycling progenitors. We developed an assay to identify the timepoint (the maximal length of this “pRb-negative” phase) after which Rb is phosphorylated in all cycling progenitors (new Figure EV 2). This assay relies on a time course of pRb detection in cohorts of FlashTag-positive pairs of sister cells born at E3. This time course experiment allowed us to identify a plateau after which the proportion of pRb-positive cells in the cohort remains constant. From this timepoint, this proportion corresponds to the proportion of cycling cells in the cohort. In this specific context, Rb phosphorylation therefore becomes a discriminating factor between cycling progenitors (pRb+) and non-cycling neurons (pRb-).

We are confident that this provides a solid foundation for the determination of the identity of pairs of sister cells in our Flash-Tag based assays, which retrospectively identify the mode of division of a progenitor on the basis of the Rb phosphorylation status of its daughter cells 6 hours after division.

We have modified the main text to describe the strategy and protocol more explicitly, by introducing the sentence highlighted in yellow in the following paragraph where the paired-cell analysis is first introduced (in the section on *Cdkn1c* knock-down p11, lines 289-298):

“This approach allows to retrospectively deduce the mode of division used by the mother progenitor cell. We injected the cell permeant dye “FlashTag” (FT) at E3 to specifically label a cohort of progenitors that undergoes mitosis synchronously (Baek et al., 2018; Telley et al., 2016 and see Methods), and analyzed the fate of their progeny six hours later (Figure 3G to 3I). Our characterization of pRb immunoreactivity in the tissue had established beforehand that 6 hours after mitosis, all progenitors can reliably be detected with this marker (Figure EV 2, Methods). Therefore, at this timepoint after FT injection, two-cell clones selected on the basis of FT incorporation and similar GFP signal intensity can be categorized as PP, PN, or NN based on pRb positivity (P) or not (N) (see Methods, Figure 3I and Figures EV 2 and EV 5).”

We also modified accordingly the legend to Figure EV 2 (previously Supplementary Figure 3, which describes the identification of the plateau of pRb.

Furthermore, retinoblastoma protein (Rb) and cyclin D interact crucially to regulate the G1/S phase transition of the cell cycle, with cyclin D/CDK complexes phosphorylating Rb. Since the authors conclude that Cdkn1c primarily acts by inhibiting the cyclin D/CDK6 complex, it is likely that Cdkn1c influences pRb expression or phosphorylation state. This raises the possibility that pRb could be a direct target of Cdkn1c, whose expression and phosphorylation would be altered in gain-of-function (GOF) and loss-of-function (LOF) analyses of Cdkn1c.

In light of this, it would be more appropriate to consider pRb as a Cdkn1c target and discuss the molecular mechanisms regulating cell cycle components.

We agree with the reviewer that Rb phosphorylation may be a direct or indirect target of *Cdkn1c* activity, and exploring the molecular aspects of the cellular and developmental phenomena that we describe in our manuscript would represent an interesting follow up study.

A more precise approach would involve using other markers or targets to quantify neural precursor division modes at earlier stages of neurogenesis.

To complement our analyses of the modes of division, we have now added a new set of experiments using a positive marker to assess neural identity in parallel to the absence of pRb within pairs of cells. This approach is the most meaningful in the gain of function context (Pax7 driven expression of Cdkn1c) because in this context, the time-point to reach the plateau of Rb phosphorylation used in our FT-based assay may indeed be delayed through G1-phase elongation mediated by Cdkn1c overexpression. On the opposite, in the context of loss of functions, the plateau may be reached earlier, which would have no effect on this assay.

This new set of experiments is now described in the main text of the revised version (p13, lines 335-345) and illustrated by new panels in Figure 4G (quantification) and Figure EV 8B (representative images).

“We next examined whether the increased neurogenesis 48 hae is linked to a change in the mode of division of progenitors misexpressing Cdkn1c. Using the FlashTag cohort labeling approach described above, we traced the fate of daughter cells born 24 hae. We observed an increase in the proportion of terminal neurogenic (NN) divisions and a decrease in proliferative (PP) divisions (Figure 4F). As above, the P and N identities in this analysis are based on our model of a plateau of pRb 6 hours after mitosis at that stage (Figure EV 2). In parallel, to rule out a possible misinterpretation of daughter cells’ identity resulting from a delay in reaching the Rb phosphorylation plateau upon Cdkn1c overexpression, we monitored HuC/D+ expression in a synchronous cohort of newborn cells 20hrs after FlashTag injection (Figure 4G, Figure EV 8B). The observed increase in HuC/D positive neurons confirms an increase in neurogenic divisions»

2.-Furthermore, the study employs FlashTag labeling to track daughter cells post-division, but the 16-hour post-injection window may result in misidentification of sister cells due to the potential presence of FlashTagged cells that did not originate from the same division.

This introduces a risk of bias in quantification, data misinterpretation, and potential errors in defining division modes. A more rigorous validation of the FlashTag strategy and its specificity in tracking division pairs is necessary to ensure the reliability of their conclusions.

The reviewer probably mistyped and meant 6-hour post injection, which is the duration that we use for paired cell tracking. We would like to emphasize that in addition to the FlashTag label, we benefit from the electroporation reporter to assess clonality. Altogether, we combine 5 criteria to define a clonal relationship :

- 2 cells are positive for Flash Tag
- The Flash Tag intensity is similar between the 2 cells
- The 2 cells are positive for the electroporation reporter
- The electroporation reporter intensity is similar between the two cells
- the position of the two cells is consistent with the radial organization of clones in this tissue (Leber and Sanes, 1995; Loulier et al, 2014): they are found on a shared line along the apico-basal axis, and share the same Dorso-Ventral and Antero-Posterior position.

This combination is described in the Methods section (p31, line 992). We have modified the paragraph to include the sentence highlighted in yellow in the text below;

“Cell identity of transfected GFP positive cells was determined as follows: cells positive for pRb and FT were classified as progenitors and cells positive for FT and negative for pRb as neurons. In addition, a similar intensity of both the GFP and FT signals within pairs of cells, and a relative position of the two cells consistent with the radial organization of clones in this tissue (Leber and Sanes, 1995; Loulier et al, 2014) were used as criteria to further ascertain sisterhood. This combination restricts the density of events fulfilling all these independent criteria, and can confidently be used to ensure a robust identification of pairs of sister cells.”

Note that in the new set of experiments where we defined the fate of daughters with HuC/D labeling 20 hours after FlashTag injection, we did not attempt to identify “pairs of sister cells” as this timing is too long to reliably assess sisterhood, and only quantified the increase in the number of double HuC/D-FT positive neurons in the cohort.

3.- The knock-in strategy used to tag the endogenous Cdkn1c protein in Figure 2 is an elegant tool to infer protein dynamics in vivo. However, since strong conclusions regarding Cdkn1c dynamics during the cell cycle are drawn from this section, it would be advisable to strengthen the results by including quantification with adequate replication and proper statistical analysis, as the current findings are preliminary and somewhat speculative.

- "Although pRb is specific for cycling cells, it is only detected once cells have passed the point of restriction during the G1 phase." Please provide literary reference confirming this observation.

We have entirely remodeled this section, which describes the expression of Myc-tagged Cdkn1c relative to pRb and we now provide several references that describe the generally accepted view that pRb is specific of cycling cells, regulated during the cell cycle, and in particular absent in early G1. We also removed the mention of the “Restriction point” in the main text to avoid any confusion on the timing of phosphorylation, as the notion of restriction point is not useful in our study. The whole changes to this section are shown after the answer to the next few comments and questions on pRb and Myc dynamics.

Given that pRb immunoreactivity is used as a marker for cycling progenitors to base many of the results of this study, it would be very valuable to characterize the dynamics of pRb in cycling cells in the studied tissue, for instance combined with the cell cycle reporter used by Molina et al. (Development 2022).

In the original version of the manuscript, the section describing the dynamics of Cdkn1c-Myc in the KI experiments presented in Figure 2 relied on the idea that the dynamics of pRb in chick spinal progenitors is similar to what is described in other tissues and cell types, without providing any references to substantiate this fact. Actually, Molina et al provide a characterization of pRb in combination with their cell cycle reporter and conclude that pRb negative progenitors are in G1 (“We also verified that phospho-Rb- and HuC/D-negative cells were in G1 by using our FUCCI G1 and PCNA reporters”). We now cite this reference to support our claim. In addition, our characterization of Rb progressive phosphorylation in the synchronic Flash-Tag cohort of newborn sister cells provides a complementary demonstration that a fraction of the progenitors are pRb-negative when they exit mitosis (i.e. in early G1). This analysis was initially only introduced in the Supplementary Figure 3, as support for the section that presents the paired-cell assay used in Figure 3. We now introduce the data from Supplementary Figure 3 earlier in the manuscript (now Figure EV 2), in order to better introduce the reader with the dynamics of pRb in cycling cells in our model. This will better support our description of the Cdkn1c-Myc dynamics in relation with pRb. We have therefore reformulated this whole section as shown below.

- *It would be valuable to analyse the dynamics of Myc immunoreactivity in combination of pRb in all three gRNAs (highlighted in Supplementary Figure 1), as it would be a strong point in favour that the dynamics reflect the endogenous Cdkn1c dynamics.*

- *It would be very valuable to provide a quantification of said dynamics (e.g. plotting myc intensity / pRb immunoreactivity along the apicobasal axis of the tissue).*

These are two interesting suggestions. To complement our data with guide #1 (new Figure 2C-D), we have also performed Myc-immunostaining experiments on transverse sections in the context of guide #3 (Figure EV 1E), showing exactly the same pattern of Myc signal, with low expression in the VZ, and a peak of signal in the part of the mantle zone that is immediately touching the VZ. This confirms the specificity of the spatial distribution of the Cdkn1c-Myc signal. In addition, quantification of Cdkn1c-Myc signal along the apico-basal axis, and in pRb⁺ versus pRb⁻ cells, are now provided in Figure 2E (gRNA#1) and Figure EV 1F (gRNA#3).

We have performed the suggested quantifications using guides #1 and #3, which both show a good KI efficiency. We do not think it would have been useful to do these experiments with guide #2, whose efficiency is much lower, and which would have led to a very sparse signal.

The whole section of the manuscript that incorporates all these changes now reads as follows (p8, line 202 – p9, line 242):

“To ascertain that Cdkn1c is translated in neural progenitors, we used an anti-pRb antibody, recognizing a phosphorylated form of the Retinoblastoma (Rb) protein that is specifically detected in cycling cells (Gookin et al, 2017; Moser et al, 2018; Spencer et al, 2013), including neural progenitors of the developing chick spinal cord (Molina et al, 2022). In the ventricular zone of transverse sections at E3.5 (33hae) (Figure 2D) and E4 (48hae) (Figure 2C and Figure EV1E), we detected triple Cdkn1c-Myc/GFP/pRb positive cells (arrowheads in Figure 2C-D), providing direct evidence for the Cdkn1c protein in cycling progenitors. We also observed many double GFP/pRb positive cells that were Myc negative (arrows in Figure 2C-D). The observation of UAS-driven GFP in these pRb-positive cells is evidence for the translation of Gal4 and therefore provides a complementary demonstration that the Cdkn1c transcript is translated in progenitors. The absence of Myc detection in these double GFP/pRb positive cells also suggests that Cdkn1c/Cdkn1c-Myc stability is regulated during the cell cycle.

Finally, we observed double Myc/GFP-positive cells that were pRb-negative (Figure 2C and Figure EV 1G; asterisks). One characteristic of Rb phosphorylation as a marker of cycling cells is a period in early G1 during which it is not detectable, as described in several cell types (Gookin et al, 2017; Moser et al, 2018; Spencer et al, 2013) including chick spinal cord neural progenitors (Molina et al, 2022). Using a method that specifically labels a synchronous cohort of dividing cells in the neural tube, we similarly observed a period in early G1 during which pRb is not detectable in some progenitors at E3 (See Figure EV 2 and Methods). Hence, the double Myc/GFP positive and pRb negative cells may correspond to progenitors in early G1. Alternatively, they may be nascent neurons whose cell body has not yet translocated basally (see Figure 2F). Finally, we observed a pool of GFP positive/pRb negative nuclei with a strong Myc signal in the region of the mantle zone that is in direct contact with the ventricular zone (VZ), corresponding to the region where the transcript is most strongly detected (see Figure 2A). This pool of cells with a high Cdkn1c expression likely corresponds to immature neurons exiting the cell cycle and on their way to differentiation (Figure 2C-D; double asterisks). In addition, a few Myc positive cells were located deeper in the mantle zone, where the transcript is no more present, suggesting that the protein is more stable than the transcript. The quantification of the level of expression of Cdkn1c-Myc along the apico-basal axis in relationship to pRb immunoreactivity

confirms that *Cdkn1c* is expressed at low level in the ventricular zone in both pRb positive and negative cells, and that it peaks in more basal, pRb-negative cells (Figure 2E). Of note, similar results were obtained with gRNA#3 (Figure EV 1E-F).

In summary, our dual Myc and Gal4 knock-in strategy which reveals the history of *Cdkn1c* transcription and translation confirms that *Cdkn1c* is expressed at low level in a subset of progenitors in the chick spinal neural tube. In addition, the restricted overlap of *Cdkn1c*-Myc detection with Rb phosphorylation suggests that in progenitors, *Cdkn1c* is degraded or less transcribed close to and/or after G1 completion. Indeed, we detect an overall lower expression in S/G2/M compared to G1 phase in neurogenic progenitors in the scRNAseq analysis (Figure Figure EV 3). Overall, this may explain why a classical immunohistochemistry approach with an anti-*Cdkn1c* antibody only detected the protein in very few progenitors in the developing mouse CNS (Gui et al., 2007; Mairet-Coello et al., 2012). “

- The characterization of dynamics is performed only with one of the gRNAs (#1) on the basis that it produces the strongest NLS-GFP signal, as a proxy for guide efficiency. It would be nice if the authors could validate guide cutting efficiency via sequencing (e.g. using a Cas9-T2A-GFP plasmid and sorting for positive cells).

- In order to make sure that the dynamics inferred from Myc-tag immunoreactivity do reflect the cell cycle dynamics of *Cdkn1c*-myc, it would be advisable to confirm in-frame insertion of the myc-tag sequence.

We have attempted to amplify genomic DNA from the *Cdkn1c* locus in order to probe the guide cutting efficiency with the Tide method, and to confirm in-frame insertion of the Myc-tag via sequencing. Unfortunately, despite our best efforts, we did not manage to obtain any PCR bands, probably due to the high GC content of the *Cdkn1c* sequence. To circumvent this point, we used three alternative approaches:

- we generated a donor vector that lacked arms of homology to the *Cdkn1c* locus and used it in combination with gRNA#1 and #3. This gave no signal, confirming that the signal obtained with the KI construct derives from the arms of homology, and therefore of an insertion at the locus.
- We made western blot analyses using an anti-Myc antibody and protein extracts from control and KI embryos, which revealed a strong band with both gRNA#1 and #3, and no signal at all with a control gRNA, showing that the Myc signal derives from a single type of events, which is identical with both gRNAs.
- Finally, the silencing of the Myc signal in neural progenitors in the context of *Cdkn1c* downregulation by *Cdkn1c* shRNA1 further validates that the Myc signal in our knock-in approach derives from *Cdkn1c* expression.

The two first results are mentioned in the following new paragraph (p8, lines 197-201):

“ Specificity of insertion at the *Cdkn1c* locus was further confirmed by several approaches: first, we did not obtain any signal when a donor vector that lacked arms of homology to the *Cdkn1c* locus was used in combination with gRNA#1 and #3 (Figure EV 1C); second, western blot analyses using an anti-Myc antibody revealed a strong band with both gRNA#1 and #3, and no signal at all with a control gRNA (Figure EV 1D).”

The third one is mentioned in the paragraph that describes the shRNA approach (p10, lines 254-258):

"We investigated the effect of shRNA1 at a cellular resolution by combining it with the knock-in approach described above. Quantification of Cdkn1c-Myc signal in comparison to a control shRNA confirmed a total silencing in progenitors and reduced but persistent expression in nascent neurons (Figure EV 4B-C). Of note, this approach further validates the specificity of the Myc signal in our knock-in approach".

4.- In Figure 3, the authors use a short-hairpin-mediated knock-down strategy to decrease the levels of Cdkn1c, and show that this manipulation leads to an increase percentage of cycling progenitors and a decrease in the number of neurons in electroporated cells.

The authors claim that their shRNA-based knockdown strategy aims to reduce low-level Cdkn1c expression in neurogenic progenitors while minimally affecting the higher expression in newborn neurons required for cell cycle exit. However, several factors need consideration. Electroporation introduces variability in shRNA delivery, making it difficult to achieve consistent gene inhibition across all cells, especially for dose-dependent genes like Cdkn1c.

Additionally, Cdkn1c generates multiple isoforms, which may not be fully annotated in the chick genome, raising the possibility that the shRNA targets specific isoforms, potentially explaining the observed low expression.

All the predicted isoforms in the chick genome contain the sequence targeted by shRNA1, which is located in the CKI domain, the region of the protein that is most conserved between species. Besides, all the isoforms annotated in the mouse and human genomes also contain the region targeted by shRNA1. We are therefore confident that shRNA1 should target all chick isoforms.

A more rigorous approach, such as qPCR analysis of sorted electroporated cells, would better validate the expression levels, rather than relying on in situ hybridization, presenting electroporated and non-electroporated cells in the same section (Supp. Figure 2).

This approach (qRT-PCR on sorted cells) would enable us to focus solely on electroporated cells, but it would result in an averaged quantification of Cdkn1c depletion. In order to obtain additional information on the shRNA-dependent decrease in Cdkn1c in the different neural cell populations (progenitor versus differentiating neuron), we used an alternative approach consisting in monitoring the level of Cdkn1c protein, assessed through Cdkn1c-Myc signal in knock-in cells, in the presence versus absence of Cdkn1c shRNA. These results are now included in the main text, as mentioned in the response to the previous question (p10, lines 254-258).

- As the authors note, "Unambiguous identification of cycling progenitors and postmitotic neurons is notoriously difficult in the chick spinal cord". "markers of progenitors usually either do not label all the phases of the cell cycle (eg. Phospho-Rb, thereafter pRb), or persist transiently in newborn neurons (eg. Sox2)." Given that pRb immunoreactivity is used as the basis for a lot of the conclusions in this study, it would be valuable to add a characterization of its dynamics as mentioned in Figure 2, as well as provide literary references/proof that Sox2 expression persists in newborn neurons.

We have addressed the case of pRb dynamics in progenitors above and added a reference documenting pRb expression during the cell cycle of chick neural progenitors (Molina et al, 2022).

Regarding Sox2 persistence: we consistently detect a small fraction of double positive Sox2/HuC/D cells in chick spinal cord transverse sections. We have shown that this marker of differentiating neurons (HuC/D) only becomes detectable more than 8 hours after mitosis in newborn neurons at E3

(Baek et al, 2018), indicating that Sox2 protein can persist for up to at least 8 hours in newborn neurons.

We now cite a paper showing that a similar persistence of Sox2 protein is reported in differentiating neurons of the human neocortex, where double Sox2/NeuN positive cells are frequently observed in cerebral organoids (Coquand et al, Nature Cell Biology 2024) (quoted p.33, line 908)

- The undefined population (pRb-/HuCD-) introduces an unknown that assumes that the percentage of progenitors in G1 phase before the restriction point and the number of newborn neurons are equal for both conditions in an experiment. Can the authors provide explanation for this assumption?

We do not think that these numbers are equal for both conditions, and we did not formulate this assumption. We only indicate (in the methods section) that this undefined/undetermined population (based on negativity for both markers) is a mix of two possible cell types. However, we do not offer any interpretation of the Cdkn1c phenotypes based on the changes in this population. Indeed, our interpretation of the knock-down phenotype is solely based on the increase in pRb-positive and decrease in HuC/D-positive cells, which both suggest a delay in neurogenesis. We understand from the reviewer's comment that depicting an "undefined" population on the graph may cause some confusion. We therefore propose to present the data on pRb and HuC/D in different graphs, rather than on a combined plot, and to remove the reference to undefined cells in Figure 3, as well as in Figures 4 and 5 depicting the gain of function and double knock-down experiments. We have implemented these changes in the updated versions of the figures.

- In Gui et al. (Dev Biol 2006), authors showed that a knockdown of Cdkn1c leads to a failure of nascent neurons to exit the cell cycle and causes them to re-entry the cell cycle, shown by ectopic mitoses. In that study, cells born from those ectopic mitoses eventually leave the cell cycle leading to an increase in the number of neurons. Can the authors check for ectopic mitoses at 24hpe and 48hpe?

We have now performed experiments with an anti phospho Histone 3 antibody, which labels mitotic cells, at 24 and 48 hours post electroporation. We do not see any ectopic mitoses upon Cdkn1c knock-down with this marker, and we now mention these data. This is consistent with the fact that we also do not see ectopic pRb or Sox2 positive cells in the mantle zone in the knock-down experiments. These data (pH3 and Sox2 immunostainings) have been added in the new Figure EV 4G and H.

We have now modified the main text to include these data (p11, lines 278-287):

"In the context of a full knock-out of Cdkn1c in the mouse spinal cord, a reduction in neurogenesis was also observed, which was attributed to a failure of prospective neurons to exit the cell cycle, resulting in the observation of ectopic mitoses in the mantle zone (Gui et al, 2007). In contrast with this phenotype, using an anti phospho-Histone3 antibody, we did not observe any ectopic mitoses 24 or 48 hours after electroporation in our knock-down condition (Figure EV 4G-H). This is consistent with the fact that we also do not observe ectopic cycling cells with pRb (Figure 3A and D) and Sox2 (Figure EV 4G-H) antibodies. We therefore postulated that the reduced neurogenesis that we observe upon a partial Cdkn1c knock-down may result from a delayed transition of progenitors from the proliferative to neurogenic modes of division."

- The authors then address the question of whether the decrease in neuron number is due to the failure of newborn neurons to exit the cell cycle or to a delay in the transition from proliferative to neurogenic divisions. For that, they implement a strategy to label a synchronized cohort of progenitors based of incorporation of a FlashTag dye.

- Given that this strategy is the basis of many of the experiments in this article, it would be very valuable to expand on the validation of this technique as cited in major comment #2. In figure 3E, the close proximity of cell pairs in PP and PN clones shown in the pictures makes their sibling status apparent. However, this is not the case for the NN clone. Can the authors further explain with what criteria they determined the clonal status of two FlashTag labelled cells?

The key criterion for cells that are not directly touching each other is that their relative position corresponds to the classical “radial” organization of clones in this tissue (Leber and Sanes, 1995; Loulier et al, Neuron, 2014). In other words, we make sure that they are located on a same apico-basal axis, as is the case for the NN clone presented on the Figure. As stated above in our response to major comment #2, we have modified the Methods section accordingly.

Can they provide further image examples of different types of clones?

We now provide additional examples in a new Figure EV 5

Can the authors show that the plateau reached in Sup Figure 3 for pRb immunoreactivity corresponds to a similar dynamic for HuC/D immunoreactivity?

The plateau for Rb phosphorylation in progenitors is reached before 6 hours post mitosis at E3. At the same age, we have previously shown (Baek et al, PLoS Biology 2018) in a similar time course experiment in pairs of FT+ cells that the HuC/D signal is not detected in newborn neurons 8 hours after mitosis. HuC/D usually/mainly only starts to appear between 8 and 12 hours, and still increases between 8 and 16 hours. The plateau would therefore be very delayed for HuC/D compared to pRb. This long delay in the appearance of this « positive » marker of neural differentiation is the main reason why we chose to use Rb phosphorylation status for the analysis of synchronous cohorts of pairs of sister cells, because pRb becomes a discriminating factor much earlier than HuC/D after mitosis.

In order to further validate the strategy, could the authors use it at different stages to validate if they can replicate the different percentages of PP/PN/NN reported in the literature (e.g. Saade Cell Rep 2013)?.

We have carried out similar experiments at E2, showing a plateau of 95% of pRb-positive cells in the FT-positive population (see graph on the right). This provides a retrospective estimate of the mode of division of the mother cells at this stage (roughly 90% of PP and 10% of PN) which is consistent with the vast majority of PP divisions described by Saade et al (2013, see Figure S1) at this stage.

Figure for referee with unpublished data has been removed upon request by the authors.

5.- In Figure 4, the strategy used to induce a low-dose overexpression of Cdkn1c is an elegant

method to introduce Cdkn1c-Myc expression under the control of the endogenous Pax7 promoter, active in proliferative progenitors. The main point to address is:

- Please provide proof that Pax7 expression is not altered in guides with a successful knock-in event (e.g. sorting and WB against the Pax7 protein) or the immunohistochemistry as performed in the Pax7-P2A-Gal4 tagging in Petit-Vargas et al., 2024.

We have now performed Pax7 immunostainings on transverse sections at 24 and 48 hours post electroporation, both with the Pax7 -Gal4-Cdkn1c and with the Pax7-Gal4 control constructs. We present these data in the new Figure EV 7F-G. In both conditions, we find that the Pax7 protein is still present in KI-positive cells. We observed a modest increase in Pax7 signal intensity in these cells, suggesting either that the insertion of exogenous sequences stabilizes the Pax7 transcript, or that the C-terminal modification of Pax7 protein with the P2A tag increases its stability. This does not affect the interpretation of the Cdkn1c overexpression phenotype, because we used the Pax7-Gal4 construct that shows the same modification of Pax7 stability as a control for this experiment. We have introduced this comment in the legend of Figure EV 7F-G (p56, lines 1765-1769).

«A similar modest increase in Pax7 immunofluorescence is observed in knock-in cells (green, asterisks) in both conditions (compare top and bottom rows). Since this increase occurs irrespective of whether Cdkn1c is overexpressed or not, it does not affect the interpretation of phenotypes comparing the two conditions»

Given the cell cycle regulated expression and activity of Cdkn1c, can the authors elaborate on whether this is regulated at the promoter level?

Cdkn1c transcription is regulated by multiple transcription factors and non-coding RNAs (see for example Creff and Besson, 2020, or Rossi et al, 2018 for a review). To our knowledge, these studies focus more on the regulation of Cdkn1c global expression than on the regulation of its levels during cell cycle progression. Although it is very likely that transcriptional regulation contributes, post-translational regulation, and in particular degradation by the proteasome, is also a key factor in the cell cycle regulation of Cdkn1c activity

If so, how does this differ from the promoter activity of Pax7?

The transcriptional regulation of Pax7 and Cdkn1c is probably controlled by different regulators, since their expression profiles are very different. Regardless of the mechanisms that control their expression, the rationale for choosing Pax7 as a driver for Cdkn1c expression was that Pax7 expression precedes that of Cdkn1c in the progenitor population, and that it disappears in newborn neurons, when that of Cdkn1c peaks. This provided us with a way to advance the timing of Cdkn1c expression onset in proliferative progenitors.

- It would be advisable to characterize the dynamics along the cell cycle for the overexpressed form of Cdkn1c-Myc relative to pRb, similarly to what was done in Figure 2B.

We have carried out experiments similar to those shown in Figure 2B in order to characterise the dynamics of Cdkn1c-Myc in a context of overexpression. In addition, we have included a more precise quantification of the “misexpressed” compared to “endogenous” Cdkn1c-Myc levels, as already mentioned in the answer to a request by reviewer#1.

We have modified the main text accordingly as follows (page 12, lines 317-328):

“We used the CRISPR/Cas9-based somatic approach to introduce a sequence including a Myc-tagged *Cdkn1c* coding sequence and the Gal4-VP16 transcription factor downstream of *Pax7* (Figure 4B and Figure EV 7B-D). As a control for these experiments, we used a construct that introduces the Gal4-VP16 reporter alone to the *Pax7* locus (Petit-Vargas et al, 2024), allowing us to target and analyze the same cell population in control and overexpression conditions. We first confirmed that the expression of the *Cdkn1c*-Myc and of a UAS-nls-GFP reporter resulting from the *Cdkn1c*-Myc-P2A-Gal4 knock-in at the *Pax7* locus was restricted to progenitors in the dorsal domain (Figure EV 7E), with minimal perturbation of the *Pax7* protein expression (Figure EV 7F-G). Importantly, Myc signal intensity was similar to the one observed for endogenous Myc-tagged *Cdkn1c* in progenitors, and remained below the endogenous level of Myc-tagged *Cdkn1c* observed in nascent neurons, confirming the validity of our strategy (Figure 4C and Figure EV 7H-I). »

6.-In figure 5, the authors use a double knock-down strategy to test the hypothesis that the effect of *Cdkn1c* in G1 length is partially at least through its inhibition of *CyclinD1*. Results show that double shRNA-mediated knock-down of *CyclinD1* and *Cdkn1c* counteracts the effects of *Cdkn1c*-sh alone on EdU incorporation, PP/PN/NN cell divisions and overall ratios of progenitors and neurons.

- In the measurement of progenitor cell cycle length in Figure 5A, it would be more appropriate to present the nonlinear regression method described by Nowakowski et al. (1989), as has been commonly used in the field (Saade et al., 2013, PMID: 23891002, Le Dreau et al., 2014, PMID: 24515346, Arai et al., 2011, PMID: 21224845).

The Nowakowski non linear regression method has been used often in the literature in the same tissue, and is generally used to calculate fixed values for Tc, Ts, etc... This method is based on several selective criteria, and in particular the assumption that “all of the cells have the same cycle times”. Yet, many studies have documented that cell cycle parameters change during the transition from proliferative to neurogenic modes of division during which our analysis is performed; live imaging data in the chick spinal cord have illustrated very different cell cycle durations at a given time point (see Molina et al). We therefore think that the proposed formulas do not reflect the heterogenous reality of neural progenitors of the embryonic spinal cord. However, the cumulative approach described by Nowakowski is useful to show qualitative differences between populations (e.g. a global decrease of the cycle length, like in our comparison between control and shRNA conditions). For these reasons, we prefer to display only the raw measurements rather than the regression curves.

- Cumulative EdU incorporation in spinal progenitors (pRb-positive) at E3 (24 hours after injection) showed that the proportion of EdU-positive progenitors reached a plateau at 14 hours in control conditions, which is later than what has been reported in Le Dreau et al., 2014 (PMID: 24515346). Can you explain why?

Le Dreau et al count the EdU positive proportion of cells in the total population of electroporated cells located in the VZ (which includes progenitors, but also future neurons that have been labelled during the previous cycles -at least for the time points after 2hours- and have not yet translocated to the mantle zone), whereas we only consider pRb+ progenitors in the analysis. In addition, the experiments are not performed at the same developmental stage. Altogether, this may account for the different curves obtained in our study.

- It would be interesting to measure G1 length as in Figure 5D for the double Cdkn1c-sh - CyclinD1-sh knock down condition, to see if it rescues G1 length. As well as in the CyclinD1 knock down condition alone to see if it increases G1 length in this context as well.

We have performed cumulative EDU incorporation experiments similar to that shown in Figure 5D to measure the impact on G1 length of a double Cdkn1c-sh - CyclinD1-sh knock down, as well as of a CyclinD1 knock down condition alone. We chose the 6h30 time point for these analyses, as this is the first timepoint that shows a significant difference between CTRL and Cdkn1c shRNAs.

These data are now presented in the new Figure 6 panel B, and we have modified the main text to include these data as follows (page 15, lines 393-400):

“Remarkably, at 48 hae, whereas Cdkn1c shRNA and CyclinD1 shRNA alone respectively increased and decreased the proportion of EdU incorporation in pRb positive progenitors, the double Cdkn1c/CyclinD1 knock-down was indistinguishable from control (Figure 6A); in addition, the concomitant downregulation of CyclinD1 rescued the reduction in G1 length observed upon Cdkn1c knock-down (Figure 6B), as assessed in a FlashTag cohort by cumulative EdU incorporation in pRb+ cells 6h30 after mitosis. These data are consistent with the hypothesis that the effect of Cdkn1c on G1 duration is mediated by CyclinD1 inhibition.”

Minor comments

Introduction:

- The introduction should include references of studies of the role of Cdkn1c in cortical development (Imaizumi et al. *Sci Rep* 2020, Colasante et al. *Cereb Cortex* 2015, Laukoter et al. *Nature Communications* 2020).

In order to include the suggested references, we have modified the introduction (page 4, lines 77-87):

“Cdkn1c (Cyclin-dependent kinase inhibitor 1c/p57^{kip2}), a member of the CDK inhibitor (CKI) family that also comprises Cdkn1a (p21cip1) and Cdkn1b (p27kip1), emerged as one of the most differentially expressed between proliferative and neurogenic progenitors. In the mammalian cortex, p57^{kip2} plays a key role in regulation of the proliferation and differentiation of embryonic and adult neural stem cells (Colasante et al, 2015; Furutachi et al, 2013; Imaizumi et al, 2020; Itoh et al, 2007; Joseph et al, 2009, 2003; Mairet-Coello et al, 2012; Tury et al, 2011) and it was shown to promote gliogenesis during late embryogenesis and early postnatal stages (Tury et al, 2011). While in the spinal cord, Cdkn1c was previously described as a regulator of cell cycle exit via its expression in newborn neurons (Gui et al., 2007; Mairet-Coello et al., 2012; Tury et al., 2011), our results showed that it is already expressed at low level in neurogenic progenitors.”

and the results section (page 6, lines 132-151):

“Cdkn1c/p57kip2, an inhibitor of Cyclin/CDK complexes classically involved in G1 phase progression and G1 to S phase transition (Hatada & Mukai, 1995; Lee et al, 1995; Matsuoka et al, 1995; Taniguchi et al, 1997), was one of the strong candidates emerging from our analyses. Cdkn1c expression is

associated with the choice of cells to exit the cell cycle in many developmental and postnatal contexts (Creff & Besson, 2020). In addition, *Cdkn1c* also controls specific cell fates by directing differentiation in various tissues via CDK independent mechanisms (Campbell et al, 2020; Duquesnes et al, 2016).

The analysis of allele-specific mouse mutants of this imprinted gene has shown that both the paternal and maternal alleles contribute, though differently, to normal corticogenesis (Imaizumi et al, 2020). In addition, clonal analyses suggest that *Cdkn1c* regulates cortical development through distinct cell-autonomous and non-cell-autonomous mechanisms that respectively promote and inhibit growth (Laukotter et al, 2020). The *Cdkn1c* protein is present in some progenitors in the mammalian embryonic cortex (Mairet-Coello et al., 2012; Tury et al., 2011), and *Cdkn1c* transcripts are specifically enriched in neurogenic *Tis21* positive progenitors (Arai et al, 2011). An increased proliferation of cortical progenitors associated with a global shortening of G1 phase length has been reported in the *Cdkn1c* knock-out mouse (Mairet-Coello et al., 2012). Conversely, in the context of a cerebral cortex-specific inactivation of the *Aristaless*-related homeobox (*ARX*) gene, upregulation of *Cdkn1c* in the ventricular and subventricular zones was associated with reduced proliferation of intermediate progenitor cells at E12/14 (Colasante et al, 2015). »

1) Transcriptional signature of the neurogenic transition (Figure 1).

- In the result section, it would be informative to include the genes used to determine the progenitor and neuron score (instead of in Methods).

We have now listed the genes used to determine the progenitor and neuron score in the main text of the result section (p. 5, line 109).

“We therefore defined a scoring system based on the levels of expression of a list of progenitor and neuron-specific genes in each cell (Progenitor genes = *Sox2*, *Notch1*, *Rrm2*, *Hmgb2*, *Cenpa*, *Ube2c*, *Hes5*; Neuron genes = *Tubb3*, *Stmn2*, *Stmn3*, *Nova1*, *Rtn1*, *Mapt*).”

- Figure 1A. It would be informative to add in the diagram what "filtering" means (eg. Neural crest cells).

We have now added the detail of what 'filtering' means in the diagram.

- In the result section, "However, while *Tis21* expression is switched off in neurons, *Cdkn1c* transiently peaks at high levels in nascent neurons before fading off in more mature cells." Missing literary reference or data to clearly demonstrate this point.

We have reworded this sentence, adding a reference to the expression profile of *Tis 21*. The paragraph now reads as follows (p.7, lines 155-158):

« However, *Cdkn1c* expression is maintained longer and transiently peaks at high levels after *Tis21* expression is switched off. Given that *Tis21* is no more expressed in neurons (Iacopetti et al, 1999), this suggests that *Cdkn1c* expression is transiently upregulated in nascent neurons before fading off in more mature cells. »

- "Interestingly, the gene cluster that contained *Tis21* also contained genes encoding proteins with known expression and/or functions at the transition from proliferation to differentiation, such as

the Notch ligand Dll1, the bHLH transcription factors Hes6, NeuroG1 and NeuroG2, and the coactivator Gadd45g." Missing references.

We have now added references linking the function and/or expression profile of these genes to the neurogenic transition (p.5, line 122):

"the Notch ligand Dll1 (Henrique et al, 1995), the bHLH transcription factors Hes6 (Fior & Henrique, 2005), NeuroG1 and NeuroG2 (Lacomme et al, 2012; Sommer et al, 1996) and the coactivator Gadd45g (Kawaue et al, 2014). »

- There is an error in the color code in Cell Clusters in Figure 1C (cluster 4 yellow in the legend but ocre in the figure)

- Figure Sup3B colour code is switched (green for PP and red for NN) compared to the rest of the paper.

Thank you for pointing this out! We have corrected the color code errors in Figure 1C and Supp Figure 3B (now changed to Figure EV 6B in the modified revision).

It would be valuable to assign cell cycle stage to neural progenitor cells (based on cell cycle score) and determine whether Cdkn1c at the transcript level also shows enrichment in G1 cells considered to be progenitors.

We had refrained from performing the suggested combined analysis based on cell cycle and cell type scores, as the "neurogenic progenitor population" (based on neurogenic progenitor score values) in which Cdkn1c expression is initiated represents a small number of cells in our scRNAseq, and felt that the significance of such an analysis is uncertain. We have now performed this analysis, which shows a slightly higher Cdkn1c expression in G1 compared to S/G2/M phases in neurogenic progenitors. These data are now presented in new Figure EV 3 in the revised version and mentioned in the main text at the end of the section describing the dynamics of Cdkn1c in the neural tube (p.9, line 238):

"Indeed, we detect an overall lower expression in S/G2/M compared to G1 phase in neurogenic progenitors in the scRNAseq analysis (Figure EV 3)."

2) Progressive increase in Cdkn1c/p57kip2 expression underlie different cellular states in the embryonic spinal neural tube (Figure 2).

- Figure 2A. Scale bar is missing in E3 and E4. It is important to consider the growth of the developing spinal cord and present it accordingly (E3 transverse section, Figure 2).

The scale bar is actually valid for the whole panel A. The E2 section in the original figure appeared as "large" as the E3 section along the DV axis probably because the cutting angle was not perfectly transverse at E2, artificially lengthening the section. In a new version of the figure, we have replaced the E2 images with another section from the same experiment. The scale bar remains valid for the whole panel.

- Figure 2 could use a diagram of the knock-in strategy used, similar as the one in Figure 4A.

We have now added a diagram for the knock-in strategy in Figure 2B, and modified the legend of the Figure accordingly.

- Indicate hours post-electroporation. Indicate which guide is used in the main text.

We have now added the post-electroporation timing and guide used in the main text.

3) Downregulation of Cdkn1c in neural progenitors delays the transition from proliferative to neurogenic modes of division (Figure 3).

- In methods: "Thus, to reason on a more homogeneous progenitor population, we restricted all our analysis to the dorsal one half or two thirds of the neural tube." Indicate when and depending on what one half or two thirds of the neural tube were analysed. - Are the clonal analysis experiments (Fig 3D, E and F) also restricted to the dorsal region?

We have modified this sentence as follows (p31, lines 860-863):

"Thus, to reason on a more homogeneous progenitor population, we restricted all our analysis to the dorsal two thirds of the neural tube, except for the Pax7-Cdkn1c misexpression analysis, which was performed in the more dorsal Pax7 domain."

This is valid both for the whole population and clonal analyses.

- Figure 3. Would have a better flow if 3C preceded 3A and 3B.

We have modified the Figure accordingly.

- Figure 3C. it would be informative to show pictures of the electroporated NT at both 24hpe and 48hpe, as well as highlighting the dorsal part of the neural tube that was used for quantification.

We have modified Figure 3 accordingly (Figure 3A and 3D)

- In methods "At each measured timepoint (1h, 4h, 7h, 10h, 12h, 14 and 17h after the first EdU injection), we quantified the number of EdU positive electroporated progenitors (triple positive for EdU, pRb and GFP) over the total population of electroporated progenitor cells (pRb and GFP positive) (Figure 3B)." Explanation does not correspond to Figure 3B.

This explanation corresponds indeed to Figure 5A. We have corrected this mistake in the new version of the manuscript.

4) Inducing a premature expression of Cdkn1c in progenitors triggers the transition to neurogenic modes of division (Figure 4.).

- "We took advantage of the Pax7 locus, which is expressed in progenitors in the dorsal domain at a level similar to that observed for Cdkn1c in neurogenic precursors (Figure EV 4A)". Missing reference or data showing that Pax7 is restricted to the dorsal domain.

We have added references to the expression profile of Pax7 in the dorsal neural tube (Jostes et al, 1990) (p12, line 314).

"We took advantage of the Pax7 locus, which is expressed in progenitors in the dorsal domain (Jostes et al., 1990) at a level similar to that observed for Cdkn1c in neurogenic precursors (Figure EV 7A)."

In addition, the new Figure EV 7E shows anti-Pax7 staining that confirm this expression pattern at E3 and E4.

- "its intensity was similar to the one observed for endogenous Myc-tagged Cdkn1c in progenitors (Figure 4B and Figure EV 4E), and remained below the endogenous level of Myc-tagged Cdkn1c

observed in nascent neurons, confirming the validity of our strategy". It would be valuable to add a quantification to demonstrate this point, either by fluorescence levels or WB of nls-GFP cells.

As stated in the response to Major Point 5 above, we have performed a quantification based on Myc immunofluorescence to compare endogenous Cdkn1c expression versus Cdkn1c expression upon overexpression (Figure EV 7I).

- "At the population level, at E4, Cdkn1c expression from the Pax7 locus resulted in a strong reduction in the number of progenitors (pRb positive cells)". Indicate in the main text that this is 48hpe.

We have added in the main text that the quantification was performed 48hae.

- Legend of figure 4D should indicate that the quantification has been done 24hpe.

We have added the timing of quantification (30hae, 6 hours after FT injection) in the legend of Figure 4D (Now Figure 4F).

- "To circumvent the cell cycle arrest that is triggered in progenitors by strong overexpression of Cdkn1c (Gui et al., 2007)". It would be advisable to expand on this reference on the text, or ideally to include a simple Cdkn1c overexpression experiment.

These experiments have been performed and presented in the study by Gui et al., 2007, which we cite in the paper. Using a strong overexpression of Cdkn1c from the CAGGS promoter, they showed a massive decrease in proliferation, assessed by BrdU incorporation, 24hours after electroporation. We have now cited this result more explicitly in the main text, and better explained the difference of our approach. We have included the following modification (p12, lines 307-316):

« We next explored whether low Cdkn1c activity is sufficient to induce the transition to neurogenic modes of division. A previous study has shown that overexpression of Cdkn1c driven by the strong CAGGS promoter triggers cell cycle exit of chick spinal cord progenitors, revealed by a drastic loss of BrdU incorporation 1 day after electroporation (Gui et al, 2007). As this precludes the exploration of our hypothesis, we developed an alternative approach designed to prematurely induce a pulse of Cdkn1c in progenitors, with the aim to emulate in proliferative progenitors the modest level of expression observed in neurogenic progenitors (Figure 4A). We took advantage of the Pax7 locus, which is expressed in progenitors in the dorsal domain (Jostes et al, 1990) at a level similar to that observed for Cdkn1c in neurogenic precursors (Figure EV 7A). »

- "We observed a massive increase in the proportion of neurogenic (PN and NN) divisions rising from 57% to 84% at the expense of proliferative pairs (43% PP pairs in controls versus 16% in misexpressing cells, Figure 4D)." adding the percentages in the main text is a bit inconsistent with how the rest of the data is presented in the rest of the sections.

This whole section has been modified in response to a question from reviewer# 1. The new version does not contain percentages in the main text, and reads as follows (p. 13, line 336):

« Using the FlashTag cohort labeling approach described above, we traced the fate of daughter cells born 24 hae. We observed an increase in the proportion of terminal neurogenic (NN) divisions and a decrease in proliferative (PP) divisions (Figure 4F).»

- Figure sup 4C includes references to 3 gRNAs even when only one is used in the study.

Thanks for pointing this out, this was an oversight from us. The three guides listed in the original Supplementary Figure 4C correspond to the guides that we tested in Petit-Vargas et al. 2024. In this study, we only used the most efficient of these three guides (gRNA#2). We have modified the figure (now Figure EV 7C) by quoting only this guide.

5) The proneurogenic activity of Cdkn1c in progenitors is mediated by modulation of cell cycle dynamics (Figure 5)

- **"we targeted the CyclinD1/CDK4-6 complex, which promotes cell cycle progression and proliferation, and is inhibited by Cdkn1c." reference missing**

We have included references related to the activity of the CyclinD1/CDK4-6 complex in the developing CNS, and the antagonistic activities of CyclinD1 and Cdkn1c in this model (page 14, lines 383-386):

" we targeted the CyclinD1/CDK4-6 complex, which promotes cell cycle progression and proliferation in the developing CNS (Gui et al, 2007; Lange et al, 2009; Lobjois et al, 2008, 2004a) , and is inhibited by Cdkn1c (Gui et al, 2007). »

- **It would be informative to include experimental set-up information (e.g. hae) in Figures 5A, 5B, 5F and 5G.**

We have added the experimental set-up information in Figure 5A,B and Figure 6A-C (ex Figure 5F-G).

- **Clarify if analysis is restricted to the dorsal progenitors or the whole dorsoventral length of the tube.**

The analyses were carried out on two thirds of the neural tube (dorsal 2/3), excluding the ventral zone, as specified above (and in the Methods section).

- **It would be valuable to add an image to illustrate what is quantified in Figure 5D, Figure F and Figure G.**

- **For Figure 4C and D, it would be valuable to add images to illustrate the quantification.**

We have added images:

- in Figure EV 8A to illustrate what is quantified in Figures 4C (now 4D and 4E);
- In Figure 5E to illustrate what is quantified in Figure 5D
- In Figure EV 9B to illustrate what is quantified in Figure 5G (now Figure 6D and 6E)

In addition, the new analyses quantified in Figure 4G are illustrated by representative images shown in Figure EV 8B.

Regarding the requested images for Figures 4D and 5F, they correspond to the same types of images already shown in Figure 3E. Since we have now added several additional examples of representative pairs of each type of mode of division in the new Figure EV 5, we do not think that adding more of these images in figures 4 and 5 would strengthen the result of the quantifications.

Discussion:

- **"Nonetheless, studies in a wide range of species have demonstrated that beyond this binary choice, cell cycle regulators also influence the neurogenic potential of progenitors, i.e the commitment of their progeny to differentiate or not (Calegari and Huttner, 2003; FUJITA, 1962;**

Kicheva et al., 2014; Lange et al., 2009; Lukaszewicz and Anderson, 2011a; Pilaz et al., 2009; Smith and Schoenwolf, 1987; Takahashi et al., 1995)." Should include maybe references to *Peco et al. Development 2012, Roussat et al. J Neurosci. 2023).*

We have now included the references suggested by the reviewer.

- "This occurs through a change in the mode of division of progenitors, acting primarily via the inhibition of the CyclinD1/CDK6 complex." The data shown in the paper does not demonstrate that Cdkn1c is inhibiting CyclinD1, only that knocking down both mRNAs counteracts the effect of knocking down Cdkn1c alone at the general tissue level and in the percentage of PP/PN/NN clones. This statement should be qualified.

We have reformulated this paragraph in the discussion as follows to take this remark into account (p.15, line 420):

"This allows us to re-interpret the role of Cdkn1c during spinal neurogenesis: while previously mostly considered as a binary regulator of cell cycle exit in newborn neurons, we demonstrate that Cdkn1c is also an intrinsic regulator of the transition from the proliferative to neurogenic status in cycling progenitors. This occurs through a change in their mode of division, and our double knock-down experiments suggest that the onset of Cdkn1c expression may promote this change by counteracting a CyclinD1/CDK6 complex dependent mechanism."

Other comments:

- To improve clarity for the reader, it would help if electroporation was shown consistently on the same side of the neural tube. If electroporation has been performed at different sides and this is reflected in the figures, it would be advisable to explain on the figure legend.

We have modified the figures to systematically show the electroporated side of the neural tube on the same side of the image for single electroporations.

- Figure legends should include the number of embryos/tissue sections analysed for each experiment, as well as information on whether the sections were cryostat or vibratome.

This information is now provided in the figure legends (numbers of cells analysed and/or numbers of embryos), except for data in Figure 5 (now split between new Figures 5 and 6), which are presented in a new Supplementary Table 1.

All experiments were performed on vibratome sections, except for in situ hybridization experiments, which were performed on cryostat sections. This last information was already indicated in the relevant figure legends.

- Overall, there is a lack of consistency in the figures regarding how much information is available to the reader (e.g. Sup Figure 2A, in the panel mRNA in situ hybridisation of Cdkn1c is referred to only as Cdkn1c whereas in Sup figure 5 the in situ reads as CYCLIND1 mRNA). Readability would improve a lot if figures included information on what is an electroporated fluorescent tag or an immunostaining (similar to the label in sup 4D) as well as the exact stage and hours after electroporation where relevant.

- There is a general lack of consistency in indicating the timing of the experiments, both in terms of embryonic stage/day and in terms of hours-post-electroporation.

We have now homogenized the nomenclature in the figures.

- *"Primary antibodies used are: chick anti-GFP (GFP-1020 - 1:2000) from Aves Labs; goat antiSox2 (clone Y-17 - 1:1000) from Santa Cruz". There is no Sox2 immunostaining in the article.*

In the original version of the manuscript, the anti-Sox2 antibody was not used; we have now added experiments using this antibody in the modified version of the manuscript; this sentence in the Methods thus remains unchanged.

Reviewer3 (Significance (Required)):

Significance:

In neural development, there is a progressive switch in competence in neural progenitor cells, that transition from a proliferative (able to expand the neural progenitor pool) to neurogenic (able to produce neurons). Several factors are known to influence the transition of neural progenitor cells from a proliferative to a neurogenic state, including the activity of extracellular signalling pathways (e.g. SHH) (Saade et al. 2013, Tozer et al. 2017). In this study, the authors perform scRNA-seq of the cervical neural tube of chick at a stage of both proliferative and neurogenic progenitors are present, and identify transcriptional differences between the two populations. Among the differently expressed transcripts, they identify Cdkn1c (p57-Kip2) as enriched in neurogenic progenitors. Initially characterized as a driver of cell cycle exit in newborn neurons, the authors investigate the role of Cdkn1c in cycling progenitors.

The authors find that knock-down of Cdkn1c leads to an increase in proliferative divisions at the expense of neurogenic divisions. Conversely, misexpression of Cdkn1c in proliferative progenitors leads to a switch to neurogenic divisions. Furthermore, they find that knock-down of Cdkn1c shortens G1 phase of the cell cycle, suggesting a link between G1 length and neurogenic competence in neural progenitor cells. Cell cycle length has previously been linked to competence of neural progenitors, and it has been described that longer G1 duration is linked to neurogenic competence (e.g. Calegari F, Huttner WB. 2003).

The strengths of the study include:

The identification of a subset of genes enriched in neurogenic vs. proliferative progenitors. Since the transition from proliferative to neurogenic competence is a gradual process at the tissue level, the classification of proliferative vs. neurogenic progenitors based on a score of transcripts and the identification of a subset of transcripts that are enriched in neurogenic progenitors is a valuable contribution to the neurodevelopmental field.

- The somatic knock-in strategy used to induce low-level overexpression of Cdkn1c in proliferative progenitors is an elegant strategy to induce overexpression in a subset of cells in a controlled manner and is a valuable technical advance.

- The characterization of a specific role of Cdkn1c in regulating cell cycle length in cycling progenitors is novel and valuable knowledge contributing to our understanding of how regulation of cell cycle length impacts competence of neural progenitors.

The aspects to improve:

- The sc-RNAseq isolated genes enriched in neurogenic versus proliferative progenitors, providing valuable insight into the gradual transition from proliferative to neurogenic competence at the tissue level. However, this gene subset requires clearer representation and detailed characterization. Additionally, the full scRNA-seq dataset should be made publicly available to support further research in neurodevelopment.

The sequencing dataset has been deposited in NCBI's Gene Expression Omnibus database. It is currently under embargo, but will be made available upon acceptance and publication of the peer reviewed manuscript. Access is nonetheless available to the reviewers via a token that can be retrieved from the Review Commons website.

The following information has been added in the revised manuscript:

"DATA AVAILABILITY"

Single cell RNA sequencing data have been deposited in NCBI's Gene Expression Omnibus (GEO) repository under the accession number GSE273710, and are available at <https://www.ncbi.nlm.nih.gov/geo/query/acc.cgi?acc=GSE273710>.

- The characterization of Cdkn1c dynamics in cycling progenitors using endogenous tagging of the Cdkn1c transcript with a Myc tag is an elegant way to investigate the dynamics of Cdkn1c-myc along the cell cycle. However, it would be much more powerful if combined with a careful characterization of pRb immunostaining along the cell cycle in this tissue, as well as the quantifications and controls proposed.

- Retinoblastoma protein (Rb) and cyclin D play a key role in regulating the G1/S transition, with cyclin D/CDK complexes phosphorylating Rb. Given that Cdkn1c primarily inhibits the cyclin D/CDK6 complex, it likely affects pRb expression or phosphorylation. This suggests pRb may be a direct target of Cdkn1c, making it an unreliable marker for tracking and quantifying neurogenic progenitors through Cdkn1c modulation. In light of this, it would be more appropriate to consider pRb as a Cdkn1c target and discuss the molecular mechanisms regulating cell cycle components. A more precise approach would involve using other markers or targets to quantify neural precursor division modes at earlier stages of neurogenesis.

- Many of the conclusions of the study are based on experiments performed using the FlashTag dye in order to perform clonal analysis of proliferative vs. neurogenic divisions. It would be very valuable to further characterize the reliability of this tool as well as to provide more information on the criteria used to determine the fate of the pairs of sister cells.

- The somatic knock-in strategy used to induce low-level overexpression of Cdkn1c in proliferative progenitors is an elegant strategy to induce overexpression in a subset of cells in a controlled manner. It would be valuable to further characterize the dynamics of Cdkn1c expression using this too and to provide proof that Pax7 expression is not altered in guides with the knock-in event.

- The presentation of the existing literature could be more up to date.

- The presentation of the data in the figures could be improved for readability. The sc-RNA seq data and the technical advances could be of interest for an audience of researchers using chick as a model organism, and working on neurodevelopment in general. Furthermore, the characterization of Cdkn1c as a regulator of G1 length in cycling progenitors and its implications for

neurogenic competence could be of general interest for people working on basic research in the neurodevelopmental field.

Field of expertise of the reviewer: neural development, cell biology, embryology.

Dear Dr. Morin,

Thank you for the submission of your revised manuscript. We have now received the enclosed reports from the referees and I am happy to say that all support its publication now. Please address the last comments by referee 3, and also some editorial requests will need to be addressed before we can proceed with the official acceptance of your manuscript.

- The keywords need to be placed after the Abstract.
- The conflict of interest subheading needs to be corrected to "Disclosure and Competing Interests Statement"
- The author credits need to be removed from the ms file. All credits need to be entered during online ms submission.
- DATA NOT SHOWN is mentioned on page 29 but not allowed per journal policy. Please either show the data or re-phrase.
- In the author checklist, please select a response in the cell D66 and send us a fully completed checklist with your final ms.
- Some FUNDING INFO is missing in our online ms submission system: the Institut Universitaire de France, from the French Ministry of Higher Education and Research (MESR), «Investissements d'Avenir» launched by the French Government. All funding info should be listed both in our online submission system and in the ms file. Additional funders can be added via More Funders and the Comments box should not be used.
- A callout for Figure EV5 is missing in the ms text, please add.
- Please remove the Reagents & Tools table from the ms file and upload it as a separate file.
- MATERIALS AND METHODS should be METHODS
- The table in the ms should either be renamed to Table 1 and placed between main and EV figure legends or removed from the ms and uploaded as Table EV1 (expanded view content file type). The callouts in the ms file also need to be updated.

* Figure Legends - Comments *

- Please define the annotated p values ****/****/**/* as well as provide the exact p-values for the same in the legend of figure EV6 as appropriate and reasonable.
- Please note that the exact p values are not provided in the legends of figures 3B, C, E, F; 4D-G; 5A, D; 6A-E; EV4 C, D, E, please add exact values as reasonable.
- Please indicate the statistical test used for data analysis in the legends of figures 3C, EV3, EV6
- Please note that the box plots need to be defined in terms of minima, maxima, centre, bounds of box and whiskers, and percentile in the legend of figure EV3
- Please note that information related to n is missing in the legend of figure EV3, both the number and the nature of n (technical versus biological) needs to be listed in all figure legends.

EMBO press papers are accompanied online by A) a short (1-2 sentences) summary of the findings and their significance, B) 2-3 bullet points highlighting key results and C) a synopsis image that is exactly 550 pixels wide and 200-600 pixels high (the height is variable). The synopsis image should provide a sketch of the major findings, like a graphical abstract. Please note that text needs to be readable at the final size. Please send us this information along with the final manuscript.

Best regard,
Esther

Referee #1:

The authors addressed all my comments and concerns. Excellent manuscript.

Referee #2:

After careful review of the author's response and the changes introduced in the manuscript, we are happy to say that the authors have satisfactorily addressed the points we raised.

In particular, we want to highlight the improved characterization of Cdkn1c expression dynamics, as well as the validation of the knock-in strategy as a faithful tool to investigate expression dynamics. We also appreciate the effort put into further describing the methodology behind the use of FlashTag as a tool to determine cell division outcomes. Moreover, we want to mention the higher clarity of the figures, helped by the use of experimental diagrams, and the better readability of the manuscript.

Referee #3:

We are happy to report that the authors have addressed our major concerns and are delighted to recommend this manuscript for publication once they have addressed our minor comments.

- Figure EV8B. The annotations in the far right panels (in greyscale) are difficult to see. The authors might consider changing the colours of these.
- Figure EV4D, E. The authors should provide precise embryo and cell numbers for each experimental condition, rather than overall ranges as currently presented.

All editorial and formatting issues were resolved by the authors.

Dr. Xavier Morin
Institut de Biologie de l'École Normale Supérieure
Developmental Biology
CNRS UMR8197
46, rue d'Ulm
Paris 75005
France

Dear Dr. Morin,

I am very pleased to accept your manuscript for publication in the next available issue of EMBO reports. Thank you for your contribution to our journal.

Yours sincerely,
